# Identifiable Markov Switching Models
# with Instantaneous Effects and Exponential Families

**Roel Hulsman** [1 2]   **Carles Balsells-Rodas** [3]   **Sara Magliacane** [1 4]

## Abstract

Temporal systems often exhibit non-stationary behaviour, such as seasonal climate variation or glucose fluctuations in patients with type-1 diabetes. One way to model non-stationarity is through discrete latent *regimes*, *i.e.*, stationary segments of time. Such systems induce a *Markov Switching Model* (MSM), a class of Hidden Markov Models with autoregressive dependencies among latent regimes and observed variables. Identifying latent regimes is challenging in the presence of frequent regime switches and nonlinear and non-Gaussian dynamics, particularly when there are *instantaneous effects* between the variables, *e.g.*, due to slow rates of measurements. In this work, we establish the identifiability of both latent regimes and regime-dependent causal structures under temporal regime dependencies, nonlinear lagged and instantaneous effects, and independent noise from the exponential family. Our identifiability theory subsumes non-temporal mixtures of causal models. Furthermore, we introduce `FlowMSM`, a regime detection framework that can be paired with any stationary causal discovery method to recover regime-dependent causal structures. Experiments on synthetic benchmarks and a financial economics dataset demonstrate the effectiveness of our approach to detect latent regimes and discover causal structures from non-stationary time series.

## 1. Introduction

Non-stationarity is ubiquitous in real-world environments, ranging from sleep-stage dynamics in Electroencephalography (EEG) (Shah et al., 2018) to the El Niño-Southern

[1]University of Amsterdam [2]Adyen [3]Imperial College London [4]Saarland University. Correspondence to: Roel Hulsman <r.p.hulsman@uva.nl>.

*Proceedings of the 43rd International Conference on Machine Learning*, Seoul, South Korea. PMLR 306, 2026. Copyright 2026 by the author(s).

Oscillation (ENSO) in climate science (Chalupka et al., 2016). A common modelling strategy is to represent non-stationarity through discrete latent *regimes*, describing stationary segments of time. Detecting latent regimes is challenging when regimes switches are frequent, the dynamics are nonlinear and non-Gaussian, and there are *instantaneous* effects between the observed variables, *e.g.*, as a result of coarse measurement granularity. For example, for patients with type-1 diabetes, the effect of insulin delivery on glucose metabolism is typically faster than the five-minutes sampling frequency of glucose monitors, thus emerging as instantaneous in measurements.

Autoregressive regime switches and observed variables give rise to *Markov Switching Models* (MSMs) (Hamilton, 1989), also known as autoregressive Hidden Markov Models (HMMs) (Poritz, 1982). MSMs are widely used in time-series modelling, with established applications in bioinformatics (Koski, 2001), financial economics (Bhar & Hamori, 2004; Ang & Timmermann, 2012) and speech analysis (Ephraim & Roberts, 2005), as well as contemporary use in energy studies (Cevik et al., 2021) and health economics (Anser et al., 2021). However, classic identifiability results for finite-state HMMs (Kruskal, 1977; Allman et al., 2009; Gassiat et al., 2016) do not trivially extend to MSMs due to the autoregressive dependencies among observed variables. Identifiability theory here characterises when the data likelihood uniquely determines latent regimes.

We build on recent identifiability results for first- and higher-order MSMs without instantaneous effects (Balsells-Rodas et al., 2024; 2026). However, these works only provide concrete instantiations of their assumptions in Gaussian settings. We draw inspiration from Barndorff-Nielsen (1965) to extend identifiability to exponential families, under possibly nonlinear lagged and instantaneous effects. Once latent regimes are identifiable, existing causal discovery theory recovers regime-dependent causal structures, for example up to a Markov equivalence class (MEC) (Spirtes et al., 2000).

Some recent studies address closely related problems (Rahmani & Frossard, 2025a;b). These works, however, assume regimes to be deterministic functions of time, thereby excluding autoregressive or state-dependent regime dynamics. In this work, we do allow for such dependencies. In

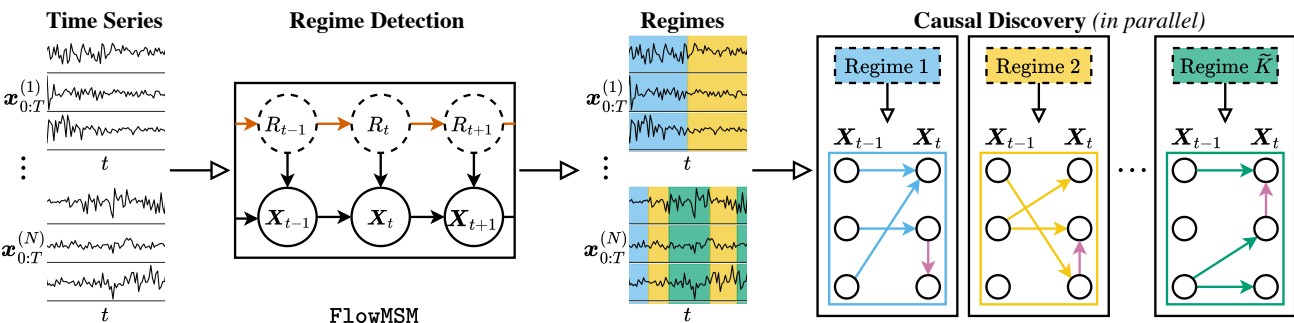

*Figure 1.* FlowMSM detects *latent regimes* $\boldsymbol{R}_{0:T}$ from time series $\boldsymbol{X}_{0:T}$, generated by a *regime-switching SCM* with dependencies between regimes (*i.e.*, the **red** edges), instantaneous effects among observed variables (*e.g.*, the **pink** edges) and independent exponential family noise. Subsequently, a causal discovery method discovers *window graphs* from regime-dependent sample sequences.

addition, the proposed methods alternate between regime detection and causal discovery within the class of *location scale noise models* (LSNMs) (Immer et al., 2023; Strobl & Lasko, 2023), fundamentally relying on identifiability of the underlying causal structure. Moreover, during the iteration process, the authors leverage causal discovery methods that assume *causal sufficiency*, which might not hold while learning the regime assignments, since the regime variable could act as a latent confounder. In contrast, our framework decouples regime detection from causal discovery, thereby avoiding the aforementioned limitations and allowing for a more broad class of causal models.

We summarise our main contributions below and provide an overview of our approach in Fig. 1.

- We establish identifiability for a broad class of regime-switching SCMs with dependencies between regimes, nonlinear lagged and instantaneous effects, and independent noise from the exponential family (Sec. 3).
- We introduce FlowMSM, a regime detection method based on conditional normalising flows, which extends to regime-dependent causal discovery (Sec. 4).
- Through experiments on synthetic benchmarks, we show accurate regime detection and causal discovery. Additionally, we present a qualitatively study of the Fama-French five-factor model (Fama & French, 2015) in financial economics (Sec. 6).

## 2. Background

We use **bold** to indicate multivariate vectors and vector-valued functions. We use uppercase letters for random variables, and lowercase letters for their instantiations. We write $\mathcal{X}^{\times T} \triangleq \mathcal{X} \times \cdots \times \mathcal{X}$ to denote a Cartesian product of $T$ sets $\mathcal{X}$. Let $\boldsymbol{X}_{0:T} = (\boldsymbol{X}_0, \ldots, \boldsymbol{X}_T) \in \mathcal{X}^{\times(T+1)}$, with $\mathcal{X} \subseteq \mathbb{R}^D$, denote a continuous, $D$-dimensional time series observed over an initial time step and $T$ subsequent time steps, where $D, T < \infty$. Let $\boldsymbol{R}_{0:T} \in \mathcal{A}_K^{\times(T+1)}$ denote discrete latent regime variables. Here $\mathcal{A}_K$ is a finite subset of

cardinality $K < \infty$, drawn from a countably infinite index set $\mathcal{A}$. For example, $\mathcal{A}$ may index a family of probability density functions (PDFs) $\{p_{\boldsymbol{\theta}}(\boldsymbol{x} \mid a) \mid a \in \mathcal{A}\}$ with parameters $\boldsymbol{\theta} \in \Theta$, where $\boldsymbol{x}$ is a realisation of $\boldsymbol{X}$. We write $\mathcal{P}^{\otimes T} \triangleq \mathcal{P} \otimes \cdots \otimes \mathcal{P}$ as a pointwise product of $T$ function families $\mathcal{P}$. We use the superscript $^0$ to denote objects specific to the initial time step. The notation $\boldsymbol{f}|_{\boldsymbol{X}=\boldsymbol{x}}$ specifies a function $\boldsymbol{f}$ evaluated at the fixed argument $\boldsymbol{X} = \boldsymbol{x}$. Finally, we often write shorthand $\boldsymbol{f}_a \triangleq \boldsymbol{f}|_{R_t=a}$ for regime-dependent mappings and regime-indexed quantities.

### 2.1. Structural Causal Models (SCMs)

We assume the data generating process can be described by a *regime-switching* version of an SCM (Pearl, 2009), where the structural equations encode the causal relations between causal variables and exogenous noises, and any non-stationary behaviour of the observed variables is directed by discrete latent regimes. Importantly, we presume no knowledge of the number of regimes $K$, nor of the structural equations governing latent regime dynamics, besides that regimes do not depend on past observations.

We write the *initial* and *transition* structural equations as

$$\boldsymbol{X}_0 \leftarrow \boldsymbol{f}^0\big(\mathbf{Pa}^0(R_0), R_0, \boldsymbol{\epsilon}_0\big), \tag{1}$$
$$\boldsymbol{X}_t \leftarrow \boldsymbol{f}\big(\mathbf{Pa}(R_t), R_t, \boldsymbol{\epsilon}_t\big), \qquad t \in \{1, \ldots, T\},$$

where $\boldsymbol{f}^0, \boldsymbol{f}$ are measurable time-invariant functions and $\boldsymbol{\epsilon}_t \in \mathcal{X}_\epsilon \subseteq \mathbb{R}^D$ is *i.i.d.* exogenous noise. The causal parents $\mathbf{Pa}^0(\cdot) \subset \boldsymbol{X}_0$ and $\mathbf{Pa}(\cdot) \subset \boldsymbol{X}_{t-1:t}$ consist of a subset of causal variables acting as direct causes in a particular regime, which we do not require to be unique across regimes. We highlight that the causal parents might include variables at the same time step, permitting instantaneous effects. Furthermore, for a given regime value $a \in \mathcal{A}_K$, the structural equations implicitly define measurable mappings $\Phi_a^0(\boldsymbol{\epsilon}_0)$ and $\Phi_a(\cdot, \boldsymbol{\epsilon}_t)|_{\boldsymbol{X}_{t-1}=\boldsymbol{x}_{t-1}}$ from the noise space $\mathcal{X}_\epsilon$ to the observed space $\mathcal{X}$ at each time step. Note that this formulation includes mixtures of stationary time series as a special case by restricting $R_t = R_{t'}$ for any $t \neq t'$.

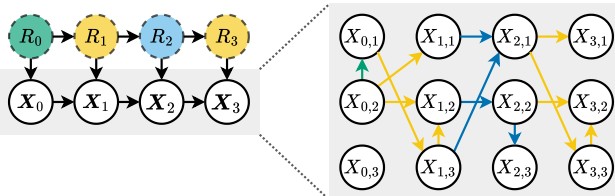

*Figure 2.* Example causal graph given regimes $\boldsymbol{R}_{0:3}$, with coloured edges belonging to regime-dependent initial and window graphs.

**Window Graphs.** The regime-switching SCM induces *window graphs* $G_a$ (Assaad et al., 2022), representing the instantaneous and transition dynamics in a particular regime. That is, the regime $R_t$ acts as a latent confounder for the variables $\boldsymbol{X}_t$, and the window graph $G_a$ contains the edges from causal parents $\mathbf{Pa}_a$ to variables $\boldsymbol{X}_t$ given the regime $R_t = a$. Analogously, the regime-switching SCM induces *initial graphs* $G_a^0$, but we omit a discussion for brevity. Fig. 2 visualises examples. We do not require initial nor window graphs to be unique across regimes.

**Assumptions.** We assume *conditional causal stationarity*, *i.e.*, that causal mechanisms are stable within each regime, and regime switches fully govern structural changes over time. In addition, we assume that within each regime there are no latent confounders or selection bias, which we call *conditional causal sufficiency*. As standard in causal discovery, we assume the *causal Markov* and *faithfulness* conditions hold within each regime (Spirtes et al., 2000), which together imply that conditional independences correspond to d-separations in the underlying window graph, and consider only *acyclic* SCMs. For simplicity, in the main paper we restrict causal parents to be of *maximum lag* $L = 1$, but this is relaxed to an arbitrary number of lags in the theory, method and experiments presented in App. B to D.

Although our identifiability theory does not restrict the latent regime dynamics beyond its independence of the observed variables, for estimation purposes we assume a *first-order stationary Markov chain*. Similarly, we assume the mappings $\boldsymbol{f}_a^0, \boldsymbol{f}_a$ are *contractive* to guarantee stability of the induced time series and enable likelihood-based estimation. In linear SCMs, this restricts the spectral radius of the regime-dependent companion matrices (Lütkepohl, 2005).

**Exponential Families.** We restrict exogenous noise to belong to an exponential family. The PDF of a continuous random variable $\boldsymbol{\epsilon} \in \mathcal{X}_\epsilon$ from a minimal regular exponential family in canonical form (Brown, 1986) is

$$p_{\boldsymbol{\eta}}(\boldsymbol{\epsilon}) = h(\boldsymbol{\epsilon}) \exp\left(\boldsymbol{\eta} \cdot \boldsymbol{\tau}(\boldsymbol{\epsilon}) - A(\boldsymbol{\eta})\right), \quad (2)$$

for natural parameters $\boldsymbol{\eta} \in \Omega$, base measure $h : \mathcal{X}_\epsilon \to \mathbb{R}_+$, sufficient statistic $\boldsymbol{\tau} : \mathcal{X}_\epsilon \to \mathbb{R}^P$, $P \geq 1$, and log-partition $A : \Omega \to \mathbb{R}$. The natural parameter space $\Omega \subseteq \mathbb{R}^P$ is open and the support $\mathcal{X}_\epsilon \subseteq \mathbb{R}^D$ does not depend on $\boldsymbol{\eta}$.

## 2.2. Markov Switching Models (MSMs)

The dynamic process induced by a regime-switching SCM, for a particular configuration of the index set $\mathcal{A}_K$ and the latent regime dynamics, is an MSM. The joint distribution can be formulated as a finite mixture over discrete regimes (Frühwirth-Schnatter, 2006), *i.e.*,

$$p_{\boldsymbol{\theta}}(\boldsymbol{x}_{0:T}) = \sum_{\boldsymbol{r}_{0:T} \in \mathcal{A}_K^{\times(T+1)}} p_{\boldsymbol{\theta}}(\boldsymbol{r}_{0:T}) p_{\boldsymbol{\theta}}(\boldsymbol{x}_{0:T} \mid \boldsymbol{r}_{0:T}), \quad (3)$$

where the $K^{T+1}$ unique sequences of regimes $\boldsymbol{r}_{0:T}$ determine the components of the mixture. Assuming conditional causal stationarity and causal parents of at most a single lag, we can further factorise

$$p_{\boldsymbol{\theta}}(\boldsymbol{x}_{0:T} \mid \boldsymbol{r}_{0:T}) = p_{\boldsymbol{\theta}}(\boldsymbol{x}_0 \mid r_0) \prod_{t=1}^{T} p_{\boldsymbol{\theta}}(\boldsymbol{x}_t \mid \boldsymbol{x}_{t-1}, r_t). \quad (4)$$

Throughout this work, we refer to $p_{\boldsymbol{\theta}}(\boldsymbol{x}_0 \mid r_0) \in \mathcal{P}_{\mathcal{A}}^0$ as the *initial distribution* and to $p_{\boldsymbol{\theta}}(\boldsymbol{x}_t \mid \boldsymbol{x}_{t-1}, r_t) \in \mathcal{P}_{\mathcal{A}}$ as the *transition distribution*. The corresponding families $\mathcal{P}_{\mathcal{A}}^0, \mathcal{P}_{\mathcal{A}}$ contain countably infinite distributions indexed by $\mathcal{A}$, of which a finite subset, indexed by $\mathcal{A}_K$, appears in a particular regime-switching SCM. For simplicity, the same set $\mathcal{A}_K$ indexes regimes for all time steps. However, the techniques discussed in this work generalise beyond such settings. For example, one could choose a time-indexed number of regimes $K_t$ and time-indexed distribution families $\mathcal{P}_{\mathcal{A}}^t$. These generalisations are discussed in Balsells-Rodas et al. (2026).

**Identifiable MSMs.** Identifiability of finite mixture distributions amounts to a one-to-one relation between the observed data likelihood $p_{\boldsymbol{\theta}}(\boldsymbol{x}_{0:T})$ and the latent regime prior and mixture components, up to a relabelling of the regimes. Following the factorisation in Eq. (4), equality of mixture components can be specified into equal initial and transition distributions. Identifiable initial regimes and regime transitions follow from a chain rule factorisation of the latent regime prior. Our Def. 2.1 simplifies notation compared to Balsells-Rodas et al. (2024, Def. 3.1) by using the regime $r_t$ as index directly to omit a path index function. Leaving a permutation equivalence is standard in the literature (Yakowitz & Spragins, 1968, Eq. (3)).

**Definition 2.1** (Identifiability Up To Permutation). An MSM is said to be *identifiable up to permutation* if for any $p_{\boldsymbol{\theta}}, p'_{\boldsymbol{\theta}'}$ with regime index sets $\mathcal{A}_K, \mathcal{A}'_{K'}$ as in Eq. (3), we have $p_{\boldsymbol{\theta}} = p'_{\boldsymbol{\theta}'}$ *almost everywhere*, or *a.e.*, implies $K = K'$ and there exist a label permutation $\phi : \mathcal{A}_K \to \mathcal{A}'_{K'}$, such that for almost every $\boldsymbol{x}_{t-1} \in \mathcal{X}$, $t \in \{1, \dots, T\}$:

1. $p_{\boldsymbol{\theta}}(r_0) = p'_{\boldsymbol{\theta}'}(\phi(r_0))$ *a.e.*;
2. $p_{\boldsymbol{\theta}}(r_t \mid \boldsymbol{r}_{0:t-1}) = p'_{\boldsymbol{\theta}'}(\phi(r_t) \mid \phi(r_0), \dots, \phi(r_{t-1}))$ *a.e.*;
3. $p_{\boldsymbol{\theta}}(\boldsymbol{x}_0 \mid r_0) = p'_{\boldsymbol{\theta}'}(\boldsymbol{x}_0 \mid \phi(r_0))$ *a.e.*;
4. $p_{\boldsymbol{\theta}}(\boldsymbol{x}_t \mid \boldsymbol{x}_{t-1}, r_t) = p'_{\boldsymbol{\theta}'}(\boldsymbol{x}_t \mid \boldsymbol{x}_{t-1}, \phi(r_t))$ *a.e.*.

Identifiability of finite mixture distributions is established via linear independence of the family of mixture components (Yakowitz & Spragins, 1968). A family of PDFs contains linear independent functions if for any finite $\mathcal{A}_K \subset \mathcal{A}$,

$$\sum_{a \in \mathcal{A}_K} \lambda_a p_{\boldsymbol{\theta}}(\boldsymbol{x} \mid a) = 0 \quad a.e. \implies \lambda_a = 0 \quad \forall a \in \mathcal{A}_K.$$

For MSMs, the family of mixture components is the *product family* $\mathcal{P}_{\mathcal{A}}^0 \otimes \mathcal{P}_{\mathcal{A}}^{\otimes T}$. However, linear independence in the product family is not guaranteed by linear independence in the marginal families $\mathcal{P}_{\mathcal{A}}^0$ and $\mathcal{P}_{\mathcal{A}}$, by virtue of an overlapping variable space, challenging linear independence for any consecutive product of linearly independent distributions. That is, for $T = 1$, with overlapping variable $\boldsymbol{x}_0$, we have

$$\mathcal{P}_{\mathcal{A}}^0 \otimes \mathcal{P}_{\mathcal{A}} = \left\{ p_{\boldsymbol{\theta}}(\boldsymbol{x}_0 \mid r_0) p_{\boldsymbol{\theta}}(\boldsymbol{x}_1 \mid \boldsymbol{x}_0, r_1) \right\}. \tag{5}$$

Balsells-Rodas et al. (2024; 2026) studied first- and higher-order overlapping variables, proposing non-parametric conditions to extend the notion of linear independence to sequences of random variables. However, the authors provide concrete instantiations to achieve linear independence only for multivariate Gaussian families. Their key idea is to allow linear dependence only on zero-measure subsets of the overlapping variable space by forcing this space to be sufficiently regular (*e.g.*, real-analytic). We build on their idea in this work, and provide further discussion in App. A.

## 3. Identifiability Theory

We derive conditions on regime-switching SCMs to identify latent regimes $\boldsymbol{R}_{0:T}$ from realisations of $\boldsymbol{X}_{0:T}$. Once the regimes are identified, we can use standard causal discovery to recover regime-dependent initial and window graphs.

### 3.1. Exponential Family Markov Switching Models

To provide sufficient conditions for identifiable regimes, we build on fundamental work on the identifiability of mixtures of exponential families (Barndorff-Nielsen, 1965). Their key idea is to exploit linear independence of the Laplace transform of a continuous minimal regular exponential family. Originally developed for arbitrary, possibly infinite mixtures, their identifiability theory directly applies also to *finite* mixtures. We repurpose these results for regime-switching SCMs. First, we assume exogenous noise along the lines of Barndorff-Nielsen (1965, Cor. 3).

**Assumption 3.1** (Exponential Family Noise). The exogenous noise $\boldsymbol{\epsilon}_t$ is from a continuous minimal regular exponential family (Eq. (2)) that satisfies:

(a1) *Real-analytic sufficient statistic*: The sufficient statistic $\boldsymbol{\tau}$ is real-analytic *a.e.*;
(a2) *Rich image of sufficient statistic*: The image $\{\boldsymbol{\tau}(\boldsymbol{\epsilon}_t) \mid h(\boldsymbol{\epsilon}_t) > 0, \boldsymbol{\epsilon}_t \in \mathcal{X}_\epsilon\}$ contains a (non-empty) open set.

Compared to Barndorff-Nielsen (1965), we strengthen continuity of the sufficient statistic to real-analyticity in condition *(a1)* to avoid linear dependence in the overlapping variable space of the joint distribution $p_{\boldsymbol{\theta}}(\boldsymbol{x}_{0:T})$. The resulting assumption is generally satisfied by well-known continuous minimal regular exponential families, such as the Gaussian, Gamma and Laplace distributions, among others. It excludes certain degenerate and curved exponential families where the image of the sufficient statistic is confined to a lower-dimensional manifold.

Second, we restrict the functions $\boldsymbol{f}_a^0, \boldsymbol{f}_a$ to guarantee linear independence in the distribution families $\mathcal{P}_{\mathcal{A}}^0, \mathcal{P}_{\mathcal{A}}$ and the family of mixture components $\mathcal{P}_{\mathcal{A}}^0 \otimes \mathcal{P}_{\mathcal{A}}^{\otimes T}$.

**Assumption 3.2** (Functional Model Restrictions). The mappings $\boldsymbol{f}_a$ in Eq. (1), and analogously $\boldsymbol{f}_a^0$, satisfy:

(b1) *Unique regimes*: For all $a, a' \in \mathcal{A}$,

$$\boldsymbol{f}_a = \boldsymbol{f}_{a'} \quad a.e. \implies a = a'; \tag{6}$$

(b2) *Pointwise diffeomorphisms*: For almost every $\boldsymbol{x}_{t-1} \in \mathcal{X}, t \in \{1, \ldots, T\}$, and for all $a \in \mathcal{A}$, the mapping $\boldsymbol{f}_a|_{\boldsymbol{X}_{t-1} = \boldsymbol{x}_{t-1}}$ is a diffeomorphism *a.e.* in $\boldsymbol{\epsilon}_t$;
(b3) *Jointly real-analytic transitions*: For all $a \in \mathcal{A}$, the mapping $\boldsymbol{f}_a$ is jointly real-analytic in $(\boldsymbol{x}_{t-1}, \boldsymbol{\epsilon}_t)$ *a.e.*.

Condition *(b1)* ensures there are functional differences between regimes. Condition *(b2)* then guarantees a smooth invertible mapping with smooth inverse between exogenous noise and the initial and transition distribution, such that we can use the change of variables formula to analytically derive the initial and transition PDF, given noise from the exponential family. Condition *(b3)*, in combination with a real-analytic sufficient statistic, forces the natural parameter space of the joint distribution $p_{\boldsymbol{\theta}}(\boldsymbol{x}_{0:T})$ to be sufficiently regular, such that discontinuities arise solely from changes in regimes. Finally, we highlight that the conditions are sufficiently weak to allow parametrisations of $\boldsymbol{f}_a^0, \boldsymbol{f}_a$ using neural networks with piecewise real-analytic activations.

Third, we avoid some rare unidentifiable cases where distinct diffeomorphisms result in identical initial or transition distributions. This includes transformations that exploit certain symmetries, such as rotations of isotropic Gaussians.

**Assumption 3.3.** **At least one** of the following holds:

(c1) *Trivial automorphisms of the noise*: The noise distribution has a trivial automorphism class, *i.e.*, for any invertible mapping $\Phi$,

$$\Phi(\boldsymbol{\epsilon}) \stackrel{d}{=} \boldsymbol{\epsilon} \implies \Phi = \boldsymbol{\epsilon} \quad a.s.; \tag{7}$$

(c2) *Monotone canonicalisation*: There exists a canonical variable order *s.t.* $\forall a \in \mathcal{A}$, $\partial \boldsymbol{f}_a / \partial \boldsymbol{\epsilon}_t$ can be permuted to a lower-triangular matrix, and it has strictly positive diagonal *a.e.* in $\boldsymbol{\epsilon}_t$, and analogously for $\boldsymbol{f}_a^0$.

Either *(c1)* restricts the noise distribution to have a trivial automorphism class, *e.g.*, excluding isotropic Gaussians, or *(c2)* restricts the class of pointwise diffeomorphisms by a monotone canonicalisation of the functions $\boldsymbol{f}_a^0, \boldsymbol{f}_a$, *e.g.*, no rotations. *(c2)* is stronger and guarantees a *unique* monotone triangular transport known as the Knothe-Rosenblatt coupling (Knothe, 1957; Rosenblatt, 1952; Villani, 2009).

Finally, we establish a reparametrisation of $\mathcal{P}_{\mathcal{A}}^0$ and $\mathcal{P}_{\mathcal{A}}$ to an exponential family with common sufficient statistic and base measure through a finite-order polynomial factorisation. This enables known theory from Barndorff-Nielsen (1965).

**Assumption 3.4** (Sufficient Variability in Finite Subspace). The mappings $\Phi_a$ implicit in Eq. (1), and analogously $\Phi_a^0$, satisfy for all $t \in \{1, \ldots, T\}$ and some finite $O < \infty$:

*(d1)* *Common support and integrability:* There exist measurable scaling functions $b_a(\boldsymbol{x}_{t-1}) > 0$ and a common base measure $\widetilde{h}$ such that for all $a \in \mathcal{A}$ and almost every $\boldsymbol{x}_{t-1} \in \mathcal{X}$,

$$h \circ \Phi_a^{-1}(\boldsymbol{x}_t, \boldsymbol{x}_{t-1}) = b_a(\boldsymbol{x}_{t-1})\widetilde{h}(\boldsymbol{x}_t) \quad a.e., \quad (8)$$

where $\int_{\mathcal{X}} \|\boldsymbol{x}_t\|^{O+\delta} \widetilde{h}(\boldsymbol{x}_t) \mathrm{d}\boldsymbol{x}_t < \infty$ for some $\delta > 0$;

*(d2)* *Finite polynomial reparametrisation:* There exists a common sufficient statistic $\widetilde{\boldsymbol{\tau}} : \mathcal{X} \to \mathbb{R}^{\widetilde{P}}$, with $\widetilde{P} \geq P$, whose components are monomials up to order $O$, and measurable matrices $\boldsymbol{C}_a(\boldsymbol{x}_{t-1}) \in \mathbb{R}^{P \times \widetilde{P}}$ with full row rank $P$, such that for all $a \in \mathcal{A}$ and almost every $\boldsymbol{x}_{t-1} \in \mathcal{X}$,

$$\boldsymbol{\tau} \circ \Phi_a^{-1}(\boldsymbol{x}_t, \boldsymbol{x}_{t-1}) = \quad (9)$$
$$\boldsymbol{C}_a(\boldsymbol{x}_{t-1})\widetilde{\boldsymbol{\tau}}(\boldsymbol{x}_t) + \mathcal{R}_O(\boldsymbol{x}_t) \quad a.e.,$$

where the remainder is regime- and history-invariant and can be absorbed into the base measure such that $\int_{\mathcal{X}} \|\mathcal{R}_O(\boldsymbol{x}_t)\|\widetilde{h}(\boldsymbol{x}_t)d\boldsymbol{x}_t < \infty$;

*(d3)* *Injectivity of polynomial coefficients:* For all $a \neq a' \in \mathcal{A}$ and almost every $\boldsymbol{x}_{t-1} \in \mathcal{X}$,

$$\boldsymbol{C}_a(\boldsymbol{x}_{t-1})^T \boldsymbol{\eta} = \boldsymbol{C}_{a'}(\boldsymbol{x}_{t-1})^T \boldsymbol{\eta} \quad \forall \boldsymbol{\eta} \in \mathbb{R}^P \quad (10)$$
$$\implies \quad \Phi_a = \Phi_{a'} \quad a.e..$$

Intuitively, this assumption imposes sufficient variability across regimes in a finite-dimensional polynomial subspace of the sufficient statistic. That is, all regime-discriminating information is captured by a finite amount of linearly independent monomial terms, while any remaining non-polynomial components are invariant across regimes. This can be interpreted as a truncation of a Taylor expansion at finite order $O$, together with the restriction that higher-order terms do not introduce additional regime dependence. In the basic setting of affine transformations of Gaussian noise, the induced sufficient statistic is a quadratic polynomial, so the decomposition holds with finite dimension $\widetilde{P} = P$ and

zero remainder. For nonlinear real-analytic transformations, the regime-invariant remainder enforces the absence of high-order regime signal beyond the low-order polynomial terms.

In short, *(d1)* requires a common support across regimes and ensures integrability of the induced PDF. Condition *(d2)* introduces a common sufficient statistic with monomial terms, possibly higher-dimensional ($\widetilde{P} \geq P$) than the noise distribution. The polynomial decomposition of finite order $O = \binom{D+O}{O}$ stores regime- and history-dependent effects, while a regime-invariant remainder term captures any non-polynomial components of the transformation. Condition *(d3)* recovers regime indices from the distribution in case $\widetilde{P} > P$, while it is vacuously true by *(d2)* if $\widetilde{P} = P$.

We are now ready to establish identifiability up to permutation for regime-switching SCMs. The proof is in App. B.3.

**Theorem 3.5** (Identifiable Regime-Switching SCMs up to Permutation). *Consider an acyclic regime-switching SCM that satisfies conditional causal stationarity, conditional causal sufficiency, and Ass. 3.1 to 3.4. Then the induced MSM is identifiable up to permutation (Def. 2.1).*

### 3.2. Identifiability of Causal Structures

Upon identification of latent regimes, we aim to establish a (one-to-one) correspondence between the data likelihood of the initial and transition distribution and the corresponding initial and window graphs $G_a^0$ and $G_a$, respectively. Here, known causal theory applies. For example, each initial and window graph is identifiable up to a Markov equivalence class (MEC) of conditional independencies under the assumption of *faithfulness* (Spirtes et al., 2000). In an alternative line of research that restricts the function class of $\boldsymbol{f}_a^0, \boldsymbol{f}_a$ (Peters et al., 2011), the initial and transition distributions may correspond to a single initial and window graph, which we refer to as a *uniquely identifiable causal structure*.

The most general class of uniquely identifiable SCMs covered in synthetic experiments is the *location scale noise model* (LSNM) (Immer et al., 2023; Strobl & Lasko, 2023). In the temporal, regime-switching setting, we obtain

$$\boldsymbol{X}_0 \leftarrow \boldsymbol{g}_a^0(\boldsymbol{X}_0) + \Lambda_a^0(\boldsymbol{X}_0)\boldsymbol{\epsilon}_0, \quad (11)$$
$$\boldsymbol{X}_t \leftarrow \boldsymbol{g}_a(\boldsymbol{X}_t, \boldsymbol{X}_{t-1}) + \Lambda_a(\boldsymbol{X}_t, \boldsymbol{X}_{t-1})\boldsymbol{\epsilon}_t, \quad t \geq 1,$$

for measurable functions $\boldsymbol{g}_a^0, \boldsymbol{g}_a$ and diagonal matrices $\Lambda_a^0, \Lambda_a$ with a strictly positive diagonal. The causal structure is identifiable up to a MEC under faithfulness. Conditions for unique bivariate identifiability of the causal structure are readily available for Gaussian noise (Khemakhem et al., 2021) and general noise (Immer et al., 2023; Strobl, 2023), and can be extended to multivariate settings under *causal minimality* (Peters et al., 2014, Rem. 30). The initial and transition distributions are generally non-Gaussian unless the noise is Gaussian and instantaneous effects are affine.

## 4. Estimation Method

We assume access to a dataset $\{\boldsymbol{x}_{0:T}^{(n)}\}_{n=1}^N$ of $N$ *i.i.d.* realisations of the time series $\boldsymbol{X}_{0:T}$. We further assume $\mathcal{A}_K = \{1, \ldots, K\}$, without loss of generality. The goal is to recover latent regimes $\{\boldsymbol{r}_{0:T}^{(n)}\}_{n=1}^N$ and the initial and window graphs $\boldsymbol{G}_{1:K}^0, \boldsymbol{G}_{1:K}$. To do so, we introduce FlowMSM, which is a parametric regime detection framework that extends to regime-dependent causal discovery. While here we explain the implementation for maximum lag $L = 1$, we extend to an arbitrary number of lags in App. C and provide more implementation details. In App. E.5, we demonstrate competitive running times compared to baseline methods.

**Regime Detection.** The goal is to obtain estimates $\widehat{p}_{\boldsymbol{\theta}}(r_t \mid \boldsymbol{x}_{0:T}^{(n)})$ of the regime posterior likelihood. We implement a first-order stationary Markov chain with an initial regime distribution $\boldsymbol{\pi} \in \mathbb{R}^{\widetilde{K}}$ and transition matrix $Q \in \mathbb{R}^{\widetilde{K} \times \widetilde{K}}$, although future work could consider more complex structures. The hyperparameter $\widetilde{K}$ models the number of regimes $K$. It could be chosen either through an elbow method using the data likelihood on a hold-out validation set, or simply as a large value exceeding the oracle value. For computational efficiency, the latter is the recommended strategy in practice. Our real-world experiments exhibit that FlowMSM is able to effectively ignore redundant regimes, and in App. E.5 we show minor loss in performance compared to using $\widetilde{K} = K$.

We leverage the *Generalised Expectation-Maximisation* (GEM) algorithm (Dempster et al., 1977) for mixture model estimation, guaranteed to converge to a local optimum of the likelihood objective. We combine this with a *conditional normalising flow* (CNF) approach to estimate initial and transition distributions. The use of CNFs is motivated by the diffeomorphic mappings in our identifiability theory in Sec. 3. We use a *neural spline flow* (NSF) architecture (Durkan et al., 2019) with parameters $\psi$ shared across regimes. A context window consisting of lagged variables $\boldsymbol{x}_{t-1}$ and a regime-dependent embedding $\mathcal{E}_a$ is fed into a hypernetwork that predicts spline transformation weights for the shared flow. Analogously, the flow modelling the initial distribution uses shared parameters $\psi^0$ and regime-dependent embeddings $\mathcal{E}_a^0$, without additional context. The GEM objective for a single sample is displayed in Fig. 3.

We utilise the regime posterior likelihood for a hard assignment of sequences $\boldsymbol{x}_{t-1:t}^{(n)}$, $t \in \{1, \ldots, T\}$, $n \in \{1, \ldots, N\}$, to the Maximum a posteriori (MAP) regime

$$\widehat{r}_t^{(n)} = \arg\max_{r_t \in \mathcal{A}_{\widetilde{K}}} \widehat{p}_{\boldsymbol{\theta}}(r_t \mid \boldsymbol{x}_{0:T}^{(n)}). \tag{12}$$

This sample splitting scheme creates $\widetilde{K}$ clusters with partially overlapping sliding windows. An analogous procedure is used for the initial samples $\boldsymbol{x}_0^{(n)}$. A perfect regime assignment would lead to clusters of causally stationary windows,

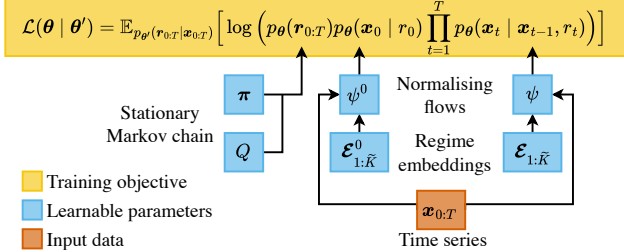

$$\mathcal{L}(\boldsymbol{\theta} \mid \boldsymbol{\theta}') = \mathbb{E}_{p_{\boldsymbol{\theta}'}(\boldsymbol{r}_{0:T} \mid \boldsymbol{x}_{0:T})}\left[\log\left(p_{\boldsymbol{\theta}}(\boldsymbol{r}_{0:T}) p_{\boldsymbol{\theta}}(\boldsymbol{x}_0 \mid r_0) \prod_{t=1}^{T} p_{\boldsymbol{\theta}}(\boldsymbol{x}_t \mid \boldsymbol{x}_{t-1}, r_t)\right)\right]$$

*Figure 3.* FlowMSM parameters $\boldsymbol{\theta} = (\boldsymbol{\pi}, Q, \psi^0, \psi, \mathcal{E}_{1:\widetilde{K}}^0, \mathcal{E}_{1:\widetilde{K}})$ in the Generalised Expectation-Maximisation (GEM) objective.

since the causal parents of variables $\boldsymbol{x}_t^{(n)}$ depend solely on the oracle regime $r_t^{(n)}$, which would allow us to use causal discovery for stationary and causally sufficient settings on each cluster. As we show in Sec. 6, our method empirically recovers the regimes with high accuracy in most settings.

**Causal Discovery.** To estimate initial and window graphs $\widehat{\boldsymbol{G}}_{1:\widetilde{K}}^0$ and $\widehat{\boldsymbol{G}}_{1:\widetilde{K}}$ from the clustered samples, FlowMSM can be paired with any stationary causal discovery method that allows for instantaneous effects. In our experiments, we use several established methods, such as VARLiNGAM (Hyvärinen et al., 2010), DYNOTEARS (Pamfil et al., 2020), PCMCI$^+$ (Runge, 2020) and Rhino (Gong et al., 2023).

## 5. Related Work

**Identifiability in MSMs.** Our work is directly inspired by the work on identifiability theory in Markov Switching Models (MSMs). While early work considered four consecutive discrete variables (An et al., 2013), recent work by Balsells-Rodas et al. (2024; 2025; 2026) provided a more general approach by studying first- and higher-order MSMs, as well as Gaussian MSMs with regimes that depend on both past regimes and observed variables. Similarly to us, these works leverage a fundamental connection between MSMs and finite mixture models (Frühwirth-Schnatter, 2006), for which identifiability theory dates back to the 1960s (Teicher, 1963; Barndorff-Nielsen, 1965; Yakowitz & Spragins, 1968). Our work extends this line of research to exponential families and (possibly nonlinear) instantaneous effects.

**Causal Discovery under Non-Stationarity.** Causal discovery (Spirtes et al., 2000) is a well-established field that focuses on learning causal relations from data. In recent years, causal discovery on time-series data has become particularly popular (Assaad et al., 2022), which has led to the development of several methods to handle non-stationarity. Similarly to us, these methods model non-stationarity as being caused by changes in a latent discrete regime.

A popular approach for causal discovery in non-stationary environnments is CD-NOD (Huang et al., 2020), which detects change-points, *i.e.*, changes in local causal mechanisms

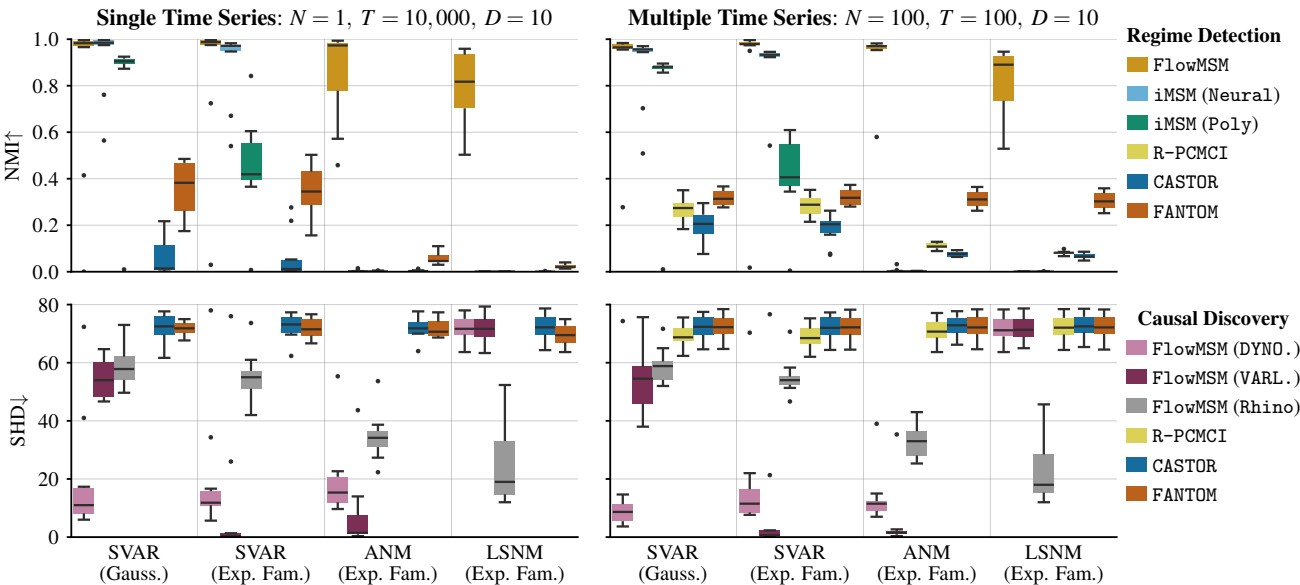

*Figure 4.* Performance on (*top*) regime detection and (*bottom*) causal discovery on (*left*) a single long time series and (*right*) multiple smaller time series, with $K = 3$ regimes and causal parents of at most a single lag. Regimes persist for 10 time steps on average and the edge density of the synthetic causal structures is close to 0.5. Each boxplot covers 10 random seeds.

driven by a surrogate variable, such as the time index, as well as a summary graph that represents the union of all causal edges in all regimes. In our work, we instead focus on identifying the regimes, as well as learning the individual causal graph for each regime. Other methods focus on this task as well. In particular, Saggioro et al. (2020) and Gao et al. (2023) generalise the popular non-parametric temporal causal discovery method `PCMCI` (Runge et al., 2019) to latent regimes and periodic non-stationarity, respectively, yet neither accommodates instantaneous effects. On the other hand, `PCMCI+` (Runge, 2020), another extension of `PCMCI`, allows for instantaneous effects, but cannot model non-stationarity. Mameche et al. (2025a) simultaneously considers context- and regime-switches in causal mechanisms, but assumes causal parents to be invariant across regimes, which can be a restrictive assumption.

Closest to our setting, Balsells-Rodas et al. (2024) introduces `iMSM`, detecting latent regimes under Gaussian transitions with autoregressive means parametrised by real-analytic neural networks, but without instantaneous effects nor causal discovery, and with a more restrictive parametric assumption. Two recent methods do focus on instantaneous effects and causal discovery: `CASTOR` (Rahmani & Frossard, 2025a) estimates regime-switching Gaussian models using the EM algorithm, while `FANTOM` (Rahmani & Frossard, 2025b) generalises this approach to non-Gaussian settings using Bayesian EM and a variational ELBO objective. As opposed to us, these two works do not allow dependencies between the regimes, which can arise in many real-world settings, and iterate between regime detection and causal discovery within the class of *location scale noise models*

(LSNMs) (Immer et al., 2023; Strobl & Lasko, 2023). We instead decouple these two phases, which allows us to also consider a broader class of causal models that might not be uniquely identifiable.

**Causal Mixtures.** Learning mixtures of causal models is a popular task in the literature (Thiesson et al., 1998; Saeed et al., 2020; Günther et al., 2023; Strobl, 2023; Mameche et al., 2025b; Rabel & Runge, 2025). In particular, Varambally et al. (2024) models a mixture of causally stationary time series generated by distinct causal models, producing mixture assignments and regime-dependent window graphs. We focus on a more general setting of non-stationarity, of which mixtures are a special case, as discussed in Sec. 2.1.

## 6. Experiments

Our experiments are reproducible using the codebase at https://github.com/roelhulsman/flowmsm.

**Methods.** For regime detection, we benchmark against Gaussian MSMs, denoted `iMSM (Neural)` and `iMSM (Poly)` (Balsells-Rodas et al., 2024), where autoregressive Gaussian means are modelled with MLPs and cubic polynomials, respectively. For both regime detection and causal discovery, we additionally compare against `R-PCMCI` (Saggioro et al., 2020), `CASTOR` (Rahmani & Frossard, 2025a) and `FANTOM` (Rahmani & Frossard, 2025b). For causal discovery, we refer to our method combined with `DYNOTEARS` (Pamfil et al., 2020) as `FlowMSM (DYNO.)`, combined with `VARLiNGAM` (Hyvärinen et al., 2010) as `FlowMSM (VARL.)`, and combined with Rhino (Gong et al., 2023)

*Table 1.* Evaluation on $N = 100$ synthetic time series of length $T = 100$ and dimension $D = 10$ with $K = 3$ regimes, averaged over 10 random seeds. Our framework and the best scores are in **bold**, and second-best scores are underlined.

| | SVAR (Gauss.) | | | | SVAR (Exp. Fam.) | | | | ANM (Exp. Fam.) | | | | LSNM (Exp. Fam.) | | | |
|---|---|---|---|---|---|---|---|---|---|---|---|---|---|---|---|---|
| **Regime Detection** | NMI↑ | ARI↑ | F1↑ | Acc.↑ | NMI↑ | ARI↑ | F1↑ | Acc.↑ | NMI↑ | ARI↑ | F1↑ | Acc.↑ | NMI↑ | ARI↑ | F1↑ | Acc.↑ |
| **FlowMSM** | **0.90** | **0.91** | **0.95** | **0.95** | 0.88 | 0.89 | 0.93 | 0.94 | **0.93** | **0.94** | **0.97** | **0.97** | **0.83** | **0.85** | **0.89** | **0.91** |
| iMSM(Neural) | 0.89 | 0.90 | 0.93 | 0.94 | **0.90** | **0.91** | **0.95** | **0.95** | 0.01 | 0.01 | 0.37 | 0.38 | 0.00 | 0.00 | 0.35 | 0.36 |
| iMSM(Poly) | 0.79 | 0.84 | 0.90 | 0.91 | 0.42 | 0.41 | 0.67 | 0.68 | 0.00 | 0.00 | 0.36 | 0.37 | 0.00 | 0.00 | 0.35 | 0.36 |
| R-PCMCI | 0.27 | 0.25 | 0.58 | 0.63 | 0.28 | 0.27 | 0.59 | 0.64 | 0.11 | 0.08 | 0.50 | 0.50 | 0.08 | 0.04 | 0.47 | 0.48 |
| CASTOR | 0.20 | 0.17 | 0.52 | 0.60 | 0.19 | 0.16 | 0.51 | 0.59 | 0.08 | 0.03 | 0.45 | 0.47 | 0.07 | 0.01 | 0.42 | 0.48 |
| FANTOM | 0.32 | 0.13 | 0.40 | 0.27 | 0.32 | 0.13 | 0.41 | 0.28 | 0.31 | 0.12 | 0.40 | 0.27 | 0.31 | 0.12 | 0.40 | 0.27 |
| **Causal Discovery** | SHD↓ | F1↑ | Acc.↑ | | SHD↓ | F1↑ | Acc.↑ | | SHD↓ | F1↑ | Acc.↑ | | SHD↓ | F1↑ | Acc.↑ | |
| DYNOTEARS | 75.50 | 0.59 | 0.59 | | 76.63 | 0.59 | 0.58 | | 66.57 | 0.64 | 0.64 | | 72.30 | 0.50 | 0.64 | |
| DYNOTEARS* | **12.37** | **0.93** | **0.93** | | 14.20 | 0.92 | 0.92 | | 10.47 | 0.92 | 0.92 | | 71.33 | 0.51 | 0.64 | |
| **FlowMSM**(DYNO.) | 14.83 | 0.91 | 0.91 | | 18.07 | 0.90 | 0.90 | | 13.53 | 0.91 | 0.91 | | 71.57 | 0.52 | 0.64 | |
| VARLiNGAM | 78.80 | 0.57 | 0.56 | | 78.17 | 0.58 | 0.57 | | 65.93 | 0.66 | 0.65 | | 72.30 | 0.50 | 0.64 | |
| VARLiNGAM* | 53.07 | 0.69 | 0.68 | | **0.07** | **1.00** | **1.00** | | **1.77** | **0.99** | **0.99** | | 71.37 | 0.51 | 0.64 | |
| **FlowMSM**(VARL.) | 53.70 | 0.68 | 0.68 | | 10.43 | 0.94 | 0.94 | | 4.80 | 0.98 | 0.98 | | 71.77 | 0.51 | 0.64 | |
| Rhino | 76.03 | 0.59 | 0.58 | | 74.37 | 0.60 | 0.59 | | 64.10 | 0.66 | 0.65 | | 58.23 | 0.70 | 0.69 | |
| Rhino* | 56.37 | 0.68 | 0.68 | | 54.37 | 0.71 | 0.71 | | 31.37 | 0.83 | 0.83 | | **14.87** | **0.92** | **0.92** | |
| **FlowMSM**(Rhino) | 58.77 | 0.67 | 0.67 | | 55.10 | 0.71 | 0.70 | | 32.70 | 0.82 | 0.82 | | 23.40 | 0.87 | 0.88 | |
| R-PCMCI | 69.34 | 0.55 | 0.65 | | 68.93 | 0.56 | 0.66 | | 70.98 | 0.53 | 0.65 | | 72.16 | 0.51 | 0.64 | |
| CASTOR | 72.17 | 0.53 | 0.64 | | 72.09 | 0.53 | 0.64 | | 72.72 | 0.53 | 0.63 | | 72.70 | 0.51 | 0.63 | |
| FANTOM | 72.27 | 0.51 | 0.64 | | 72.22 | 0.51 | 0.64 | | 72.29 | 0.51 | 0.64 | | 72.30 | 0.51 | 0.64 | |

*Methods aided with oracle regime information.

as FlowMSM (Rhino). In App. E.3, we further evaluate MCD (Varambally et al., 2024) on non-temporal synthetic mixtures of causal models. In App. E.4, we assess the combination of our method with PCMCI$^+$, which we call FlowMSM (PCMCI$^+$), CD-NOD (Huang et al., 2020) and SpaceTime (Mameche et al., 2025a) on *CPDAGs* and *summary window graphs*, respectively.

**Metrics.** To evaluate the estimated regimes $\{\widehat{r}_{0:T}^{(n)}\}_{n=1}^{N}$, we report normalised mutual information (NMI↑) and the adjusted Rand index (ARI↑). In addition, we compute accuracy↑ and F1↑ after a label permutation using the Hungarian algorithm (Kuhn, 1955). For the estimated window graphs $\widehat{G}_{1:\widetilde{K}}$, we report Structural Hamming Distance (SHD↓) (Tsamardinos et al., 2006), accuracy↑ and F1↑ of the estimated edges, averaged over the $K$ regimes. For real-world data, we do not have access to the ground truth causal graph nor regimes, so we provide a qualitative study.

**Synthetic Dataset.** The synthetic data-generating process for the linear SVAR, nonlinear ANM and the LSNM class are detailed in App. D.1. We consider one "easy" Gaussian and three "challenging" exponential family settings with skewed Gamma noise. In the latter, the initial and transition distributions belong to a complex unknown exponential family, consistent with our identifiability theory. The LSNM class represents our "most challenging" setting, where we construct initial and transition distributions whose mean is invariant across regimes, such that regime changes are expressed solely through heterogenous distribution scale.

**Results.** Fig. 4 summarises performance on regime detection and causal discovery. R-PCMCI is omitted from the left column due to memory constraints. A complete comparison of evaluation metrics is provided in Tab. 1, with further discussion in App. E.1. In simple linear settings, our approach achieves performance comparable to Gaussian iMSM baselines, as expected. In contrast, in nonlinear and non-Gaussian settings, FlowMSM maintains strong performance, whereas baselines degrade substantially. This pattern is consistent across both a single long time series and a collection of shorter sequences. We empirically observe that FANTOM rarely deviates from its initial window assignments in the shorter sequences, resulting in nearly identical NMI scores across different SCMs. The moderate to high F1 and accuracy scores are likely artificially inflated by the label permutation, while the agreement between the true and estimated partition is close to random chance.

For causal discovery, our approach shows strong performance contingent on the chosen discovery method. DYNOTEARS performs well when regime changes primarily affect distribution location, while VARLiNGAM excels in non-Gaussian settings, except the LSNM class, for which Rhino performs the best. None of the baselines achieve competitive performance in these settings, possibly due to frequent regime switches that might be challenging for iterative algorithms in terms of initialization. Moreover, the baselines infer fixed regime indices for a time series, necessitating independent deployment on each of $N$ sequences and limiting information pooling in short sequences.

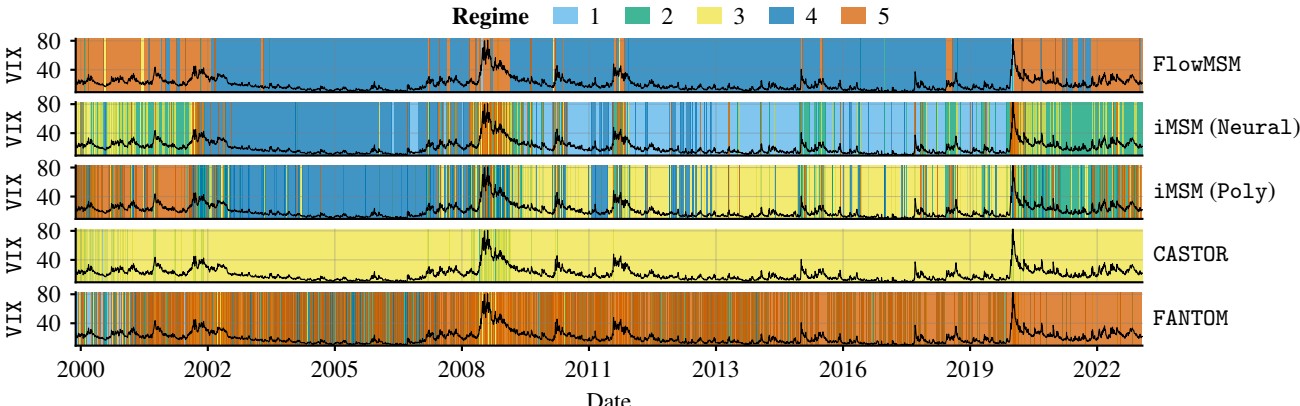

*Figure 5.* Estimated regimes on daily data from the Fama-French model and `AAPL` stock, with an overlay of the `VIX` volatility index, which is not included in training. We use $\widehat{K} = 5$ for `FlowMSM` and `iMSM` and initial windows of size $1,000$ for `CASTOR` and `FANTOM`.

**Ablations.** In App. E.1, we study the impact of oracle regime assignments on the estimation of window graphs, thereby isolating the effect of regime uncertainty on causal discovery. In App. E.2, we examine various non-Gaussian exogenous noise distributions. In App. E.5, we analyse sensitivity to data-generation and model hyperparameters, and we provide computational runtimes.

**Fama-French Five-Factor Model.** We explore regime detection and causal discovery in asset pricing to qualitatively evaluate the Fama-French five-factor asset-pricing model (Fama & French, 2015). The model posits five risk factors that capture systemic patterns in stock returns: size, value, market risk, profitability and investment. Following Sadeghi et al. (2024), we also add the excess returns of Apple's stock (`AAPL`) and analyse data spanning the early 2000s to the end of 2022, for a total of 5,786 trading days.

Fig. 5 shows regimes inferred by `FlowMSM` and baselines, where `R-PCMCI` is omitted due to memory constraints. `FlowMSM` identifies two dominant regimes aligned with periods of low and high market volatility, the latter including the 2008 financial crisis, the COVID-19 pandemic, and the early-2000s dot-com bubble. In contrast, other methods infer regimes and regime switches that are not easily mapped to market events.

Even though the Fama-French model is presented as a *statistical* rather than a *causal* model, we investigate its causal implications and limitations in App. E.6. For example, under the efficient market hypothesis, causal dependencies across time steps should be absent, although we expect periods of market distress may give rise to arbitrage opportunities. We only find partial support for these (and other) hypotheses, suggesting causal interpretations of the Fama-French model on this dataset are limited in scope. Furthermore, on *monthly* instead of *daily* data, we fail to identify meaningful interpretations for any method. In this case, we posit the available months are too few to derive substantiated conclusions.

## 7. Conclusion

We explore regime detection and causal discovery in non-stationary time series in the presence of regime switches and nonlinear and non-Gaussian dynamics. We propose novel theory on identifiability of latent regimes under temporal regime dependencies, nonlinear lagged and instantaneous effects, and independent noise from the exponential family. Our regime detection framework, `FlowMSM`, readily extends to post-hoc regime-dependent causal discovery.

Our work provides a principled framework for regime identification in non-stationary time series, and we believe most of our assumptions to be reasonably mild restrictions. Specifically, Ass. 3.4 is only violated when regimes are indistinguishable in any finite polynomial subspace of the sufficient statistic, which we believe to be largely confined to theoretical scenarios. Similarly, Ass. 3.3 filters out boundary anomalies. Finally, conditional causal stationarity can be motivated to (approximately) hold in many real-world settings, including but not limited to seasonal patterns, physiological and psychological states, and network congestion.

Conversely, some of our other assumptions might be unrealistic in real-world settings. For example, our identifiability theory requires that the regime variable $R_t$ is unaffected by the observations $\boldsymbol{X}_{0:T}$. Balsells-Rodas et al. (2025) relaxes this assumption in Gaussian settings, but extending to exponential families or beyond is nontrivial. Moreover, our identifiability theory critically relies on exponential family noise and real-analytic diffeomorphisms in Ass. 3.1 and 3.2. Preserving linear independence in more relaxed settings is a challenging task that might warrant a vastly different proof strategy than the one deployed in this work. Finally, future work could explore non-stationary causal models with latent confounders other than regime variables, thus further relaxing (conditional) causal sufficiency.

## Acknowledgements

We express gratitude to Yingzhen Li, Mátyás Schubert, Alex Egg, Richard Price, Tobias Witte and anonymous reviewers for helpful comments. RH was supported by Adyen, a global financial technology platform. SM would like to acknowledge the VIDI grant CANES (VI.Vidi.243.247). We thank professor Kenneth R. French for public availability of the research-factor data in his online Data Library. To conduct experiments, we used the Dutch national supercomputer Snellius with the support of the SURF Cooperative.

## Impact Statement

This paper presents work whose goal is to advance the field of Machine Learning. There are many potential societal consequences of our work, none which we feel must be specifically highlighted here.

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

# A. Background Identifiability Theory

## A.1. Finite Mixture Distributions

We start with well-established results on identifiable finite mixture distributions. Consider a family of continuous CDFs

$$\mathcal{F}_{\mathcal{A}} \triangleq \big\{ F(\boldsymbol{x} \mid a) \mid a \in \mathcal{A}, \boldsymbol{x} \in \mathcal{X} \big\}, \tag{13}$$

where $\mathcal{A}$ and $\mathcal{X} \subseteq \mathbb{R}^D$ are as in the main text such that the functions $F$ are measurable on $\mathcal{X} \times \mathcal{A}$.

We construct a family of finite mixture distributions by linearly combining distributions from the family of CDFs, *i.e.*,

$$\mathcal{H}_{\mathcal{A}} \triangleq \left\{ H(\boldsymbol{x}) = \sum_{a \in \mathcal{A}_K} \lambda_a F(\boldsymbol{x} \mid a) \;\middle|\; \lambda_a > 0 \, \forall a \in \mathcal{A}_K, \sum_{a \in \mathcal{A}_K} \lambda_a = 1, \mathcal{A}_K \subset \mathcal{A}, K < \infty, \boldsymbol{x} \in \mathcal{X} \right\}. \tag{14}$$

The finite mixture distributions in $\mathcal{H}$ are said to be identifiable up to permutation when the following definition is satisfied.

**Definition A.1.** The finite mixture distributions in the family $\mathcal{H}_{\mathcal{A}}$ are said to be *identifiable up to permutation* if for any two finite mixtures $H_1(\boldsymbol{x}) = \sum_{a \in \mathcal{A}_K} \lambda_a F(\boldsymbol{x} \mid a)$ and $H_2(\boldsymbol{x}) = \sum_{a' \in \mathcal{A}'_{K'}} \lambda'_{a'} F(\boldsymbol{x} \mid a')$ in $\mathcal{H}_{\mathcal{A}}$, we have $H_1(\boldsymbol{x}) = H_2(\boldsymbol{x})$ *a.e.* implies $K = K'$ and there exists some permutation $\phi : \mathcal{A}_K \to \mathcal{A}'_{K'}$ such that for all $a \in \mathcal{A}_K$ we have $\lambda_a = \lambda'_{\phi(a)}$ and $F(\boldsymbol{x} \mid a) = F(\boldsymbol{x} \mid \phi(a))$ *a.e.*.

Yakowitz & Spragins (1968) introduced a necessary and sufficient condition for identifiable finite mixture distribution families. Before restating their result, we include the definition of linearly independent functions under finite mixtures. The addition of *under finite mixtures* reflects that linear dependency is allowed for linear combinations of *infinitely many* other functions in the family. Throughout this work, we tend to write linearly independent functions without specifying the weakened requirement of *under finite mixtures*, but this is meant as shorthand.

**Definition A.2.** The family $\mathcal{F}_{\mathcal{A}}$ is said to contain *linearly independent functions under finite mixtures* on $\mathcal{X}$ if for any finite subset $\mathcal{A}_K \subset \mathcal{A}$ with $K < \infty$, the functions in $\{ F(\boldsymbol{x} \mid a) \mid a \in \mathcal{A}_K, \boldsymbol{x} \in \mathcal{X} \} \subset \mathcal{F}_{\mathcal{A}}$ are linearly independent *a.e.*, *i.e.*,

$$\sum_{a \in \mathcal{A}_K} \lambda_a F(\boldsymbol{x} \mid a) = 0 \quad a.e. \quad \Longrightarrow \quad \lambda_a = 0 \quad \text{for all } a \in \mathcal{A}_K. \tag{15}$$

**Lemma A.3** (Thm. 1 in Yakowitz & Spragins (1968)). *The finite mixture distributions in the family $\mathcal{H}_{\mathcal{A}}$ are identifiable up to permutation if and only if the functions in $\mathcal{F}_{\mathcal{A}}$ are linearly independent under finite mixtures.*

## A.2. Mixtures of Exponential Families

A few years prior, Barndorff-Nielsen (1965) derived conditions under which exponential families generate identifiable mixtures. Let

$$\overline{\mathcal{F}} \triangleq \left\{ \overline{F}_{\boldsymbol{\eta}}(\boldsymbol{x}) \;\middle|\; \boldsymbol{\eta} \in \Omega, \boldsymbol{x} \in \mathcal{X} \right\} \tag{16}$$

be a family of CDFs corresponding to an exponential family of Borel probability measures with natural parameters $\boldsymbol{\eta}$ and sufficient statistic $\boldsymbol{\tau}$. Now consider the family of mixture distributions

$$\overline{\mathcal{H}} \triangleq \left\{ \overline{H}(\boldsymbol{x}) = \int_{\Omega} \overline{F}_{\boldsymbol{\eta}}(\boldsymbol{x}) \mathrm{d}\gamma(\boldsymbol{\eta}) \right\}, \tag{17}$$

where $\gamma$ is a probability measure on $\Omega$. *Finite* mixtures are a special case where $\gamma$ has finite support. We highlight that this is a mixture over natural parameters $\boldsymbol{\eta}$, keeping the sufficient statistic $\boldsymbol{\tau}$ and support $\mathcal{X}$ fixed across regimes.

Identifiability of the finite mixture distributions in $\overline{\mathcal{H}}$ here means that for any two mixtures $\overline{H}_1 = \int_{\Omega} \overline{F}_{\boldsymbol{\eta}}(\boldsymbol{x}) \mathrm{d}\gamma_1(\boldsymbol{\eta})$ and $\overline{H}_2 = \int_{\Omega} \overline{F}_{\boldsymbol{\eta}}(\boldsymbol{x}) \mathrm{d}\gamma_2(\boldsymbol{\eta})$, we have

$$\int_{\Omega} \overline{F}_{\boldsymbol{\eta}}(\boldsymbol{x}) \mathrm{d}\gamma_1(\boldsymbol{\eta}) = \int_{\Omega} \overline{F}_{\boldsymbol{\eta}}(\boldsymbol{x}) \mathrm{d}\gamma_2(\boldsymbol{\eta}) \quad a.e. \quad \Longrightarrow \quad \gamma_1 = \gamma_2. \tag{18}$$

Below, we restate one particular set of conditions (Barndorff-Nielsen, 1965, Cor. 3), adapting notation to our setting. Although the statement targets arbitrary mixtures in the original work, it holds for finite mixtures in particular.

**Lemma A.4** (Cor. 3 in Barndorff-Nielsen (1965)). *Consider the family of continuous CDFs $\overline{\mathcal{F}}$ and assume the following:*

*($\overline{a1}$) The probability measures corresponding to $\overline{F}_{\boldsymbol{\eta}}$ are absolutely continuous with respect to the Lebesgue measure;*

*($\overline{a2}$) The sufficient statistic $\boldsymbol{\tau}$ is continuous;*

*($\overline{a3}$) The set $\{\boldsymbol{\tau}(\boldsymbol{x}) \mid h(\boldsymbol{x}) > 0, \boldsymbol{x} \in \mathcal{X}\}$ contains a (non-empty) open set.*

*Then the mixture distributions in the family $\overline{\mathcal{H}}$ are identifiable.*

### A.3. Markov Switching Models

MSMs are studied in Balsells-Rodas et al. (2024; 2026) and we restate some of their main results for future reference. Throughout, we adapt notation to match the main text, and for proofs we refer to the original work.

The attentive reader will notice that the results on finite mixture distributions in the previous section focus on mixtures of CDFs, while MSMs concern mixtures of PDFs. However, under the assumption below, the results in the previous section transfer directly to families of PDFs, without loss of generality. We proceed to discuss families of PDFs.

**Assumption A.5** (Existing Densities, p.13 in Balsells-Rodas et al. (2024)). The probability measures of $\boldsymbol{X}_{0:T} \mid \boldsymbol{R}_{0:T}$ are dominated by the Lebesgue measure on $\mathcal{X}^{\times(T+1)}$ and the CDFs are continuously differentiable.

Consider the following countably infinite family of continuous PDFs, which we often refer to as the *product family* of initial and transition distribution families, *i.e.*,

$$\mathcal{P}_{\mathcal{A}}^0 \otimes \mathcal{P}_{\mathcal{A}}^{\otimes T} \triangleq \left\{ p_{\boldsymbol{\theta}}(\boldsymbol{x}_{0:T} \mid \boldsymbol{r}_{0:T}) \mid \boldsymbol{r}_{0:T} \in \mathcal{A}^{\times(T+1)}, \boldsymbol{x}_{0:T} \in \mathcal{X}^{\times(T+1)} \right\}. \tag{19}$$

An MSM is a finite mixture distribution with $K^{T+1}$ components taken from $\mathcal{P}_{\mathcal{A}}^0 \otimes \mathcal{P}_{\mathcal{A}}^{\otimes T}$, and the family of MSMs $\mathcal{M}_{\mathcal{A}}^T$ defined below consists of all such finite mixtures. We simplify notation compared to Balsells-Rodas et al. (2024) by using the regime $r_t$ as index directly, to omit an injective index mapping. This gives

$$\mathcal{M}_{\mathcal{A}}^T \triangleq \left\{ p_{\boldsymbol{\theta}}(\boldsymbol{x}_{0:T}) = \sum_{\boldsymbol{r}_{0:T}} p_{\boldsymbol{\theta}}(\boldsymbol{r}_{0:T}) p_{\boldsymbol{\theta}}(\boldsymbol{x}_0 \mid r_0) \prod_{t=1}^{T} p_{\boldsymbol{\theta}}(\boldsymbol{x}_t \mid \boldsymbol{x}_{t-1}, r_t) \;\middle|\; p_{\boldsymbol{\theta}}(\boldsymbol{x}_0 \mid r_0) \in \mathcal{P}_{\mathcal{A}}^0, \; p_{\boldsymbol{\theta}}(\boldsymbol{x}_t \mid \boldsymbol{x}_{t-1}, r_t) \in \mathcal{P}_{\mathcal{A}}, \right. \tag{20}$$
$$\left. r_0, r_t \in \mathcal{A}_K \subset \mathcal{A}, \; t \in \{1, \dots, T\}, K < \infty \right\}.$$

Given the specific structure of MSMs in terms of conditional causal stationarity and the factorisation in Eq. (4), the definition of identifiability up to permutation in the main text (Def. 2.1) follows from identifiability up to permutation of a finite mixture distribution family (Def. A.1). Similarly, the following is a direct consequence of Lem. A.3 for MSMs.

**Lemma A.6** (Thm. B.2 in Balsells-Rodas et al. (2024)). *Given Ass. A.5, if the PDFs in $\mathcal{P}_{\mathcal{A}}^0 \otimes \mathcal{P}_{\mathcal{A}}^{\otimes T}$ are linearly independent under finite mixtures (Def. A.2), then the mixture distributions in the family of MSMs $\mathcal{M}_{\mathcal{A}}^T$ are identifiable up to permutation.*

In order to show identifiability of the mixture distributions in the family of MSMs, it is thus sufficient to show the component family $\mathcal{P}_{\mathcal{A}}^0 \otimes \mathcal{P}_{\mathcal{A}}^{\otimes T}$ contains linearly independent functions. The following lemma summarises non-parametric sufficient conditions ensuring this property. For clarity, we combine Balsells-Rodas et al. (2024, Lem. B.3 and Thm. B.4) and adapt notation to our setting. This means we can drop conditions *(b1)-(b2)* in their Lem. B.3, as they are vacuously true when the functions are PDFs that satisfy their conditions *(b3)-(b4)*. That is, their assumption *(b1)* follows directly from positivity of a PDF on its support, and their assumption *(b2)* is implied by their linear independence conditions *(b3)–(b4)*, as shown in Balsells-Rodas et al. (2026).

**Lemma A.7** (Adaptation of Lem. B.3 and Thm. B.4 in Balsells-Rodas et al. (2024)). *Given Ass. A.5, consider the family of MSMs $\mathcal{M}_{\mathcal{A}}^T$ and assume the following hold for the initial and transition families $\mathcal{P}_{\mathcal{A}}^0, \mathcal{P}_{\mathcal{A}}$, respectively:*

*($\overline{b1}$)* Linear independence under finite mixtures on specific non-zero measure subsets in $\mathcal{P}_{\mathcal{A}}^0$: *For any non-zero measure subset $\mathcal{Y} \subset \mathcal{X}$, the initial distribution family $\mathcal{P}_{\mathcal{A}}^0$ contains linearly independent functions under finite mixtures on $\boldsymbol{x}_0 \in \mathcal{Y}$;*

($\overline{b2}$) Linear independence under finite mixtures on specific non-zero measure subsets in $\mathcal{P}_\mathcal{A}$: *There exists a non-zero measure subset $\mathcal{Y} \subset \mathcal{X}$ such that for any non-zero measure subsets $\mathcal{Y}' \subset \mathcal{Y}$ and $\mathcal{Z} \subset \mathcal{X}$, the transition distribution family $\mathcal{P}_\mathcal{A}$ contains linearly independent functions under finite mixtures on $(\boldsymbol{x}_t, \boldsymbol{x}_{t-1}) \in \mathcal{Z} \times \mathcal{Y}'$;*

($\overline{b3}$) Linear dependence under finite mixtures for subsets of functions in $\mathcal{P}_\mathcal{A}$ implies repeating functions*: For any $\boldsymbol{\beta} \in \mathcal{X}$, any non-zero measure subset $\mathcal{Z} \subset \mathcal{X}$ and any subset $\mathcal{A}_K \subset \mathcal{A}$ such that $K < \infty$, the family $\{p_{\boldsymbol{\theta}}(\boldsymbol{x}_t \mid \boldsymbol{x}_{t-1} = \boldsymbol{\beta}, a) \mid a \in \mathcal{A}_K\}$ contains linearly dependent functions on $\boldsymbol{x}_t \in \mathcal{Z}$ only if there exists $a \neq a' \in \mathcal{A}_K$ such that $p_{\boldsymbol{\theta}}(\boldsymbol{x}_t \mid \boldsymbol{x}_{t-1} = \boldsymbol{\beta}, a) = p_{\boldsymbol{\theta}}(\boldsymbol{x}_t \mid \boldsymbol{x}_{t-1} = \boldsymbol{\beta}, a')$ for all $\boldsymbol{x}_t \in \mathcal{X}$;*

($\overline{b4}$) Continuity for $\mathcal{P}_\mathcal{A}$*: For any $a \in \mathcal{A}$, $p_{\boldsymbol{\theta}}(\boldsymbol{x}_t \mid \boldsymbol{x}_{t-1}, a)$ is continuous in $\boldsymbol{x}_{t-1} \in \mathcal{X}$.*

*Then for any $T \geq 1$ and any subset $\mathcal{Z} \subset \mathcal{X}$, the joint distribution family $\mathcal{P}_\mathcal{A}^0 \otimes \mathcal{P}_\mathcal{A}^{\otimes T}$ contains linearly independent functions under finite mixtures on $(\boldsymbol{x}_{0:T-1}, \boldsymbol{x}_T) \in \mathcal{X}^T \times \mathcal{Z}$ (Def. A.2).*

Linear independence can be an abstract notion, and it can be unclear when it is satisfied in real-world settings. To illustrate, Balsells-Rodas et al. (2024) only provides an instantiation of Lem. A.7 for Gaussian initial and transition families with transition distributions parametrised by real-analytic functions. Our contributions generalise these results to exponential families, and translate linear independence to more fine-grained assumptions on regime-switching SCMs.

# B. Proofs for Identifiability Theory

In App. B.1, we elaborate on how our setting with dependencies between regimes relates to independently sampled regimes at each time step, *i.e.*, the setting considered in Rahmani & Frossard (2025a;b). We restate a minor adaptation of their main theoretical result, and subsequently show that linear independence of the transition distribution family is sufficient to identify the full regime trajectory, when regimes are independent.

In App. B.2, we generalise established results on identifiable Gaussian MSMs with dependencies between regimes to other continuous minimal regular exponential family distributions. We leverage fundamental work from Barndorff-Nielsen (1965), set out in Lem. A.4. Furthermore, we build on the nonparametric conditions for linear independence of sequences of random variables proposed in Balsells-Rodas et al. (2024) and summarised in Lem. A.7.

In App. B.3, we translate the conditions for linear independence of the initial and transition distribution family to more fine-grained conditions on regime-switching SCMs, and provide a proof of our main theoretical result, Thm. 3.5.

In App. B.4, we relax the mathematically convenient assumption of *first-order* MSMs to an arbitrary number of lags, following work from Balsells-Rodas et al. (2026).

## B.1. Identifiable Markov Switching Models under Independent Regimes

Below, we adapt Rahmani & Frossard (2025b, Thm. 4.3) to our setting and notation. We omit the condition that the causal structure is identifiable, since it is not needed to show identifiability of the induced finite mixture distributions. This does not alter the strength of the claim, since identifiable causal structure can simply be added to both the assumptions and the conclusion to retain the original statement.

**Lemma B.1** (Adaptation of Thm. 4.3 in Rahmani & Frossard (2025b))**.** *Consider the family of transition distributions $\mathcal{P}_\mathcal{A}$ (Eq. (4)). Assume that $\mathcal{P}_\mathcal{A}$ contains linearly independent functions and the regimes $R_t$ are independent. Then the finite mixture distributions induced by $\mathcal{P}_\mathcal{A}$, i.e.,*

$$\left\{ p_{\boldsymbol{\theta}}(\boldsymbol{x}_t \mid \boldsymbol{x}_{t-1}) = \sum_{r_t \in \mathcal{A}_K} p_{\boldsymbol{\theta}}(r_t) p_{\boldsymbol{\theta}}(\boldsymbol{x}_t \mid \boldsymbol{x}_{t-1}, r_t) \;\Big|\; \mathcal{A}_K \subset \mathcal{A}, K < \infty, \boldsymbol{x}_t, \boldsymbol{x}_{t-1} \in \mathcal{X}, t \in \{1, \ldots, T\} \right\}, \quad (21)$$

*are identifiable up to permutation (Def. A.1).*

*Proof:* The result follows from Yakowitz & Spragins (1968, Thm. 1), using that $p_{\boldsymbol{\theta}}(r_t)$ does not depend on $\boldsymbol{r}_{0:t-1}$. ∎

We call this a *local* form of identifiable finite mixtures, since it considers finite mixture distributions at each transition time step. We proceed to show that under the assumption that regimes are independent, local identifiability implies a stronger result on identifiable *trajectories* over the full time-sequence. That is, we can identify the full regime trajectory $\boldsymbol{R}_{0:T}$ from realizations of $\boldsymbol{X}_{0:T}$, since the trajectory cannot introduce linear dependence when regimes are independent.

The proposition below is closely related to the less restrictive setting in this work and Balsells-Rodas et al. (2024; 2026) with dependencies between regimes, and results on Gaussian MSMs where regimes may depend on both past regimes and observed variables (Balsells-Rodas et al., 2025, Thm. A.6). However, the proposition below may be of independent interest, since it considers a more restrictive setting and correspondingly weakened conditions.

**Proposition B.2** (Identifiable Regime Trajectories). *Consider the families $\mathcal{P}_{\mathcal{A}}^0$, $\mathcal{P}_{\mathcal{A}}$ of initial and transition distributions (Eq. (4)) corresponding to a regime-switching SCM (Eq. (1)). Assume that either the initial or the transition distribution family contains linearly independent functions, and the regimes $R_t$ are independent. Then the MSMs in $\mathcal{M}_{\mathcal{A}}^T$ (Eq. (20)), induced by the product family $\mathcal{P}_{\mathcal{A}}^0 \otimes \mathcal{P}_{\mathcal{A}}^{\otimes T}$ (Eq. (19)), are identifiable up to permutation (Def. 2.1).*

*Proof*: Recall the factorization in Eqs. (3) and (4), where without loss of generality, we assumed temporal dependencies of at most one lag. Suppose now that the latent regime prior factorizes as $p_{\boldsymbol{\theta}}(\boldsymbol{r}_{0:T}) = \prod_{t=0}^{T} p_{\boldsymbol{\theta}}(r_t)$. That is, the latent regime trajectory consists of independent regimes, which may be a function of time, such as in Rahmani & Frossard (2025b, p. 5).

Following independent regimes, we can factorize Eqs. (3) and (4) into independent finite mixtures at each time step, *i.e.*,

$$p_{\boldsymbol{\theta}}(\boldsymbol{x}_{0:T}) = \Big( \sum_{r_0 \in \mathcal{A}_K} p_{\boldsymbol{\theta}}(r_0) p_{\boldsymbol{\theta}}(\boldsymbol{x}_0 \mid r_0) \Big) \prod_{t=1}^{T} \Big( \sum_{r_t \in \mathcal{A}_K} p_{\boldsymbol{\theta}}(r_t) p_{\boldsymbol{\theta}}(\boldsymbol{x}_t \mid \boldsymbol{x}_{t-1}, r_t) \Big). \tag{22}$$

The global trajectory is now written as a product of local mixtures at each time step.

Under this factorization, and assuming linear independence of either the transition PDFs in $\mathcal{P}_{\mathcal{A}}$ or the initial PDFs in $\mathcal{P}_{\mathcal{A}}^0$, we obtain linear independence in the product family $\mathcal{P}_{\mathcal{A}}^0 \otimes \mathcal{P}_{\mathcal{A}}^{\otimes T}$. To observe that this is the case, we first write

$$p_{\boldsymbol{\theta}}(\boldsymbol{x}_{0:T}) = 0 \quad a.e. \quad \Longleftrightarrow \quad \Big( \sum_{r_0 \in \mathcal{A}_K} p_{\boldsymbol{\theta}}(r_0) p_{\boldsymbol{\theta}}(\boldsymbol{x}_0 \mid r_0) \Big) \prod_{t=1}^{T} \Big( \sum_{r_t \in \mathcal{A}_K} p_{\boldsymbol{\theta}}(r_t) p_{\boldsymbol{\theta}}(\boldsymbol{x}_t \mid \boldsymbol{x}_{t-1}, r_t) \Big) = 0 \quad a.e.. \tag{23}$$

This product of functions is equal to zero *a.e.* if and only if at least one of its components is zero *a.e.*, assuming not all components are degenerate. Then by linear independence of the functions in either the initial or transition family, we obtain $p_{\boldsymbol{\theta}}(r_t) = 0$ *a.e.* for at least one $t \in \{0, \ldots, T\}$. Since we started by assuming $p_{\boldsymbol{\theta}}(\boldsymbol{r}_{0:T}) = \prod_{t=0}^{T} p_{\boldsymbol{\theta}}(r_t)$, it follows that

$$\Big( \sum_{r_0 \in \mathcal{A}_K} p_{\boldsymbol{\theta}}(r_0) p_{\boldsymbol{\theta}}(\boldsymbol{x}_0 \mid r_0) \Big) \prod_{t=1}^{T} \Big( \sum_{r_t \in \mathcal{A}_K} p_{\boldsymbol{\theta}}(r_t) p_{\boldsymbol{\theta}}(\boldsymbol{x}_t \mid \boldsymbol{x}_{t-1}, r_t) \Big) = 0 \quad a.e. \quad \Longrightarrow \quad p_{\boldsymbol{\theta}}(\boldsymbol{r}_{0:T}) = 0 \quad a.e.. \tag{24}$$

Hence, the product family $\mathcal{P}_{\mathcal{A}}^0 \otimes \mathcal{P}_{\mathcal{A}}^{\otimes T}$ contains linearly independent functions. By Lem. A.6, this is sufficient to obtain identifiability up to permutation (Def. 2.1). ∎

We showed that linear independence of the transition distribution family implies linear independence of the product family, under the assumption of independent regimes. However, this generally does not hold when regimes may be dependent. In that case, an overlapping variable space challenges linear independence for any consecutive product of linearly independent distributions, as illustrated in Eq. (5) in the main text. This motivates the identifiability theory on MSMs with first- and higher-order overlapping variables proposed in Balsells-Rodas et al. (2024; 2026), summarised in App. A.3, the extension to Gaussian MSMs with regimes that depend on both past regimes and observed variables in Balsells-Rodas et al. (2025), and the extension to instantaneous effects and exponential families in this work.

For example, consider temporal regime dependence via a first-order Markov process. Then the summation indices $r_t$ are coupled across time and cannot be decomposed into a product of marginal summations. Subsequently, the resulting distribution remains a mixture over entire regime trajectories, rather than a product of one-step mixtures. We obtain

$$p_{\boldsymbol{\theta}}(\boldsymbol{x}_{0:T}) = \sum_{\boldsymbol{r}_{0:T} \in \mathcal{A}_K^{\times(T+1)}} p_{\boldsymbol{\theta}}(r_0) p_{\boldsymbol{\theta}}(\boldsymbol{x}_0 \mid r_0) \prod_{t=1}^{T} p_{\boldsymbol{\theta}}(r_t \mid r_{t-1}) p_{\boldsymbol{\theta}}(\boldsymbol{x}_t \mid \boldsymbol{x}_{t-1}, r_t). \tag{25}$$

This mixture cannot be simplified into a product of independent local mixtures without further assumptions.

### B.2. Identifiable Exponential Family Markov Switching Models

We generalise identifiable Gaussian MSMs with dependencies between regimes (Balsells-Rodas et al., 2024, Thm. 3.2) to other continuous minimal regular exponential family distributions, by leveraging fundamental work from Barndorff-Nielsen (1965), set out in Lem. A.4.

To start, by necessity of the condition in Yakowitz & Spragins (1968), the following lemma follows.

**Lemma B.3.** *Given Ass. A.5, assume the PDFs in the family $\mathcal{P}_{\mathcal{A}}$ (Eq. (4)) satisfy the conditions $(\overline{a1})$-$(\overline{a3})$ in Lem. A.4. Then the family $\mathcal{P}_{\mathcal{A}}$ contains linearly independent functions under finite mixtures (Def. A.2).*

*Proof*: The result follows directly from Lem. A.3 and Lem. A.4, which transfer to families of PDFs under Ass. A.5 without loss of generality. ∎

We proceed to show exponential family MSMs are identifiable up to permutation. We highlight that the result remains a special case of the nonparametric conditions in Lem. A.7.

**Proposition B.4** (Identifiable Exponential Family Markov Switching Models). *Given Ass. A.5, consider the family of MSMs $\mathcal{M}_{\mathcal{A}}^T$. Assume the PDFs in the initial and transition distribution family $\mathcal{P}_{\mathcal{A}}^0, \mathcal{P}_{\mathcal{A}}$ are from continuous minimal regular exponential families with regime- and history-dependent natural parameters $\boldsymbol{\theta}_a^0$ and $\boldsymbol{\theta}_a(\boldsymbol{x}_{t-1})$, respectively, given by*

$$\mathcal{P}_{\mathcal{A}}^0 = \left\{ p_{\boldsymbol{\theta}}(\boldsymbol{x}_0 \mid a) = h^0(\boldsymbol{x}_0) \exp\left(\boldsymbol{\theta}_a^0 \cdot \boldsymbol{\tau}^0(\boldsymbol{x}_0) - A^0(\boldsymbol{\theta}_a^0)\right) \mid a \in \mathcal{A}, \boldsymbol{x}_0 \in \mathcal{X} \right\}, \tag{26}$$

$$\mathcal{P}_{\mathcal{A}} = \left\{ p_{\boldsymbol{\theta}}(\boldsymbol{x}_t \mid \boldsymbol{x}_{t-1}, a) = h(\boldsymbol{x}_t) \exp\left(\boldsymbol{\theta}_a(\boldsymbol{x}_{t-1}) \cdot \boldsymbol{\tau}(\boldsymbol{x}_t) - A\left(\boldsymbol{\theta}_a(\boldsymbol{x}_{t-1})\right)\right) \mid a \in \mathcal{A}, \boldsymbol{x}_t, \boldsymbol{x}_{t-1} \in \mathcal{X}, t \in \{1, \ldots, T\} \right\}.$$

*Assume the following conditions hold:*

$(\overline{c1})$ *The sufficient statistic $\boldsymbol{\tau}^0$ and $\boldsymbol{\tau}$ are continuous a.e.;*

$(\overline{c2})$ *Rich image of the sufficient statistic: The set $\{\boldsymbol{\tau}(\boldsymbol{x}_t) \mid h(\boldsymbol{x}_t) > 0, \boldsymbol{x}_t \in \mathcal{X}\}$ contains a (non-empty) open set, and analogously for the sufficient statistic of the initial distribution;*

$(\overline{c3})$ *Unique natural parameters: For any $a \neq a' \in \mathcal{A}, t \geq 1$, we have*

$$\boldsymbol{\theta}_a(\boldsymbol{x}_{t-1}) = \boldsymbol{\theta}_{a'}(\boldsymbol{x}_{t-1}) \quad a.e. \quad \implies \quad a = a', \tag{27}$$

*and analogously for the natural parameters of the initial distribution;*

$(\overline{c4})$ *Real-analytic transition parameters: For all $a \in \mathcal{A}, t \geq 1$, the natural parameters $\boldsymbol{\theta}_a(\boldsymbol{x}_{t-1})$ are real-analytic a.e..*

*Then the MSMs in $\mathcal{M}_{\mathcal{A}}^T$ are identifiable up to permutation (Def. 2.1).*

*Proof:* By Lem. A.6 it suffices to show that the joint component family $\mathcal{P}_{\mathcal{A}}^0 \otimes \mathcal{P}_{\mathcal{A}}^{\otimes T}$ satisfies the conditions $(\overline{b1})$–$(\overline{b4})$ in Lemma A.7. We verify these using $(\overline{c1})$–$(\overline{c4})$.

$(\overline{b1})$ By $(\overline{c3})$, the initial PDFs $\mathcal{P}_{\mathcal{A}}^0$ form an exponential family with distinct parameters. Barndorff-Nielsen (1965) with $(\overline{c1})$-$(\overline{c3})$ shows that distinct natural parameters imply identifiability, which by Lem. B.3 imply linear independence under finite mixtures for any finite subset of $\mathcal{A}$. Therefore $\mathcal{P}_{\mathcal{A}}^0$ contains linearly independent functions under finite mixtures on $\mathcal{X}$.

$(\overline{b2})$ For any $a \neq a' \in \mathcal{A}$, consider the set $\mathcal{X}_{a,a'} = \left\{ \boldsymbol{x}_{t-1} \in \mathcal{X} \mid \boldsymbol{\theta}_a(\boldsymbol{x}_{t-1}) = \boldsymbol{\theta}_{a'}(\boldsymbol{x}_{t-1}) \right\}$. Again by Barndorff-Nielsen (1965) with $(\overline{c1})$-$(\overline{c3})$ and Lem. B.3, the transition family $\mathcal{P}_{\mathcal{A}}$ forms an exponential family with distinct natural parameters and is thus linearly independent under finite mixtures on $\mathcal{X}$. Therefore, for any $\mathcal{A}_K \subset \mathcal{A}$ with $K < \infty$, linear dependence occurs only at points where natural parameters coincide, that is, on $\widetilde{\mathcal{X}} = \bigcup_{a \neq a' \in \mathcal{A}_K} \mathcal{X}_{a,a'}$. By $(\overline{c3})$ and $(\overline{c4})$, we can define the following non-trivial analytic function $\widetilde{\boldsymbol{\theta}}_{a,a'}(\boldsymbol{x}_{t-1}) \triangleq \boldsymbol{\theta}_a(\boldsymbol{x}_{t-1}) - \boldsymbol{\theta}_{a'}(\boldsymbol{x}_{t-1})$. From Mityagin (2020, Prop. 0), the zero set of a non-trivial analytic function has Lebesgue measure zero, which implies zero measure of $\mathcal{X}_{a,a'}$ for any $a \neq a' \in \mathcal{A}$. Therefore, for any finite $\mathcal{A}_K \subset \mathcal{A}$, linear dependence can only occur on the set $\widetilde{\mathcal{X}}$ which has Lebesgue measure zero. Therefore, there cannot exist non-zero-measure sets $\mathcal{Y}' \subset \mathcal{X}$ and $\mathcal{Z} \subset \mathcal{X}$ on which linear dependence occurs, and hence condition $(\overline{b2})$ holds.

$(\overline{b3})$ Again, Barndorff-Nielsen (1965) with $(\overline{c1})$-$(\overline{c3})$ and Lem. B.3 show distinct natural parameters in exponential families imply linear independence. Therefore, for any $\boldsymbol{\beta} \in \mathcal{X}$, and any subset $\mathcal{A}_K \subset \mathcal{A}$ with $K < \infty$, the family $\{p_{\boldsymbol{\theta}}(\boldsymbol{x}_t \mid \boldsymbol{x}_{t-1} = \boldsymbol{\beta}, a) \mid a \in \mathcal{A}_K\}$ contains linearly dependent functions only if there exist $a \neq a' \in \mathcal{A}_K$ such that $\boldsymbol{\theta}_a(\boldsymbol{\beta}) = \boldsymbol{\theta}_{a'}(\boldsymbol{\beta})$. Equality of natural parameters implies repeating functions.

$(\overline{b4})$ Continuity in $\boldsymbol{x}_{t-1}$ follows from $(\overline{c1})$ and continuity of the exponential family.

Therefore, by Lems. A.6 and A.7, the MSMs family $\mathcal{M}_{\mathcal{A}}^T$ under (b1-b6) is identifiable up to permutation (Def. 2.1). ∎

## B.3. Identifiable Regime-Switching Structural Causal Models

We turn to more fine-grained conditions on regime-switching SCMs instead of distribution families. We restate Thm. 3.5 from the main text and include a proof.

**Theorem 3.5** (Identifiable Regime-Switching SCMs up to Permutation). *Consider an acyclic regime-switching SCM that satisfies conditional causal stationarity, conditional causal sufficiency, and Ass. 3.1 to 3.4. Then the induced MSM is identifiable up to permutation (Def. 2.1).*

*Proof strategy:*

- It suffices to show distinct regime-switching SCMs induce a family of MSMs $\mathcal{M}_{\mathcal{A}}^T$ that satisfies Prop. B.4;

- Ass. 3.1 to 3.3 constitute a unique pushforward of the noise distribution up to zero-measure sets for each regime $a \in \mathcal{A}$, conditional on previous time steps, while preserving the exponential family structure and properties;

- A finite basis expansion (Ass. 3.4) then guarantees a continuous minimal regular exponential family with distinct natural parameters and common sufficient statistic along the lines of Barndorff-Nielsen (1965);

- A real-analytic sufficient statistic and transition functions ensure real-analytic natural parameters, such that linear dependence only happens on zero-measure subsets of the overlapping variable space in the product distribution family.

*Proof:* We proceed to derive the PDFs in the transition distribution family $\mathcal{P}_{\mathcal{A}}$, induced by Eq. (1). Fix an arbitrary $a \in \mathcal{A}$ and $\boldsymbol{x}_{t-1} \in \mathcal{X}$, $t \in \{1, \ldots, T\}$. Consider exogenous noise from an exponential family that satisfies Ass. 3.1. Acyclicity implies there exists a topological order of variables such that we can recursively substitute $f_{a,1}, \ldots, f_{a,D}$ to obtain regime- and history-dependent measurable mappings $\Phi_a(\boldsymbol{x}_{t-1}, \cdot)$ from the noise space to the observed space at each time step.

Under Ass. 3.2 and 3.3, pointwise diffeomorphism with respect to $\boldsymbol{\epsilon}_t$ is preserved under partial composition and substitution, such that $\Phi_a(\boldsymbol{x}_{t-1}, \cdot)$ is a diffeomorphism *a.e.* for fixed $\boldsymbol{x}_{t-1}$. Acyclicity guarantees the induced mappings $\Phi_a(\boldsymbol{x}_{t-1}, \cdot)$ are well-defined. By Ass. 3.3, either the noise distribution has a trivial automorphism class, or there exists a canonical permutation (shared across regimes) under which the Jacobian $\frac{\partial f_a}{\partial \boldsymbol{\epsilon}_t}$ with respect to instantaneous noise can be permuted into a lower-triangular matrix with a strictly positive diagonal. The positive diagonal enforces coordinate-wise monotonicity along the canonical ordering, obtaining a *unique* monotone triangular transport, up to zero-measure sets, known as the Knothe-Rosenblatt rearrangement (Knothe, 1957; Rosenblatt, 1952; Villani, 2009). Consequently, the unique mappings condition in Ass. 3.2 implies that for $a \neq a' \in \mathcal{A}$ we obtain for almost every $\boldsymbol{x}_{t-1} \in \mathcal{X}$,

$$\Phi_a(\boldsymbol{x}_{t-1}, \cdot) = \Phi_{a'}(\boldsymbol{x}_{t-1}, \cdot) \quad a.e. \quad \implies \quad a = a'. \tag{28}$$

That is, we established injectivity between the regime- and history-dependent measurable mappings and the regime indices.

Continuing, using the change of variables formula and the exponential family form of Eq. (2), we obtain

$$
\begin{aligned}
p_{\boldsymbol{\theta}}(\boldsymbol{x}_t \mid \boldsymbol{x}_{t-1}, a) &= p_{\boldsymbol{\eta}}\big(\Phi_a^{-1}(\boldsymbol{x}_t, \boldsymbol{x}_{t-1})\big) \cdot \big| \det J_{\Phi_a^{-1}(\cdot, \boldsymbol{x}_{t-1})}(\boldsymbol{x}_t)\big| \\
&= h\big(\Phi_a^{-1}(\boldsymbol{x}_t, \boldsymbol{x}_{t-1})\big) \cdot \big| \det J_{\Phi_a^{-1}(\cdot, \boldsymbol{x}_{t-1})}(\boldsymbol{x}_t)\big| \cdot \exp\Big(\boldsymbol{\eta} \cdot \boldsymbol{\tau}\big(\Phi_a^{-1}(\boldsymbol{x}_t, \boldsymbol{x}_{t-1})\big) - A(\boldsymbol{\eta})\Big),
\end{aligned}
\tag{29}
$$

where $\Phi_a^{-1}(\cdot, \boldsymbol{x}_{t-1})$ denotes the inverse of $\Phi_a(\boldsymbol{x}_{t-1}, \cdot)$ with respect to $\boldsymbol{\epsilon}_t$ for fixed $\boldsymbol{x}_{t-1}$. This inverse exists because $\Phi_a(\cdot, \boldsymbol{x}_{t-1})$ is a pointwise diffeomorphism, and its Jacobian has non-zero determinant because the inverse of a pointwise diffeomorphism is continuously differentiable. The result is an exponential family distribution, still in canonical form, but with transformed base measure and sufficient statistic.

Since the original exponential family is minimal, the components of the sufficient statistic $\boldsymbol{\tau}$ are linearly independent and finite-dimensional in $\mathbb{R}^P$. By Ass. 3.4, the composition $\boldsymbol{\tau} \circ \Phi_a^{-1}(\cdot, \boldsymbol{x}_{t-1})$ lies in a $\widetilde{P}$-dimensional vector space ($\widetilde{P} \geq P$) spanned by a common sufficient statistic $\widetilde{\boldsymbol{\tau}}$, whose components are monomials and thus linearly independent *a.e.*. That is, there exists a regime- and history-dependent coefficient matrices $\boldsymbol{C}_a(\boldsymbol{x}_{t-1}) \in \mathbb{R}^{P \times \widetilde{P}}$ with full row rank $P$, such that

$$\boldsymbol{\tau} \circ \Phi_a^{-1}(\boldsymbol{x}_t, \boldsymbol{x}_{t-1}) = \boldsymbol{C}_a(\boldsymbol{x}_{t-1})\widetilde{\boldsymbol{\tau}}(\boldsymbol{x}_t) + \mathcal{R}_O(\boldsymbol{x}_t) \quad a.e.. \tag{30}$$

where $\mathcal{R}_O$ is a regime- and history-invariant remainder that can be absorbed into the base measure. This finite basis expansion ensures the original sufficient statistic remains linearly independent, while the sufficient statistic of the transformed

distribution can have $\widetilde{P} - P$ additional components. Furthermore, all regime- and history-dependent effects can be absorbed into the natural parameters, yielding a function $\boldsymbol{\theta} : \mathcal{X} \times \mathcal{A} \to \Theta$ representing the coefficients of the transformed basis.

We can similarly factorise the base measure into a common base measure and a regime- and history-dependent offset that can be absorbed into the log-partition, such that

$$p_{\boldsymbol{\theta}}(\boldsymbol{x}_t \mid \boldsymbol{x}_{t-1}, a) = \underbrace{\widetilde{h}(\boldsymbol{x}_t) \exp\left(\mathcal{R}_O(\boldsymbol{x}_t)\right)}_{\overline{h}(\boldsymbol{x}_t)} \exp\left( \underbrace{\left(\boldsymbol{C}_a^T(\boldsymbol{x}_{t-1})\boldsymbol{\eta}\right)}_{\boldsymbol{\theta}_a(\boldsymbol{x}_{t-1})} \cdot \widetilde{\boldsymbol{\tau}}(\boldsymbol{x}_t) - \underbrace{\log b_a(\boldsymbol{x}_{t-1}) - \widetilde{A}\left(\boldsymbol{C}_a^T(\boldsymbol{x}_{t-1})\boldsymbol{\eta}\right)}_{\overline{A}_a\left(\boldsymbol{\theta}_a(\boldsymbol{x}_{t-1}), \boldsymbol{x}_{t-1}\right)} \right), \quad (31)$$

where $\widetilde{A}\left(\boldsymbol{C}_a^T(\boldsymbol{x}_{t-1})\boldsymbol{\eta}\right) \triangleq \log\left( \int_{\mathcal{X}} \widetilde{h}(\boldsymbol{x}_t) \exp\left(\mathcal{R}_O(\boldsymbol{x}_t)\right) \exp\left(\left(\boldsymbol{C}_a^T(\boldsymbol{x}_{t-1})\boldsymbol{\eta}\right) \cdot \widetilde{\boldsymbol{\tau}}(\boldsymbol{x}_t)\right) d\boldsymbol{x}_t \right)$.

Thus, the transition distribution remains a minimal regular exponential family, albeit not necessarily the same family as the noise distribution, and the dimension of the sufficient statistic might increase. A standard example where the distribution class and sufficient statistic dimension is preserved is a Gaussian noise distribution with affine transformation. This results in a transition distribution that is again Gaussian, since the Gaussian family is closed under affine transformation.

An identical argument for the initial distribution family $\mathcal{P}_{\mathcal{A}}^0$, without conditioning on $\boldsymbol{x}_{t-1}$, gives

$$p_{\boldsymbol{\theta}}(\boldsymbol{x}_0 \mid a) = \overline{h^0}(\boldsymbol{x}_0) \exp\left(\boldsymbol{\theta}_a^0 \cdot \widetilde{\boldsymbol{\tau}}^0(\boldsymbol{x}_0) - \overline{A^0}_a(\boldsymbol{\theta}_a^0)\right), \quad (32)$$

with regime-dependent natural parameters $\boldsymbol{\theta}_a^0$.

It remains to show the conditions in Prop. B.4 are satisfied for the family of finite mixture distributions $\mathcal{M}_{\mathcal{A}}^T$ induced by the component family $\mathcal{P}_{\mathcal{A}}^0 \otimes \mathcal{P}_{\mathcal{A}}^{\otimes T}$. We consider the conditions one by one. We show the conditions hold for the PDFs in the transition distribution family, and note that an identical argument holds for the initial distribution family.

$(\overline{c1})$ By Ass. 3.1, the sufficient statistic $\boldsymbol{\tau}$ is real-analytic and thus continuous. By Ass. 3.2, $\Phi_a(\boldsymbol{x}_{t-1}, \cdot)$ is a pointwise diffeomorphism, such that the composition $\boldsymbol{\tau} \circ \Phi_a^{-1}(\cdot, \boldsymbol{x}_{t-1})$ is a continuous function of $\boldsymbol{x}_t$. This property is preserved in the finite basis expansion following Ass. 3.4, since the expansion corresponds to a full row rank linear reparametrisation of a continuous map, which preserves continuity.

$(\overline{c2})$ By Ass. 3.1, the set $\{\boldsymbol{\tau}(\boldsymbol{\epsilon}_t) \mid h(\boldsymbol{\epsilon}_t) > 0, \boldsymbol{\epsilon}_t \in \mathcal{X}_\epsilon\}$ contains an open set. By Ass. 3.2, $\Phi_a(\boldsymbol{x}_{t-1}, \cdot)$ is a pointwise diffeomorphism mapping open sets to open sets, such that the image of the composition $\boldsymbol{\tau} \circ \Phi_a^{-1}(\cdot, \boldsymbol{x}_{t-1})$ contains an open set. This property is preserved in the finite basis expansion following Ass. 3.4, since it corresponds to a full row rank surjective linear map from $\mathbb{R}^{\widetilde{P}}$ onto $\mathbb{R}^P$, and by the Open Mapping Theorem (Rudin, 1991) it maps open sets to open sets.

$(\overline{c3})$ We showed that under Ass. 3.1 to 3.3, the mappings $\Phi_a(\boldsymbol{x}_{t-1}, \cdot)$ are unique up to null sets. Furthermore, for minimal regular exponential families, PDFs are uniquely defined by their natural parameters (Brown, 1986), pointwise for $\boldsymbol{x}_{t-1}$. Following the finite basis expansion under Ass. 3.4, all regime- and history-dependent effects that lie in the span of the common sufficient statistic $\widetilde{\boldsymbol{\tau}}$ are absorbed into the natural parameters. By the injectivity condition in Ass. 3.4, equal natural parameters $\boldsymbol{\theta}_a(\boldsymbol{x}_{t-1})$ implies equal mappings $\Phi_a(\boldsymbol{x}_{t-1}, \cdot)$. Hence, for any $a \neq a' \in \mathcal{A}, t \in \{1, \ldots, T\}$,

$$\boldsymbol{\theta}_a(\boldsymbol{x}_{t-1}) = \boldsymbol{\theta}_{a'}(\boldsymbol{x}_{t-1}) \quad a.e. \quad \implies \quad \Phi_a(\boldsymbol{x}_{t-1}, \cdot) = \Phi_{a'}(\boldsymbol{x}_{t-1}, \cdot) \quad a.e. \quad \implies \quad a = a'. \quad (33)$$

$(\overline{c4})$ The property that $\boldsymbol{f}_a$ is jointly real-analytic in $(\boldsymbol{x}_{t-1}, \boldsymbol{\epsilon}_t)$ a.e. is preserved for $\Phi_a$ under acyclicity, partial composition and substitution. Define the mapping $(\boldsymbol{x}_{t-1}, \boldsymbol{\epsilon}_t) \mapsto (\boldsymbol{x}_{t-1}, \Phi_a(\boldsymbol{x}_{t-1}, \boldsymbol{\epsilon}_t))$. This mapping is jointly real-analytic a.e. in $(\boldsymbol{x}_{t-1}, \boldsymbol{\epsilon}_t)$ since this holds for $\Phi_a$. Furthermore, its Jacobian can be permuted to a block-triangular matrix with the identity in the block pertaining to $\boldsymbol{x}_{t-1}$. Pointwise for each $\boldsymbol{x}_{t-1} \in \mathcal{X}$, the Jacobian is non-zero. Then by the Real-Analytic Inverse Function Theorem (Krantz & Parks, 2002), the inverse $(\boldsymbol{x}_{t-1}, \boldsymbol{x}_t) \mapsto (\boldsymbol{x}_{t-1}, \Phi_a^{-1}(\boldsymbol{x}_{t-1}, \boldsymbol{x}_t))$ is jointly real-analytic on each connected component where the Jacobian is non-vanishing. By Ass. 3.1, the sufficient statistic $\boldsymbol{\tau}$ is real-analytic and thus the composition $\boldsymbol{\tau} \circ \Phi_a^{-1}$ of real-analytic functions is jointly real-analytic in $(\boldsymbol{x}_{t-1}, \boldsymbol{x}_t)$. By Ass. 3.4 and following a finite basis expansion of the real-analytic space spanned by $\boldsymbol{\tau} \circ \Phi_a^{-1}$, the coefficients of the transformed basis are an analytic function of $\boldsymbol{x}_{t-1} \in \mathcal{X}$.

Hence, we can apply Prop. B.4 to conclude that the finite mixture distributions in the family $\mathcal{M}_{\mathcal{A}}^T$ are identifiable up to permutation (Def. 2.1). $\blacksquare$

## B.4. Generalising to an Arbitrary Number of Lags

Extending identifiability theory to an arbitrary number of lags $L$ is possible under real-analytic assumptions on the transition distributions, as shown in Balsells-Rodas et al. (2026). To illustrate, we first re-define the initial and transition distribution families to accommodate $L$-order temporal dependencies:

$$\overline{\mathcal{P}}_{\mathcal{A}}^{0,L} := \Big\{ p_{\boldsymbol{\theta}}(\boldsymbol{x}_{0:L-1} \mid a) \mid a \in \mathcal{A} \Big\}, \quad \overline{\mathcal{P}}_{\mathcal{A}}^{L} = \Big\{ p_{\boldsymbol{\theta}}(\boldsymbol{x}_t \mid \boldsymbol{x}_{t-L:t-1}, a) \mid a \in \mathcal{A} \Big\}, \tag{34}$$

where $\boldsymbol{x}_0, \ldots, \boldsymbol{x}_{L-1}$ denote $L$ initial variables. The main challenge in the multi-lag setting is to ensure linear independence is preserved on the joint distribution family $\overline{\mathcal{P}}_{\mathcal{A}}^{0,L} \otimes \overline{\mathcal{P}}_{\mathcal{A}}^{L \otimes T}$ as $T$ grows. When $L > 1$, consecutive distributions share multiple overlapping variables. For example, let $L = 2$ and consider the joint family at time $T = 3$, i.e.,

$$\overline{\mathcal{P}}_{\mathcal{A}}^{0,L} \otimes \overline{\mathcal{P}}_{\mathcal{A}}^{L} \otimes \overline{\mathcal{P}}_{\mathcal{A}}^{L} = \Big\{ p_{\boldsymbol{\theta}}(\boldsymbol{x}_0, \boldsymbol{x}_1 \mid r_1) p_{\boldsymbol{\theta}}(\boldsymbol{x}_2 \mid \boldsymbol{x}_0, \boldsymbol{x}_1, r_2) p_{\boldsymbol{\theta}}(\boldsymbol{x}_3 \mid \boldsymbol{x}_1, \boldsymbol{x}_2, r_3) \Big\}, \tag{35}$$

where the same variables (e.g., $\boldsymbol{x}_1$) appear in multiple factors, both as conditioned and observed variables. This longer overlap interaction prevents a direct application of Lemma A.7 for general $T$. However, we can re-write the above structure and enable a similar technique as in the single-lag case by grouping variables into blocks, so that each block interacts through a single overlap.

$$\overline{\mathcal{P}}_{\mathcal{A}}^{0,L} \otimes \big( \overline{\mathcal{P}}_{\mathcal{A}}^{L} \otimes \overline{\mathcal{P}}_{\mathcal{A}}^{L} \big) = \Big\{ p_{\boldsymbol{\theta}}(\boldsymbol{x}_0, \boldsymbol{x}_1 \mid r_1) p_{\boldsymbol{\theta}}(\boldsymbol{x}_2, \boldsymbol{x}_3 \mid \boldsymbol{x}_0, \boldsymbol{x}_1, r_2, r_3) \Big\}, \tag{36}$$

where we want to prove linear independence where the transition distribution is a pointwise product of $L$-lagged transition families ($\overline{\mathcal{P}}_{\mathcal{A}}^{L} \otimes \overline{\mathcal{P}}_{\mathcal{A}}^{L}$ in our example). This formulation allows us to re-use the strategy on Lemma A.7, provided that the assumptions the $(\overline{b2}) - (\overline{b4})$ placed on $\overline{\mathcal{P}}_{\mathcal{A}}^{L}$ extend to the pointwise product $\overline{\mathcal{P}}_{\mathcal{A}}^{L} \otimes \overline{\mathcal{P}}_{\mathcal{A}}^{L}$.

Balsells-Rodas et al. (2026) shows this extension holds if assumptions $(\overline{b2})$ and $(\overline{b3})$ are strengthened. First, the non-zero Lebesgue measure set $\mathcal{Y}$ defined for $(\overline{b2})$ must be a full measure set. This is necessary because linear independence in the product family is constrained by the interaction between conditioned variable, which restricts the effective domain in which linear independence can be verified. Second, $(\overline{b3})$ needs to hold not only when fixing all the conditioned variables, but also when fixing subsets of the conditioned variables. Under parametric assumptions based on real-analytic transition distributions, these strengthened conditions are naturally satisfied. Analyticity already ensures that intersections between distinct regimes have Lebesgue measure zero, which allows $\mathcal{Y}$ in $(\overline{b2})$ to have full measure. Furthermore, the repeating functions condition extends to subsets of conditioned variables by virtue of assuming conditioned transition distributions and real-analytic functions. Therefore, the assumptions required to establish linear independence under finite mixtures extend to arbitrary lag $L$, resulting in identifiable multi-lag MSMs in our setting.

# C. Estimation Method Details

In the main text, we simplified causal parents to at most a single lag. Here, we describe our estimation method for an arbitrary number of lags $L$, modelled with a hyperparameter $\widetilde{L}$. This can be chosen through an elbow method using the data likelihood on a hold-out validation set, or simply as a large value exceeding the oracle value. In App. E.5, we show negligible loss in performance compared to using the oracle $\widetilde{L} = L$.

## C.1. `FlowMSM`

We follow Balsells-Rodas et al. (2024) in a Generalised Expectation-Maximisation (GEM) approach to estimate model parameters. We deviate in parametrising the initial and transition distributions with conditional normalizing flows (CNFs) instead of Gaussian transitions with autoregressive means parametrised by real-analytic neural networks.

### C.1.1. GENERALISED EXPECTATION-MAXIMISATION (GEM)

The GEM algorithm (Dempster et al., 1977) finds maximum likelihood estimates of parameters in latent variable models, in particular finite mixture models. Compared to the classic EM algorithm (Bishop, 2006), it uses gradient ascent to maximise the likelihood objective, as opposed to exact updates through the Viterbi algorithm, for example. A schematic overview of the model parameters in relation to the GEM objective is provided in Fig. 3 in the main text.

**E-Step.** Given some arrangement of model parameters $\boldsymbol{\theta}'$, the GEM objective is the expected log-likelihood of observations and latent regimes, *i.e.*,

$$\mathcal{L}(\boldsymbol{\theta} \mid \boldsymbol{\theta}') = \mathbb{E}_{p_{\boldsymbol{\theta}'}(\boldsymbol{r}_{0:T} \mid \boldsymbol{x}_{0:T})}\left[\log p_{\boldsymbol{\theta}}(\boldsymbol{x}_{0:T}, \boldsymbol{r}_{0:T})\right] \tag{37}$$

$$= \mathbb{E}_{p_{\boldsymbol{\theta}'}(\boldsymbol{r}_{0:T} \mid \boldsymbol{x}_{0:T})}\left[\log \left(p_{\boldsymbol{\theta}}(\boldsymbol{r}_{0:T})p_{\boldsymbol{\theta}}(\boldsymbol{x}_{0:\widetilde{L}-1} \mid r_{0:\widetilde{L}-1})\prod_{t=\widetilde{L}}^{T} p_{\boldsymbol{\theta}}(\boldsymbol{x}_t \mid \boldsymbol{x}_{t-\widetilde{L}:t-1}, r_t)\right)\right]$$

$$= \mathbb{E}_{p_{\boldsymbol{\theta}'}(\boldsymbol{r}_{0:T} \mid \boldsymbol{x}_{0:T})}\left[\log p_{\boldsymbol{\theta}}(\boldsymbol{r}_{0:T})\right] + \mathbb{E}_{p_{\boldsymbol{\theta}'}(\boldsymbol{r}_{0:T} \mid \boldsymbol{x}_{0:T})}\left[\log p_{\boldsymbol{\theta}}(\boldsymbol{x}_{0:\widetilde{L}-1} \mid r_{0:\widetilde{L}-1})\right] +$$

$$\sum_{t=\widetilde{L}}^{T} \mathbb{E}_{p_{\boldsymbol{\theta}'}(\boldsymbol{r}_{0:T} \mid \boldsymbol{x}_{0:T})}\left[\log p_{\boldsymbol{\theta}}(\boldsymbol{x}_t \mid \boldsymbol{x}_{t-\widetilde{L}:t-1}, r_t)\right].$$

Assuming a first-order stationary Markov chain $p_{\boldsymbol{\theta}}(\boldsymbol{r}_{0:T}) = p_{\boldsymbol{\theta}}(r_0)\prod_{t=1}^{T} p_{\boldsymbol{\theta}}(r_t \mid r_{t-1})$ for the latent regime structure, we introduce shorthand for the posterior probability, $\gamma_{t,a} \triangleq p_{\boldsymbol{\theta}}(r_t = a \mid \boldsymbol{x}_{0:T})$, and the posterior probability of two consecutive states, $\xi_{t,a,b} \triangleq p_{\boldsymbol{\theta}}(r_t = a, r_{t-1} = b \mid \boldsymbol{x}_{0:T})$.

Given the full set of model parameters $\boldsymbol{\theta} = (\boldsymbol{\pi}, Q, \psi^0, \psi, \boldsymbol{\mathcal{E}}^0_{1:\widetilde{K}}, \boldsymbol{\mathcal{E}}_{1:\widetilde{K}})$ introduced in the main text, the objective becomes

$$\mathcal{L}(\boldsymbol{\theta} \mid \boldsymbol{\theta}') = \sum_{a=1}^{\widetilde{K}} \gamma_{0,a} \log \pi_a + \sum_{t=1}^{T}\sum_{a=1}^{\widetilde{K}}\sum_{b=1}^{\widetilde{K}} \xi_{t,a,b} \log Q_{b,a} + \tag{38}$$

$$\sum_{a=1}^{\widetilde{K}}\sum_{t=0}^{\widetilde{L}-1} \gamma_{t,a} \log p_{\psi^0, \mathcal{E}_a^0}(\boldsymbol{x}_t \mid r_t = a) + \sum_{t=\widetilde{L}}^{T}\sum_{a=1}^{\widetilde{K}} \gamma_{t,a} \log p_{\psi, \mathcal{E}_a}(\boldsymbol{x}_t \mid \boldsymbol{x}_{t-\widetilde{L}:t-1}, r_t = a).$$

**M-Step.** The classic EM algorithm would update model parameters according to $\boldsymbol{\theta}^{\text{new}} \leftarrow \arg\max_{\boldsymbol{\theta}} \mathcal{L}(\boldsymbol{\theta} \mid \boldsymbol{\theta}')$. Instead, the GEM algorithm relaxes this to an iterative increase of the objective function, so using gradient ascent we obtain

$$\boldsymbol{\theta}^{\text{new}} \leftarrow \boldsymbol{\theta}' + \alpha \, \nabla_{\boldsymbol{\theta}} \mathcal{L}(\boldsymbol{\theta} \mid \boldsymbol{\theta}')|_{\boldsymbol{\theta}=\boldsymbol{\theta}'}, \tag{39}$$

evaluating the gradient $\nabla_{\boldsymbol{\theta}} \mathcal{L}(\boldsymbol{\theta} \mid \boldsymbol{\theta}')$ at $\boldsymbol{\theta}'$. Convergence is guaranteed to a local maximum (Dempster et al., 1977).

The equations above assume a single sample ($N = 1$), but in practice we use mini-batch stochastic gradient ascent with batch size $B$ when the number of samples $N$ is sufficiently large. Furthermore, exact update rules for $\boldsymbol{\pi}$ and $Q$ can be found in the literature for HMMs (Bishop, 2006), but this empirically requires a sufficiently large batch size to ensure consistent updates during training. Therefore, we use gradient ascent to update $\boldsymbol{\pi}$ and $Q$ together with the other parameters in $\boldsymbol{\theta}$.

**Sliding Window Representation.** For single samples ($N = 1$), we observe unstable training due to the relatively large number of model parameters and the inherent numerical instability that comes with the high flexibility of normalizing flows. Therefore, in the case of $N = 1$, we cut the time series into equal windows of size $\widetilde{T}$ with $\widetilde{d}_R/\widetilde{T}\%$ overlap, where $\widetilde{d}_R$ models the expected regime persistence. For example, for a single time series of length $T = 10,000$ with expected regime persistence of $\widetilde{d}_R = 20$, we cut the time series into equal windows of length $\widetilde{T} = 100$ with $20\%$ overlap. These hyperparameters balance computational stability versus losses in long-range dependencies in the regime posterior.

C.1.2. CONDITIONAL NORMALISING FLOWS (CNFs)

**Normalising Flows.** A normalising flow (Tabak & Vanden-Eijnden, 2010; Tabak & Turner, 2013) consists of a sequence of invertible mappings that transform a simple initial density into a complex target distribution. To illustrate, consider a random variable $\boldsymbol{z}_0 \sim p_0$ with known base density. Let $\boldsymbol{g}_1, \ldots, \boldsymbol{g}_M$ be a sequence of diffeomorphisms such that $\boldsymbol{z}_m = \boldsymbol{g}_m(\boldsymbol{z}_{m-1})$ for $m = 1, \ldots, M$, with $\boldsymbol{z}_M = \boldsymbol{g}_M \circ \cdots \circ \boldsymbol{g}_1(\boldsymbol{z}_0)$. Using the change of variables formula, the density of $\boldsymbol{z}_M$ becomes

$$p(\boldsymbol{z}_M) = p(\boldsymbol{z}_0) \prod_{m=1}^{M} \left| \det \frac{\partial \boldsymbol{g}_m^{-1}}{\partial \boldsymbol{z}_m} \right|, \tag{40}$$

where the determinant of the Jacobian is evaluated at $\boldsymbol{z}_m$.

*Table 2.* Design choices in the synthetic data-generating process.

(a) Synthetic Benchmarks

| Dataset | Structural Equations | Instantaneous Effects | Additive Noise | Linear SCM | Gaussian Noise |
|---|---|---|---|---|---|
| SVAR (Gauss.) | Eq. (42) | ✓ | ✓ | ✓ | ✓ |
| SVAR (Exp. Fam.) | Eq. (42) | ✓ | ✓ | ✓ | ✗ |
| ANM (Exp. Fam.) | Eq. (43) | ✓ | ✓ | ✗ | ✗ |
| LSNM (Exp. Fam.) | Eq. (44) | ✓ | ✗ | ✗ | ✗ |

(b) Data-Generation Parameters

| Hyperparameter | Value |
|---|---|
| Nr. Samples ($N$) | 100 |
| Nr. Time-Steps ($T$) | 100 |
| Nr. Dimensions ($D$) | 10 |
| Nr. Regimes ($K$) | 3 |
| Nr. Lags ($L$) | 1 |
| Edge Probability ($p_{\text{edge}}$) | 0.5 |
| Avg. Regime Persistence ($d_R$) | 10 |
| Min. Edge Weight ($w_{\text{edge}}$) | 0.25 |

**Conditional Normalising Flows.** To model the initial and transition distribution, we define $z_0 \sim p_0$ as exogenous noise of our choice and define $z_M$ as the observations $x_t$. The transformations $g_1, \ldots, g_M$ for the transition distribution are parametrised by $\psi$, and the transformation $g_1^0, \ldots, g_M^0$ for the initial distribution by $\psi^0$. We generalise to *conditional* normalising flows by conditioning the transformations on embeddings $\mathcal{E}_{1:\widetilde{K}}^0$ and $\mathcal{E}_{1:\widetilde{K}}$ for regimes $r_t \in \{1, \ldots, \widetilde{K}\}$, such that we can share parameters across regimes. For the transition distribution, we additionally condition on lagged variables $x_{t-\widetilde{L}:t-1}$ to obtain the density

$$p_{\psi, \mathcal{E}_{1:\widetilde{K}}}(x_t \mid x_{t-\widetilde{L}:t-1}, r_t) = p_0(z_0) \prod_{m=1}^{M} \left| \det \frac{\partial g_m^{-1}(z_m; x_{t-\widetilde{L}:t-1}, r_t)}{\partial z_m} \right|, \tag{41}$$

where the base density $p_0$ is independent of the conditioning variables.

Normalising flows fit well with our theoretical narrative of transforming exponential family noise through real-analytic diffeomorphisms. However, we stress that the invertible transformations estimated by a normalising flow do not equal the functional relationships in a regime-switching SCM, but are here merely used to estimate complex unknown initial and transition densities.

**Flow Architecture.** The initial and transition distribution are modelled with a separate flow. Preliminary experimentation among popular CNF architectures suggests the *Neural Spline Flow* (NSF) (Durkan et al., 2019) as a suitable design choice. We adopt the implementation provided by the `Zuko` package. The regime embedding and optional additional context are passed to a hypernetwork, which predicts spline transformation weights for a shared normalising flow. This shared flow consists of a sequence of monotonic rational-quadratic spline transformations.

### C.2. Causal Discovery

After clustering samples using the sample splitting scheme outlined in the main text, we deploy several well-known causal discovery methods to estimate window graphs $\widehat{G}_{1:\widetilde{K}}$ from the clustered samples. In our experiments we use several established methods, such as `VARLiNGAM` (Hyvärinen et al., 2010), `DYNOTEARS` (Pamfil et al., 2020), `PCMCI`$^+$ (Runge, 2020) and `Rhino` (Gong et al., 2023), which allow for instantaneous and lagged causal dependencies. Hyperparameter choices are described in App. D.2. We run causal discovery on each cluster embarrassingly parallel. We omit initial graph estimation for the sake of brevity, but this can be done using any non-temporal causal discovery method, such as the well-known `PC` algorithm (Spirtes et al., 2000).

## D. Experiments Details

### D.1. Synthetic Data-Generating Process

Tab. 2 specifies the design choices in the synthetic data-generating process. We generate synthetic data according to three classes of structural equations. Contrary to the simplifying assumption in the main text, where we restrict causal parents to at most a single lag, we allow for an arbitrary number of lags $L$ in synthetic experiments. As a sanity check, a representative sample is visualised in Fig. 6, demonstrating that linear and nonlinear transitions, as well as Gaussian and non-Gaussian noise distributions, could result in visually similar time series, although the underlying temporal dynamics are fundamentally different.

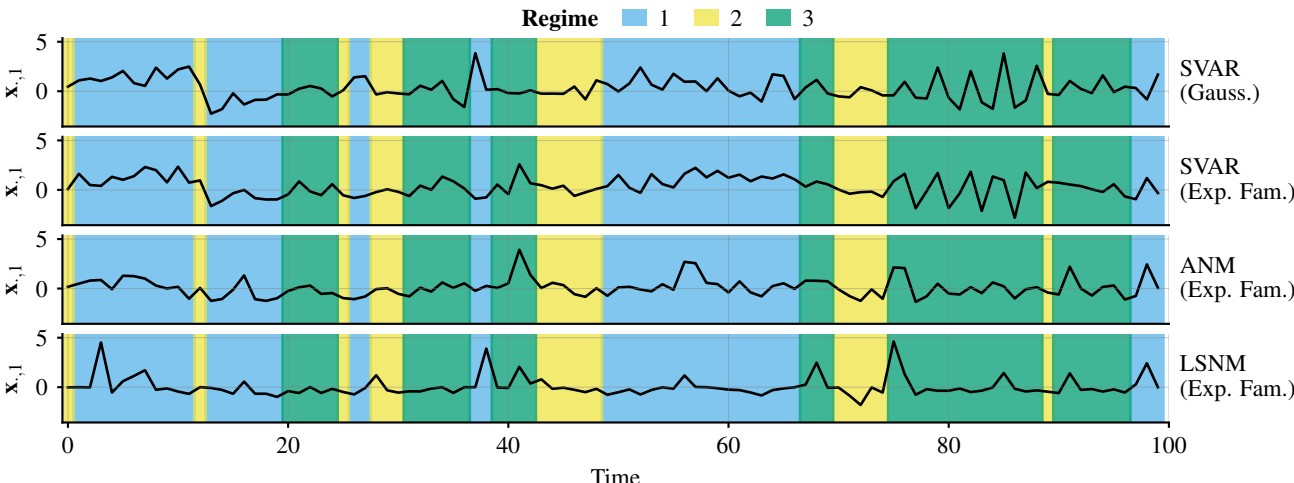

*Figure 6.* Representative time series corresponding to the synthetic regime-switching SCMs described in Tab. 2. Each row visualizes the same variable over time, where the only differences amount to the structural equations and exogenous noises.

Each dataset contains $N$ time series of $T$ time-points across $D$ dimensions, switching between $K$ regimes with an expected regime persistence of $d_R$. The expected regime persistence is inversely related to the persistence probability, such that $d_R = \frac{1}{1-p_{\text{persist.}}}$. We sample the regime persistence for each regime uniformly over the interval $[0.5 \cdot d_R, 1.5 \cdot d_R]$. Subsequently, we enforce the corresponding persistence probability in $\text{diag}(Q)$, while distributing the remaining probability mass uniformly over the other regimes. Each regime consists of a window graph with edges drawn independently with probability $p_{\text{edge}}$. For the lagged edges, this corresponds to an Erdös-Rényi random graph. For the instantaneous edges, we generate a lower triangular matrix with zero diagonal to secure a DAG, while randomly permuting the topological order of the nodes in each regime. Adding edge weights, we obtain coefficient matrices $\boldsymbol{W}_{a,0:L}$, where at most $L$ lags are included. Non-zero weights are drawn uniformly from $[-2 \cdot w_{\text{edge}}, -w_{\text{edge}}] \cup [w_{\text{edge}}, 2 \cdot w_{\text{edge}}]$, where $w_{\text{edge}}$ is the minimal edge weight.

**Linear SVAR** (Sims, 1980). A regime-switching version of the classic structural vector autoregressive model with simple linear equations. For a fixed regime $R_t = a$, we have

$$(0 \leq t \leq L-1) \quad \boldsymbol{X}_t \leftarrow \boldsymbol{W}_{a,0}\boldsymbol{X}_t + \boldsymbol{\mu}_a + \boldsymbol{\epsilon}_t, \tag{42}$$

$$(L \leq t \leq T) \quad \boldsymbol{X}_t \leftarrow \sum_{l=0}^{L} \boldsymbol{W}_{a,l}\boldsymbol{X}_{t-l} + \boldsymbol{\epsilon}_t,$$

for initial mean $\boldsymbol{\mu}_a$ and regime-dependent weighted adjacency matrices $\boldsymbol{W}_{a,0:L}$, where $\boldsymbol{W}_{a,0}$ is acyclic to guarantee recursive structural equations. We further require $(\boldsymbol{W}_{a,0}, \boldsymbol{\mu}_a) \neq (\boldsymbol{W}_{a',0}, \boldsymbol{\mu}_{a'})$ and $\boldsymbol{W}_{a,0:L} \neq \boldsymbol{W}_{a',0:L}$ for $a \neq a' \in \mathcal{A}_K$ to obtain distinct regimes. This is assumed to hold for the other SCMs below as well.

To ensure stability of the generated time series, we scale the weighted coefficient matrices by the spectral radius of the associated companion matrices. This guarantees the eigenvalues of the regime-dependent companion matrices lie within the unit circle, a necessary and sufficient condition for a stable VAR process (Lütkepohl, 2005).

**Nonlinear ANM** (Hoyer et al., 2008). We generalise to nonlinear equations to obtain a regime-switching temporal version of a nonlinear additive noise model. As the name suggests, ANMs are characterised by additive independent noise, and nonlinear functional dependencies are sufficient to obtain uniquely identifiable causal structures. The initial and transition PDFs are generally non-Gaussian, unless instantaneous effects are affine and exogenous noise is Gaussian.

We choose a $\tanh$ transform of instantaneous and lagged variables to introduce real-analytic nonlinearities that are contractive when the eigenvalues of the companion matrices lie within the unit circle. The structural equations become

$$(0 \leq t \leq L-1) \quad \boldsymbol{X}_t \leftarrow \tanh\left(\boldsymbol{W}_{a,0}\boldsymbol{X}_t + \boldsymbol{\mu}_a\right) + \boldsymbol{\epsilon}_t, \tag{43}$$

$$(L \leq t \leq T) \quad \boldsymbol{X}_t \leftarrow \tanh\left(\sum_{l=0}^{L} \boldsymbol{W}_{a,l}\boldsymbol{X}_{t-l}\right) + \boldsymbol{\epsilon}_t.$$

*Table 3.* Model hyperparameters in synthetic and real-world experiments.

(a) `FlowMSM`

| Hyperparameter | Value |
|---|---|
| Model Nr. Regimes ($\widetilde{K}$) | 3 |
| Model Nr. Lags ($\widetilde{L}$) | 1 |
| Expected Regime Persistence ($\widetilde{d}_R$) | 20 |
| Sliding Window Size ($\widetilde{T}$) | 100 |
| Batch Size ($B$) | 100 |
| Nr. Training Epochs | 500 |
| Nr. Warm-up Epochs | 20 |
| Nr. Training Restarts | 10 |
| Early-Stopping Patience | 10 |

(b) `Neural Spline Flow`

| Hyperparameter | Initial Flow | Transition Flow |
|---|---|---|
| Regime Embedding Dim. | 10 | 10 |
| Nr. Transforms | 1 | 2 |
| Nr. Bins | 4 | 8 |
| Nr. Hidden Layers | 1 | 1 |
| Hidden Layer Dim. | 8 | 32 |
| Activation Function | `GeLU` | `GeLU` |
| Random Feature Perm. | ✗ | ✓ |
| Residual Blocks | ✗ | ✗ |

**LSNM** (Immer et al., 2023; Strobl & Lasko, 2023). We relax the additive noise assumption to obtain a temporal regime-switching version of a location scale noise model, where the LSNM is also known as a heteroskedastic noise model. This constitutes our most challenging experimental setting. In the equations below, both instantaneous and lagged variables affect the scale of the noise distribution, not its location, introducing complex causal dependencies. Thus, our LSNM reduces to a multiplicative noise model, such that we obtain

$$(0 \leq t \leq L-1) \quad \boldsymbol{X}_t \leftarrow \Big(\mathbf{1} + \delta - \tanh\big(\boldsymbol{W}_{a,0}\boldsymbol{X}_t\big)\Big) \circ \Big(\mathbf{1} + \delta - \tanh\big(\boldsymbol{\mu}_a\big)\Big) \circ \boldsymbol{\epsilon}_t, \tag{44}$$

$$(L \leq t \leq T) \quad \boldsymbol{X}_t \leftarrow \Big(\mathbf{1} + \delta - \tanh\big(\boldsymbol{W}_{a,0}\boldsymbol{X}_t\big)\Big) \circ \Big(\mathbf{1} + \delta - \tanh\big(\sum_{l=1}^{L}\boldsymbol{W}_{a,l}\boldsymbol{X}_{t-l}\big)\Big) \circ \boldsymbol{\epsilon}_t,$$

where $\circ$ denotes the Hadamard product and $\delta > 0$ is a tiny constant to avoid multiplying by zero.

**Exogenous Noise.** In the setting with Gaussian noise, we sample $\boldsymbol{\epsilon}_t \stackrel{i.i.d.}{\sim} \mathcal{N}(\mathbf{0}, \boldsymbol{I})$. *Only for the linear SVAR, this means the initial and transition distribution family $\mathcal{P}_\mathcal{A}^0$ and $\mathcal{P}_\mathcal{A}$ remain Gaussian with PDFs, respectively,*

$$(0 \leq t \leq L-1) \quad p_{\boldsymbol{\theta}}(\boldsymbol{x}_t \mid a) = \mathcal{N}\big(\boldsymbol{M}_a \boldsymbol{\mu}_a, \Sigma_a\big), \tag{45}$$

$$(L \leq t \leq T) \quad p_{\boldsymbol{\theta}}(\boldsymbol{x}_t \mid \boldsymbol{x}_{t-L:t-1}, a) = \mathcal{N}\Big(\boldsymbol{M}_a \sum_{l=1}^{L}\boldsymbol{W}_{a,l}\boldsymbol{x}_{t-l}, \Sigma_a\Big),$$

using shorthand $\boldsymbol{M}_a \triangleq \big(\boldsymbol{I} - \boldsymbol{W}_{a,0}\big)^{-1}$ and $\Sigma_a \triangleq \boldsymbol{M}_a \boldsymbol{M}_a^T$. Acyclicity guarantees invertibility of $\boldsymbol{M}_a$. Gaussianity is further preserved in the product family, such that the resulting MSM amounts to a classic Gaussian mixture model.

In the exponential family setting, we sample noise from a Gamma distribution such that *(i)* $\epsilon_{t,d} \stackrel{i.i.d.}{\sim} \text{Gamma}(0.25, 2) - 0.5$, centred at zero, for all $d \in \{1, \ldots, D\}$. The Gamma distribution is heavily right-skewed to enforce non-Gaussianity. In App. E.2, we experiment with other exponential family distributions, such as *(ii)* $\text{Laplace}(0, 1/\sqrt{2})$ noise and *(iii)* the signed power nonlinear transformation of standard Gaussian independent noise, with exponent 1.5, introduced in Shimizu et al. (2006). Finally, we test distributions outside the exponential family, such as *(iv)* uniform $\mathcal{U}(-\sqrt{3}, \sqrt{3})$ noise and *(v)* $\text{Cauchy}(0, 1)$ noise. Unless specified otherwise, Gamma noise is used in the exponential family settings in synthetic experiments. For fairness of the comparison, all noise distributions are constructed to have zero mean and unit variance, except for the Cauchy distribution with undefined variance.

### D.2. Model Hyperparameters

Tab. 3 describes model hyperparameters used in synthetic and real experiments, unless specified otherwise. We make use of a 128-core AMD Rome CPU with 224GB memory for efficient parallelisation of model restarts, regimes and random seeds. Our implementation of `FlowMSM` allows for GPU computational speed-ups upon availability of such resources, although for small models parallelizing model restarts on a single CPU generally yields a favourable overall runtime.

### D.2.1. FLOWMSM

**Model Training.** We rely on the infrastructure provided in the `lightning` package, using default hyperparameters whenever unspecified. We split the available data into a training and validation set using an $80\%/20\%$ split. Subsequently, we train a model on the training set for at least 10 but a maximum of 500 epochs with a maximum batch size $B = 1,000$, and we use model checkpoints to select the model with the highest likelihood on the hold-out validation set. We repeat this procedure for 10 random initialisations of model parameters and pick the model with the highest validation likelihood. The transition matrix $Q$ is initialised with diagonal probabilities proportional to an initial regime persistence of 20, while the initial regime distribution is uniform. We use 20 warm-up epochs and to save compute we stop training early when the validation likelihood has not improved for 10 epochs. We further use the `AdamW` optimiser with decoupled weight decay, gradient clipping and learning rate decay for stable parameter updates.

**Nr. Regimes and Nr. Lags.** Although our identifiability theory does not presume knowledge of the oracle number of regimes $K$ and the maximum number of lags $L$ included in any particular causal parent set, during experiments we set the model number of regimes $\widetilde{K}$ and the model number of lags $\widetilde{L}$ equal to the oracle value. In App. E.5, we examine the effect of overspecifying $\widetilde{K}$ and $\widetilde{L}$ compared to the oracle value.

**Normalising Flows.** The initial and transition distribution are modelled with a separate flow. In the NSF architecture from the `zuko` package, we use 2 monotonic rational-quadratic spline transformation with 8 bins and a single hidden layer with 32 dimensions, `GeLU` activation functions and no residual blocks. Features are randomly permuted after transformation. Each regime embedding $\mathcal{E}_a$ has dimension $D \cdot \widetilde{L}$ equal to the dimension of the lagged context variables $x_{t-\widetilde{L}:t-1}$. For the initial flows modelling the initial distribution, a more shallow flow avoids overfitting to the low number of initial time steps.

### D.2.2. CAUSAL DISCOVERY

We use the sample splitting scheme described in the main text to obtain $\widetilde{K}$ clusters, and we run one of the causal discovery methods described below in parallel on each cluster of samples. As a simple heuristic, we require at least 30 short sample sequences to run causal discovery, otherwise we return an empty graph for that regime.

**DYNOTEARS** (Pamfil et al., 2020). We rely on the implementation in the `causalnex` package. Default hyperparameters apply, except for the regularisation factors for intra-slice and inter-slice weighted coefficients that are both set to 0.01, and the algorithm runs for a maximum of 1,000 iterations.

**VARLiNGAM** (Hyvärinen et al., 2010). We rely on the implementation in the `lingam` package. However, this implementation assumes a single time series, not $N$ such sequences. Therefore, we update the available implementation, but do so without adjusting the core algorithm logic. Furthermore, we prune edges using adaptive LASSO and identify edges as non-zero weights with absolute value exceeding a small threshold of 0.05. Code for our adaptation is available online.

**Rhino** (Gong et al., 2023). We rely on the implementation in the codebase of Varambally et al. (2024). We split the available data into a training and validation set using an $80\%/20\%$ split. Subsequently, we train a model on the training set for a maximum of 500 epochs with a maximum batch size of 1,000. The model with the highest likelihood on the hold-out validation set is selected through model checkpoints. We use a uniform graph prior and opt for the default hyperparameters set in Varambally et al. (2024). However, we adjust the early stopping patience and maximum number of steps in the inner loop of the augmented Lagrangian procedure to reduce computational costs and accommodate our resources.

**PCMCI$^+$** (Runge, 2020). We rely on the implementation in the `tigramite` package. Conditional independence is tested using a partial correlation $t$-test with significance level 0.05. We use majority ruling in the collider phase and we mark conflicts in orientation rules to obtain an order-independent algorithm. Orientation conflicts are later interpreted as undirected edges. Furthermore, we allow all lagged links to be detected in the MCI step. This improves detection power at the cost of a slightly larger runtime.

Finally, we highlight that the PCMCI$_0^+$ algorithm (Runge, 2020, Alg. 1-2) would be more appropriate for our regime-switching setting, yet to the best of our knowledge, no open-source implementation is available at the time of writing. The difference between the two methods is that the MCI test for variables $X_{t,i}$ and $X_{t-l,j}$ in PCMCI$_0^+$ does not condition on the lagged adjacency set of $X_{t-l,j}$. This is desirable in our setting, since we split samples based on the MAP estimate of $R_t$, while the parent set of $X_{t-l,j}$ is determined by $R_{t-l}$. Therefore, each cluster contains a mixed sample of variables $X_{t-l,j}$, and the MCI test may include a superset or subset of lagged adjacencies. We opt for a quick partial fix, available in the `tigramite` package, which excludes any variables preceding $X_{t-\widetilde{L}}$ in an MCI test that includes $X_{t,i}$.

### D.3. Baselines

Whenever applicable to the baselines below, the model number of regimes $\widetilde{K}$ and the model number of lags $\widetilde{L}$ equal the oracle value. Furthermore, we run a maximum of 500 epochs with a maximum batch size of $1,000$ whenever possible.

`iMSM (Neural)` (Balsells-Rodas et al., 2024). We rely on the implementation in the codebase of Balsells-Rodas et al. (2024), using default hyperparameters. The conceptual difference in regime detection compared to `FlowMSM` amounts to a parametrisation of initial and transition distributions using Gaussian densities with autoregressive means and fixed covariance matrices. Autoregressive means are parametrised through an MLP with 2 hidden layers with 32 dimensions and `SoftPlus` activation functions. This method does not estimate initial nor window graphs.

`iMSM (Poly)` (Balsells-Rodas et al., 2024). The conceptual difference with `iMSM (Neural)` is that the autoregressive means in the Gaussian transition distributions are parametrised using cubic polynomials. This allows for exact parameter updates through the EM algorithm, see Balsells-Rodas et al. (2024, App. E) for details. Due to instability observed during training, we decrease the number of epochs to 100 and the number of warmup epochs to 10.

`CASTOR` (Rahmani & Frossard, 2025a). We rely on the implementation in the codebase of Rahmani & Frossard (2025a), using default hyperparameters whenever possible. We are forced to implement a minor modification to avoid an infinite training loop when the loss objective does not converge. We opt for the nonlinear version with default hyperparameters, 5 iterations, initial windows of 10 time steps and no minimum regime persistence, which we think is reasonable given $T = 100$ and a ground-truth regime persistence of 10. When $T$ does not equal 100 in synthetic experiments, we adjust the initial window length accordingly.

`FANTOM` (Rahmani & Frossard, 2025b). We rely on the implementation in the codebase of Rahmani & Frossard (2025b), with the same minor modification to avoid an infinite training loop when the loss objective does not converge. We opt for hyperparameters aligned with `CASTOR`, and default ones whenever available. We decrease the maximum number of steps in the inner loop of the augmented Lagrangian procedure to reduce computational costs and accommodate our resources. We are able to reproduce the results of the synthetic experiments in Rahmani & Frossard (2025b) with these hyperparameters.

`R-PCMCI` (Saggioro et al., 2020). We rely on the implementation in the `tigramite` package, with default hyperparameters and design choices. We set the maximum number of transitions to 10, which we think is reasonable given $T = 100$ and a ground-truth regime persistence of 10. `CASTOR`, `FANTOM` and `R-PCMCI` all expect a single time series, necessitating independent deployment on each of $N$ time series individually.

`MCD` (Varambally et al., 2024). We rely on the implementation in the codebase of Varambally et al. (2024). Hyperparameters are set identical to `Rhino`, with the addition of a uniform prior over mixture membership.

`CD-NOD` (Huang et al., 2020). We rely on the implementation in the `causallearn` package, with default hyperparameters. However, only phase I and II of the method are implemented in this package. We estimate summary window graphs along the lines of Huang et al. (2020, Alg. 5), resulting in a PDAG that is *CD-NOD-equivalent* (Huang et al., 2020, Def. 3). For fairness, we evaluate against a summary window graph constructed as the union of regime-dependent window graphs.

The output of this implementation of `CD-NOD` is a summary partially-directed acyclic graph (PDAG) with an additional time/context surrogate variable that indicates non-stationarity/heterogeneity across time/contexts. Variables with changing causal mechanisms over time are adjacent to the surrogate variable, and causal direction between non-context nodes can in certain cases be inferred by leveraging invariance of causal mechanisms, independently changing modules and orientation rules. For fairness of the evaluation to oracle window graphs, we omit the surrogate variable during evaluation.

`SpaceTime` (Mameche et al., 2025a). We rely on the implementation in the codebase of Mameche et al. (2025a), with default hyperparameters. We set the minimal regime duration equal to the oracle regime persistence. Due to the large number of computationally expensive Gaussian processes, it is for us computationally infeasible to run the algorithm on $N$ time series simultaneously. Arguably, this would also not be desirable, since the output is a regime partition that is identical across all $N$ sequences, while the oracle regime partition is not. Therefore, we deploy on each of $N$ time series individually, obtaining a regime partition and a single window graph for each. Furthermore, `SpaceTime` assumes parent sets are invariant across regimes, while the oracle window graphs are not, by design of our synthetic data-generating process. Therefore, to obtain a fair evaluation, we interpret the graphical output as a summary window graph and evaluate it against an oracle summary window graph, identical to the procedure for `CD-NOD`.

### D.4. Evaluation Metrics

**Regimes.** We evaluate the estimated regimes $\{\widehat{r}_{0:T}^{(n)}\}_{n=1}^N$ as compared to the oracle $\{r_{0:T}^{(n)}\}_{n=1}^N$. We compute the Normalised Mutual Information (NMI) and Adjusted Rand Index (ARI) to measure similarity between two labellings. We further compute accuracy and F1 after applying a label permutation $\widehat{\phi} : \{1, \ldots, \widetilde{K}\} \to \{1, \ldots, K\}$ using Hungarian matching (Kuhn, 1955). In the ablation studies where the model number of regimes $\widetilde{K}$ exceeds the true number of regimes $K$, the Hungarian algorithm leaves some labels unmatched.

**DAGs.** `FlowMSM`, combined with `VARLiNGAM`, `DYNOTEARS` or `Rhino`, returns DAGs $\widehat{G}_{1:\widetilde{K}}$. After applying the label permutation $\widehat{\phi}$, these can be directly evaluated against oracle window graphs $G_{1:K}$. We compute accuracy, F1 and SHD of the corresponding adjacency matrices and average over $K$ regimes. SHD (Tsamardinos et al., 2006, Alg. 4) counts the number of edge insertions, deletions and flips in order to transform an estimated partially-directed acyclic graph (PDAG) into the oracle PDAG. Thus, a lower score is better. Wrongly oriented edges are counted as one mistake.

**CPDAGs.** `FlowMSM`, combined with `PCMCI`$^+$, returns completed partially-directed acyclic graphs (CPDAGs) that cannot be readily evaluated against the oracle window graphs. Therefore, we transform each oracle DAG to its corresponding CPDAG and evaluate performance with respect to the corresponding graph estimate, averaging over $K$ regimes. We treat edges with conflicts in orientation rules as undirected edges. The resulting scores are incomparable to the metrics computed over DAGs, due to the presence of undirected edges. Therefore, for all methods returning DAGs, we transform the estimated graphs to CPDAGs, and evaluate against oracle CPDAGs.

**Summary PDAG.** `CD-NOD` returns a summary PDAG, and for fairness of the evaluation we interpret the output of `SpaceTime` as a summary PDAG too. This is because `SpaceTime` outputs a single window graph per time series, following the assumption that parent sets are invariant across regimes. However, the parent sets of the oracle window graphs are not, by design of our synthetic data-generating process.

We compute the summary PDAG associated with the set of oracle window graphs by taking the union adjacency matrix over regimes and lags. An oriented edge indicates this orientation agrees in all regimes, while an undirected edge indicates both orientations appear in at least one regime. We note that the resulting PDAG is generally a fully connected undirected graph, since by design the causal mechanisms are rarely invariant across regimes. The resulting scores are incomparable to the metrics computed over DAGs or CPDAGs, since we now deal with summary graphs. Therefore, for all methods returning DAGs, we transform the estimated graphs to summary PDAGs, and evaluate against oracle summary PDAGs.

## E. Additional Experiments

We provide a brief overview of the experiments on synthetic and real-world benchmarks provided in this section.

In App. E.1, we study the impact of oracle regime assignments on the estimation of window graphs, thereby isolating the effect of regime uncertainty on causal discovery.

In App. E.2, we examine various non-Gaussian noise distributions.

In App. E.3, we evaluate `MCD` (Varambally et al., 2024) on non-temporal synthetic mixtures of causal models.

In App. E.4, we assess `FlowMSM (PCMCI`$^+$`)`, `CD-NOD` (Huang et al., 2020) and `SpaceTime` (Mameche et al., 2025a) on *CPDAGs* and *summary window graphs*, respectively.

In App. E.5, we analyse sensitivity to data-generation and model hyperparameters, and provide computation runtimes.

In App. E.6, we explore the causal implications and limitations of the Fama-French model, and run our analysis on *monthly* instead of *daily* data.

### E.1. Oracle Regime Information

In Tab. 1, we provide an overview of our evaluation metrics, of which the NMI and SHD score are visualised in the main text in the right column of Fig. 4.

Notably, several baselines report moderate to high F1 and accuracy scores, yet a small NMI and ARI. This is not a reporting mistake, but the result of a fundamental difference between these evaluation metrics. While the respective methods seem to estimate individual regimes reasonably well after applying a label permutation, the predicted clustering contains essentially

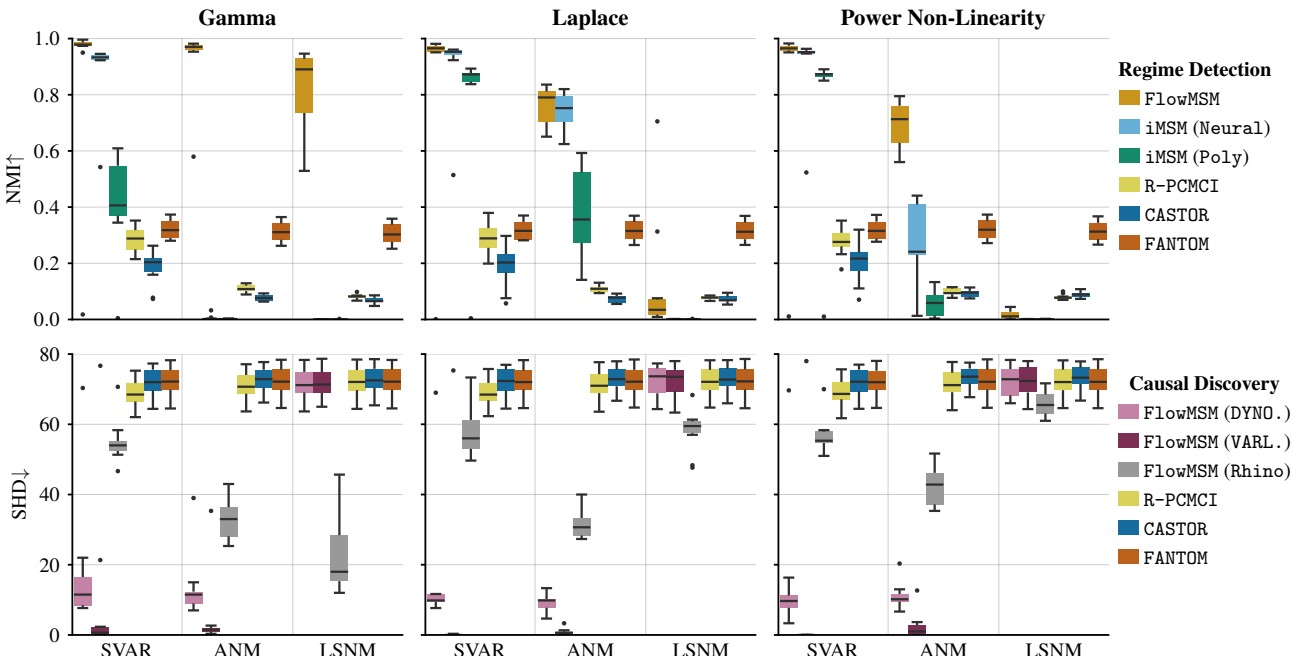

*Figure 7.* Evaluation of (*top*) regime detection and (*bottom*) causal discovery on synthetic benchmarks generated with (*left*) noise $\epsilon_{t,d} \overset{i.i.d.}{\sim}$ Gamma$(0.25, 2) - 0.5$, (*centre*) noise $\epsilon_{t,d} \overset{i.i.d.}{\sim}$ Laplace$(0, 1/\sqrt{2})$, and (*right*) the signed power nonlinear transformation of standard Gaussian noise, with exponent 1.5, introduced in Shimizu et al. (2006).

no information about the true regime partition. This may be because the label permutation artificially inflates F1 and accuracy scores, while the agreement between the true and estimated partition is close to random chance.

We include oracle causal discovery methods in the causal discovery step, aided by oracle regime information. This isolates the effect of regime uncertainty on causal discovery. We observe a moderate to small gap between `FlowMSM` and the oracle methods. Although the chosen causal discovery method is sensitive to the quality of the recovered regime partitions, in particular in the case of `VARLiNGAM`, we see these synthetic results as evidence that decoupling regime detection from regime-dependent causal discovery may be regarded as a reasonable modelling strategy.

### E.2. Non-Gaussian Noise Distributions

We experiment with several non-Gaussian noise distributions within the exponential family, shown in Fig. 7, among which the Gamma distribution displayed in the main text. For fairness, all noise distributions are constructed to have zero mean and unit variance.

Notably, the LSNM class with either the Laplace distribution or the power non-linearity seem to be challenging for `FlowMSM`. We hypothesize this is due to the non-smooth likelihood distortions with sharp ridges and heavy tails introduced by these distributions, which are preserved in the observed data likelihood under diffeomorphic transformations. Contrary, the conditional normalizing flows in `FlowMSM` rely on smooth maps with well-conditioned and numerically stable Jacobian estimates. These inductive biases may better suit Gamma independent noise. This is further substantiated by similar behaviour observed in causal discovery by `FlowMSM (Rhino)`, which also relies on conditional normalizing flows.

On the other hand, Laplace independent noise seems to be better suited than Gamma distributed noise for the Gaussian `iMSM` variants on the linear SVAR and nonlinear ANM benchmarks. Possibly, the sharp cusp at the origin in the Laplace distribution is less challenging than the heavy tail of the Gamma distribution.

We further experiment with distributions outside the exponential family in Fig. 8. Although `FlowMSM` remains strong for uniform noise, performance degrades substantially for the Cauchy distribution, yet the same holds for baseline methods. A reason could be that we observe a small number of large outliers in the generated time-series due to the undefined variance of the Cauchy distribution, which likely destabilizes training for all methods. We posit such complex non-stationary time-series warrant novel theory and methods.

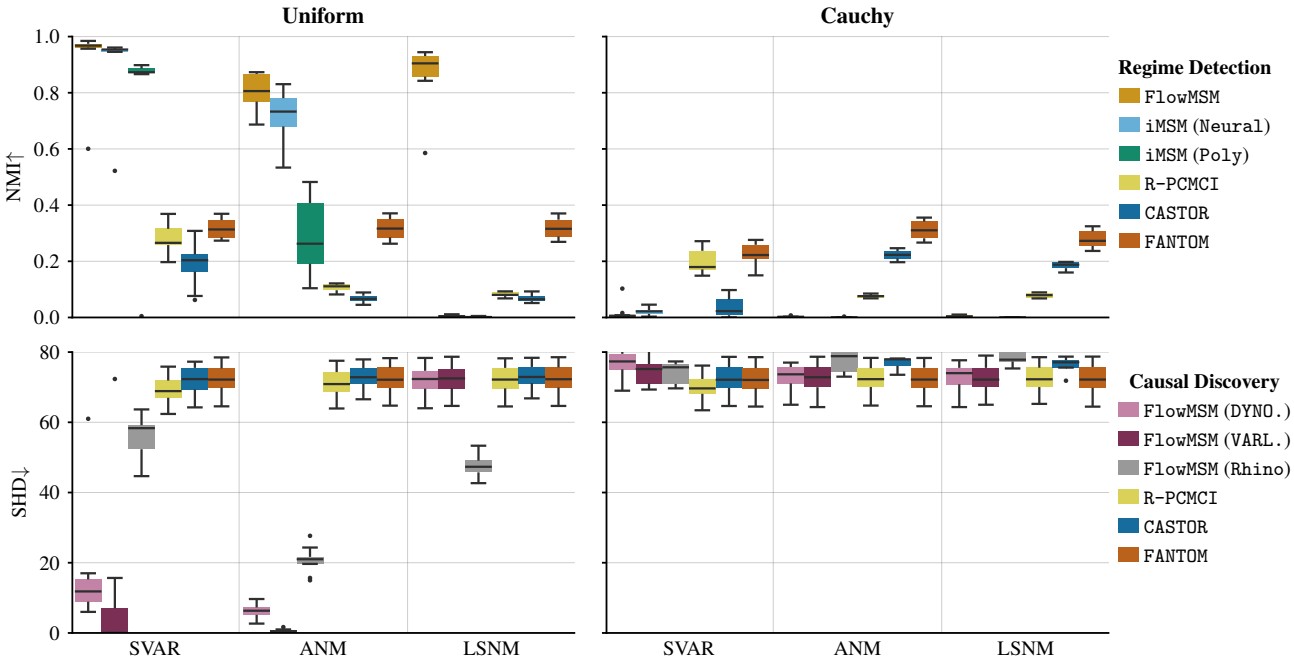

*Figure 8.* Evaluation of (*top*) regime detection and (*bottom*) causal discovery on synthetic benchmarks generated with non-exponential family noise (*left*) $\epsilon_{t,d} \overset{i.i.d.}{\sim} \mathcal{U}(-\sqrt{3}, \sqrt{3})$ and (*right*) $\epsilon_{t,d} \overset{i.i.d.}{\sim}$ Cauchy$(0, 1)$.

### E.3. Mixtures of Causal Models

As discussed in Sec. 2, our theory and method include (non-temporal) mixtures of causally stationary time series as a special case. We empirically validate this claim for temporal mixtures of causally stationary time series here. We repeat the experiments in the main text on multiple time series of length $T = 100$, but with the restriction that $R_t = R_{t'}$ for any $t \neq t'$. This corresponds to the conceptual setup in Varambally et al. (2024), although the exact synthetic data-generating process and regime-switching SCMs are different.

Experimental results are summarised in Tab. 4. We observe both FlowMSM and MCD recover near perfect mixture assignments across all synthetic benchmarks. The other baselines, with the exception of FANTOM, show moderate to strong performance on linear SVARs, yet deteriorate on the more complex nonlinear additive and multiplicative noise settings.

In causal discovery, it is unsurprising that the causally stationary methods that are aided by oracle regime information demonstrate strong performance, contingent on the regime-switching SCM. Given the near perfect mixture assignments, FlowMSM is able to perform close to, or in some cases even better than, the oracle methods. This is not mimicked by MCD, except for the LSNM class, which is expected given that Rhino is the underlying causal discovery method. The difference with the oracle method may be a consequence of hyperparameter tuning.

### E.4. CPDAGs and Summary Graphs

We compare against FlowMSM (PCMCI+), CDNOD and SpaceTime. Experimental results are shown in Fig. 9.The results on CPDAGs do not differ substantially from the results on DAGs, shown in Fig. 4. The reason is that, by design of the synthetic data-generating process, most edges in the CPDAG corresponding to a synthetic DAG are oriented. For lagged edges, this is due to the temporal arrow of time. Furthermore, most instantaneous edges are involved in a $v$-structure due to an edge density of close to 0.5, and can thus be oriented.

By design of the synthetic data-generating process, the oracle summary PDAG is generally a fully connected undirected graph, since the causal mechanisms are rarely invariant across regimes. This is correctly picked up on by FlowMSM, contingent on the chosen causal discovery method. None of the baselines seem to recover the correct graph. Besides the challenging setting, for CD-NOD we hypothesize this may be partially due to the partial correlation conditional independence test, chosen for computational feasibility given our resources.

*Table 4.* Mixtures of causal models, *i.e.*, each time series belongs to a regime and there are no regime switches. Evaluation on $N = 100$ synthetic time series of length $T = 100$ and dimension $D = 10$ with $K = 3$ regimes, averaged over 10 seeds. Our framework and the best scores are in **bold**, and second-best scores are underlined.

| Regime Detection | SVAR (Gauss.) | | | | SVAR (Exp. Fam.) | | | | ANM (Exp. Fam.) | | | | LSNM (Exp. Fam.) | | | |
|---|---|---|---|---|---|---|---|---|---|---|---|---|---|---|---|---|
| | NMI↑ | ARI↑ | F1↑ | Acc.↑ | NMI↑ | ARI↑ | F1↑ | Acc.↑ | NMI↑ | ARI↑ | F1↑ | Acc.↑ | NMI↑ | ARI↑ | F1↑ | Acc.↑ |
| **FlowMSM** | **0.99** | **1.00** | **1.00** | **1.00** | 0.94 | 0.96 | 0.96 | 0.96 | **0.99** | **1.00** | **1.00** | **1.00** | 0.97 | 0.99 | 0.99 | 0.99 |
| iMSM(Neural) | 0.95 | 0.98 | 0.99 | 0.99 | 0.75 | 0.71 | 0.83 | 0.81 | 0.02 | 0.03 | 0.44 | 0.42 | 0.00 | 0.00 | 0.40 | 0.38 |
| iMSM(Poly) | 0.87 | 0.93 | 0.98 | 0.97 | 0.42 | 0.41 | 0.72 | 0.67 | 0.00 | 0.01 | 0.41 | 0.39 | 0.00 | 0.00 | 0.40 | 0.38 |
| R-PCMCI | 0.95 | 0.95 | 0.99 | 0.98 | 0.86 | 0.86 | 0.97 | 0.95 | 0.00 | 0.00 | 0.74 | 0.60 | 0.00 | 0.00 | 0.71 | 0.56 |
| CASTOR | 0.56 | 0.56 | 0.97 | 0.96 | 0.54 | 0.54 | 0.97 | 0.96 | 0.00 | 0.00 | 0.80 | 0.69 | 0.00 | 0.00 | 0.87 | 0.78 |
| FANTOM | 0.00 | 0.00 | 0.25 | 0.14 | 0.00 | 0.00 | 0.26 | 0.15 | 0.00 | 0.00 | 0.22 | 0.12 | 0.00 | 0.00 | 0.22 | 0.12 |
| MCD | 0.94 | 0.92 | 0.92 | 0.94 | **0.99** | **0.99** | **0.99** | **1.00** | 0.98 | 0.99 | 0.98 | 0.98 | 0.94 | 0.91 | 0.90 | 0.93 |

| Causal Discovery | SHD↓ | F1↑ | Acc.↑ | SHD↓ | F1↑ | Acc.↑ | SHD↓ | F1↑ | Acc.↑ | SHD↓ | F1↑ | Acc.↑ |
|---|---|---|---|---|---|---|---|---|---|---|---|---|
| DYNOTEARS | 74.60 | 0.60 | 0.59 | 75.53 | 0.59 | 0.58 | 69.00 | 0.62 | 0.62 | 72.83 | 0.49 | 0.64 |
| DYNOTEARS* | **30.47** | **0.81** | **0.83** | 32.90 | 0.80 | 0.81 | 17.93 | 0.88 | 0.88 | 72.30 | 0.51 | 0.64 |
| **FlowMSM**(DYNO.) | 30.60 | **0.81** | **0.83** | 36.53 | 0.78 | 0.79 | 17.63 | 0.88 | 0.89 | 72.80 | 0.52 | 0.63 |
| VARLiNGAM | 78.27 | 0.57 | 0.56 | 75.50 | 0.59 | 0.59 | 65.57 | 0.65 | 0.65 | 72.87 | 0.49 | 0.64 |
| VARLiNGAM* | 54.87 | 0.67 | 0.67 | **5.13** | **0.96** | **0.97** | **6.90** | **0.96** | **0.97** | 72.07 | 0.50 | 0.64 |
| **FlowMSM**(VARL.) | 54.93 | 0.67 | 0.68 | 13.73 | 0.92 | 0.93 | **6.90** | **0.96** | **0.97** | 72.37 | 0.51 | 0.64 |
| Rhino | 74.27 | 0.60 | 0.60 | 74.57 | 0.60 | 0.60 | 64.40 | 0.66 | 0.65 | 56.40 | 0.71 | 0.70 |
| Rhino* | 59.20 | 0.67 | 0.67 | 58.57 | 0.68 | 0.68 | 33.37 | 0.81 | 0.82 | 23.83 | 0.86 | 0.88 |
| **FlowMSM**(Rhino) | 59.50 | 0.66 | 0.67 | 58.83 | 0.68 | 0.69 | 33.53 | 0.81 | 0.82 | **21.57** | **0.88** | **0.89** |
| R-PCMCI | 72.67 | 0.50 | 0.64 | 72.21 | 0.50 | 0.64 | 70.01 | 0.54 | 0.65 | 72.40 | 0.52 | 0.64 |
| CASTOR | 72.42 | 0.52 | 0.64 | 72.43 | 0.52 | 0.64 | 72.99 | 0.53 | 0.63 | 73.16 | 0.51 | 0.63 |
| FANTOM | 72.62 | 0.51 | 0.64 | 72.63 | 0.51 | 0.64 | 72.84 | 0.51 | 0.64 | 72.86 | 0.50 | 0.64 |
| MCD | 62.90 | 0.63 | 0.65 | 58.00 | 0.68 | 0.69 | 33.03 | 0.81 | 0.83 | 30.30 | 0.82 | 0.84 |

*Methods aided with oracle regime information.

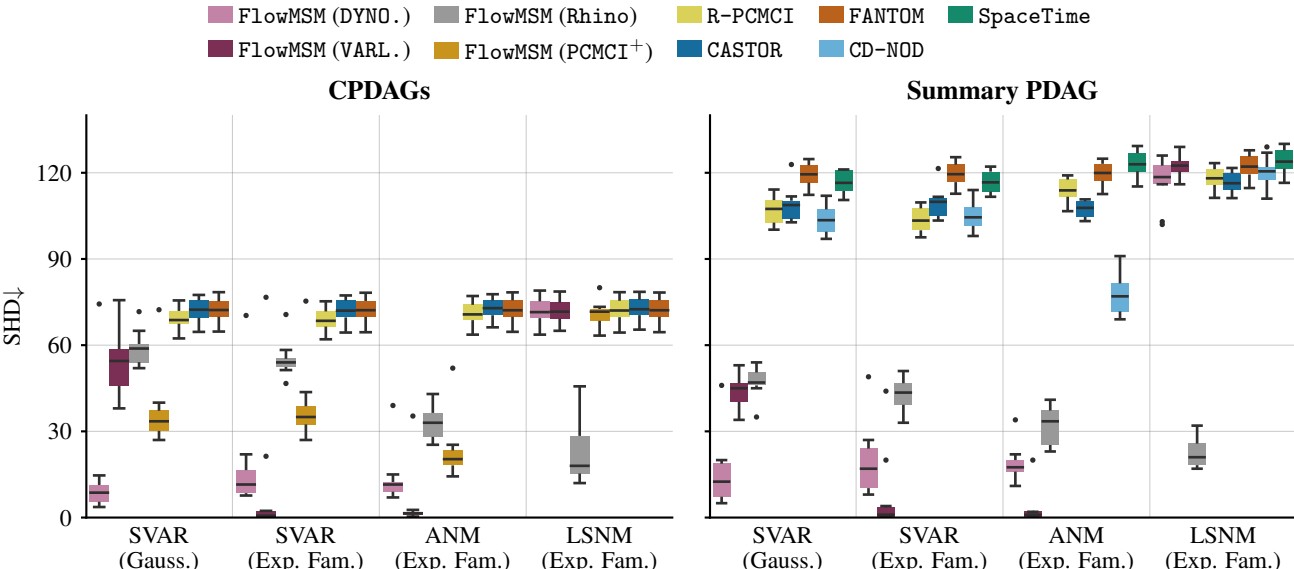

*Figure 9.* Evaluation of (*bottom*) causal discovery on synthetic benchmarks, evaluated as (*left*) CPDAGs and (*right*) summary PDAGs, as opposed to the DAGs in the main text.

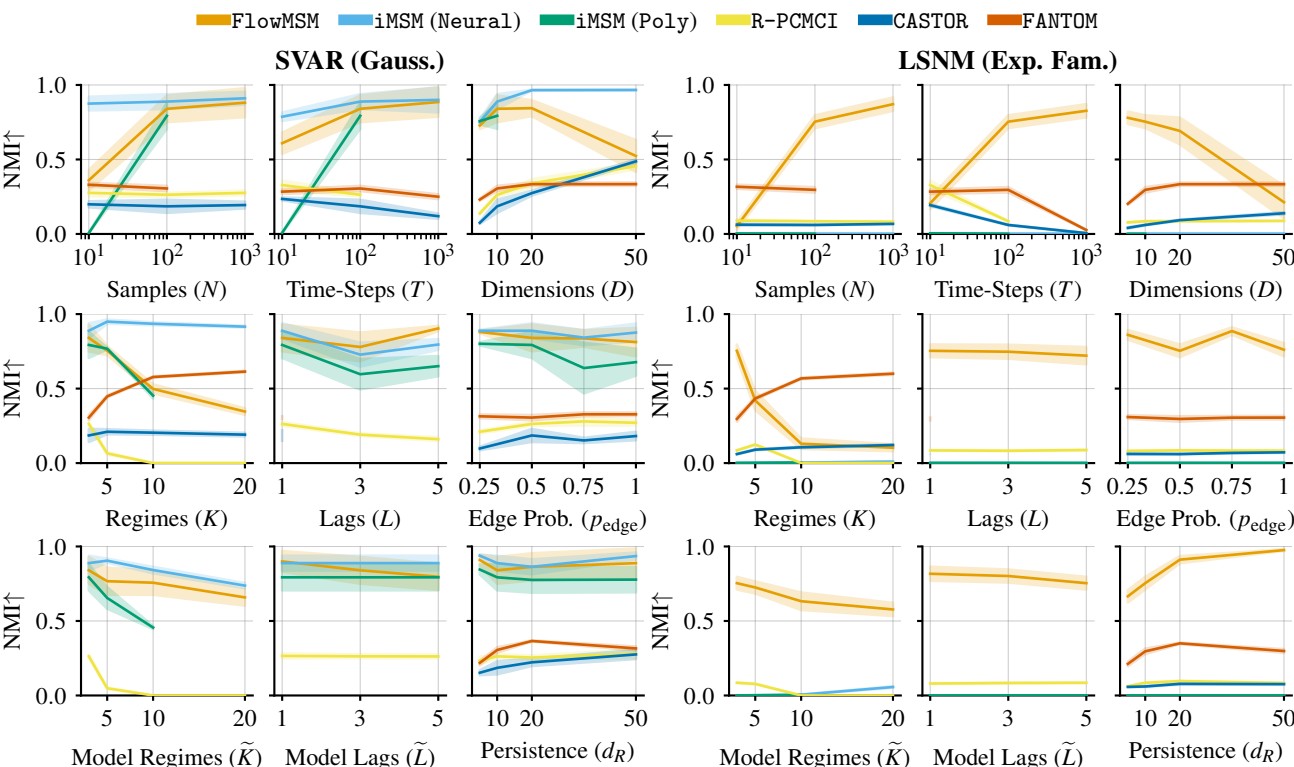

*Figure 10.* Sensitivity of regime detection to several data-generation parameters in Tab. 2 and the model number of regimes and lags, for both (*left*) the linear SVAR model with Gaussian noise and (*right*) the LSNM with exponential family noise.

## E.5. Hyperparameter Ablation Studies

We provide ablation studies to several data-generation parameters. We vary one parameter at a time compared to the "default" setting described in Tab. 2. For fairness, we do not engage in additional hyperparameter tuning, and we repeat the procedure for both our "easy" linear Gaussian SVAR model, and our "most challenging" LSNM with exponential family noise. Figs. 10 to 12 show experimental results for regime detection, causal discovery and computational running times.

To clarify, several baseline curves exhibit missing datapoints. For example, iMSM(Poly) appears to be unstable during training, as a result of exact parameter updates in the EM-algorithm, described in App. C. The implementation of R-PCMCI suffers from memory constraints and a large computational runtime, due to the number of data structures retained in memory and the number of conditional independence tests. Finally, FANTOM yields a large computational runtime, following the combination of a variational ELBO objective, estimation of regime-dependent conditional normalizing flows, and an augmented Lagrangian procedure for parameter optimisation.

Regarding regime detection in Fig. 10, we observe that FlowMSM benefits from an increase in sample size $(N)$ from small to medium, yet it seems to be saturated after $N = 100$, and similarly for the number of time-steps $(T)$. Our framework does not seem to scale well beyond 20-50 dimensions with the given amount of data and without additional hyperparameter tuning. This does not seem to be the case for the baseline methods in the Gaussian SVAR setting.

In the ablation for the oracle number of regimes $(K)$, we make sure to adjust the model number of regimes $(\widetilde{K})$ accordingly. While FlowMSM does not seem to scale well with the number of regime $(K)$, we surprisingly find iMSM and CASTOR to be fairly robust to such changes in the Gaussian SVAR setting. FANTOM even increases in NMI, although this is for a counterintuitive reason. That is, we empirically observe that this method rarely deviates from its initial window assignments in the shorter time-sequences, and it is initiated with a window length equal to the oracle regime persistence. This causes the estimated regime partition to mostly align with the oracle partition for the first $K$ windows, up until recurrent regimes occur. Hence, the high NMI score should be classified as deceptive. We performed some small-scale experimentation to test where this phenomenon persists, and found it to be invariant to increases in the number of iterations and alterations to the initialisation window size. We note that we are able to reproduce the results on synthetic data in Rahmani & Frossard (2025b)

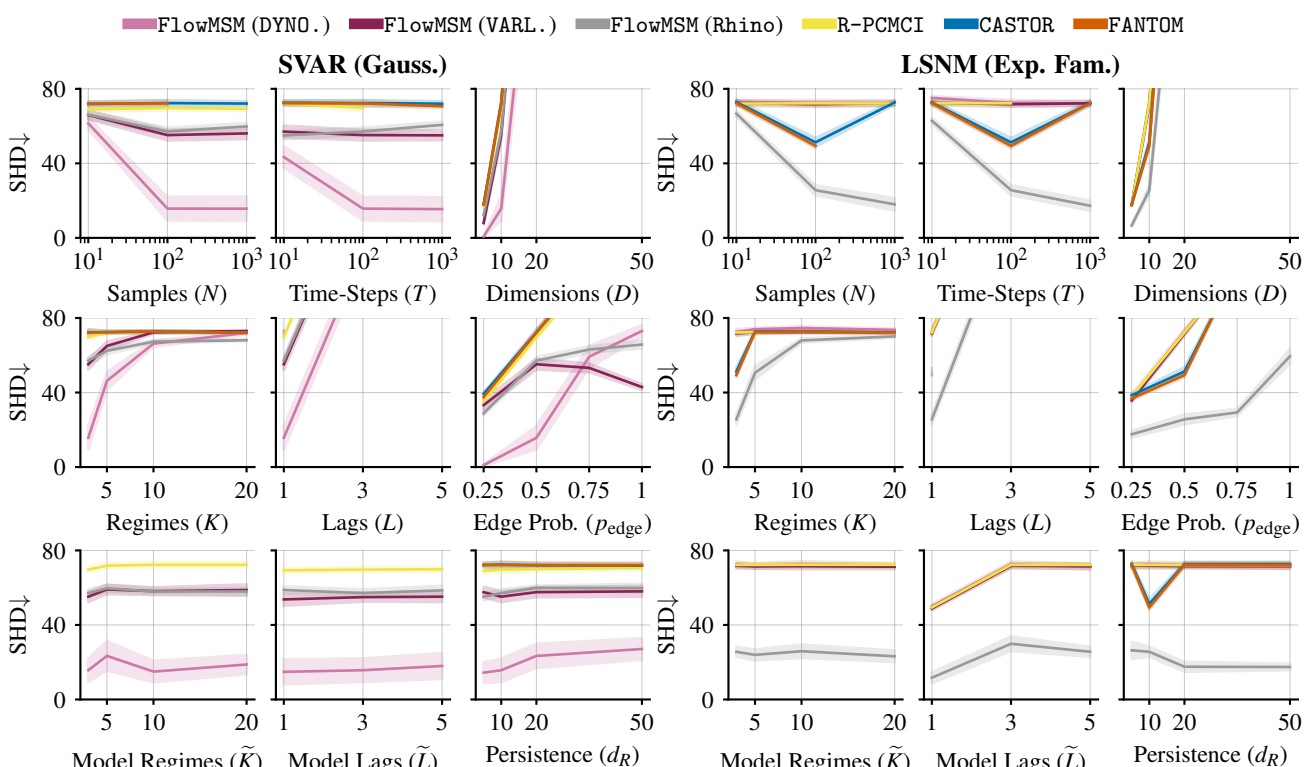

*Figure 11.* Sensitivity of causal discovery to several data-generation parameters in Tab. 2 and the model number of regimes and lags, for both (*left*) the linear SVAR model with Gaussian noise and (*right*) the LSNM with exponential family noise.

with the chosen set of hyperparameters, indicating meaningful differences with our synthetic data-generating process.

In the ablation for the model number of regimes ($\widetilde{K}$), we keep the oracle number of regimes ($K$) fixed while overspecifying the degrees of freedom for potential regimes. We leave out CASTOR and FANTOM here, since they do not inherit this hyperparameter. We observe a decreasing effect of overspecifying the model number of regimes. The experiment is similar for the model number of lags ($\widetilde{L}$), although CASTOR and FANTOM are omitted, since the available implementation does not appear to be functional for multi-lag dependencies. We observe little sensitivity of any method to the oracle number of lags ($L$), the model number of lags ($\widetilde{L}$), the sparsity of the synthetic graphs ($p_{\text{edge}}$) and the oracle regime persistence ($d_R$).

Regarding causal discovery in Fig. 11, the observations are similar in nature. For visualisation purposes, we chose to not display an SHD above 80 to not distort the scale compared to the other figures. The SHD metric is highly sensitive to the sparsity and dimension of the causal structure, such that it is not meaningful to make comparisons in the ablations for the oracle number of lags ($L$), the number of dimensions ($D$) and the graph sparsity ($p_{\text{edge}}$).

We observe that FlowMSM (DYNOTEARS) and FlowMSM (Rhino) greatly benefit from the increase in sample size and time-steps from 10 to 100 in the linear Gaussian SVAR and exponential family LSNM setting, respectively, yet seem to saturate beyond that point. The baselines do not show sensitivity to these parameters. When increasing the number of regimes ($K$), we observe deteriorating performance of FlowMSM (DYNOTEARS) and FlowMSM (Rhino) in the linear Gaussian SVAR and exponential family LSNM setting, respectively. We hypothesize that keeping the other parameters fixed, the amount of data per regime decreases, and performance converges to the setting with low sample size ($N$).

Regarding the running times in Fig. 12, we observe that on the default parameters defined in Tab. 2, FlowMSM runs for $\pm 200s$ when combined with DYNOTEARS or VARLiNGAM, or for $\pm 300s$ when combined with Rhino. This is competitive compared to baseline methods, where only iMSM (Neural) is faster, yet this baseline does not do causal discovery. All methods appear to follow roughly similar computational scaling laws with respect to the sample size ($N$) and number of dimensions ($D$), for example, although some baselines suffer from memory constraints.

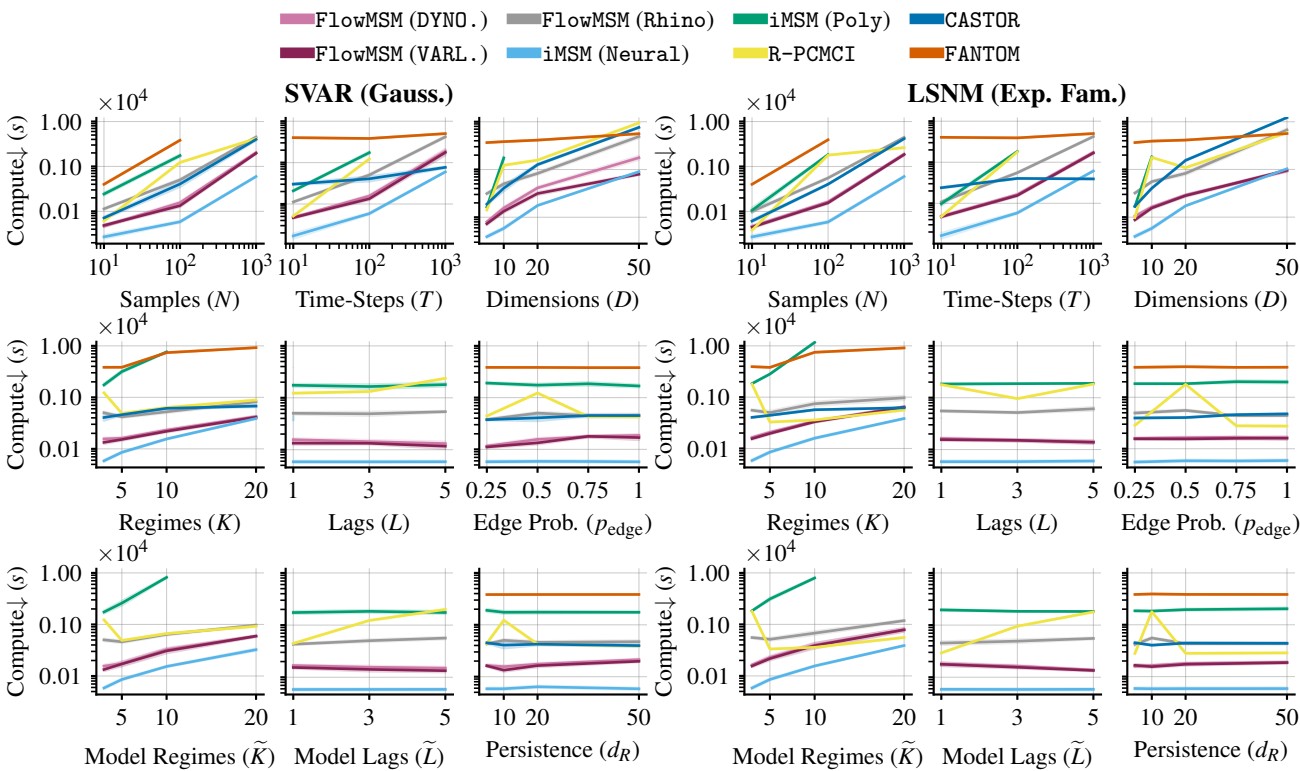

*Figure 12.* Sensitivity of computation runtime ($s$) to several data-generation parameters in Tab. 2 and the model number of regimes and lags, for both (*left*) the linear SVAR model with Gaussian noise and (*right*) the LSNM with exponential family noise.

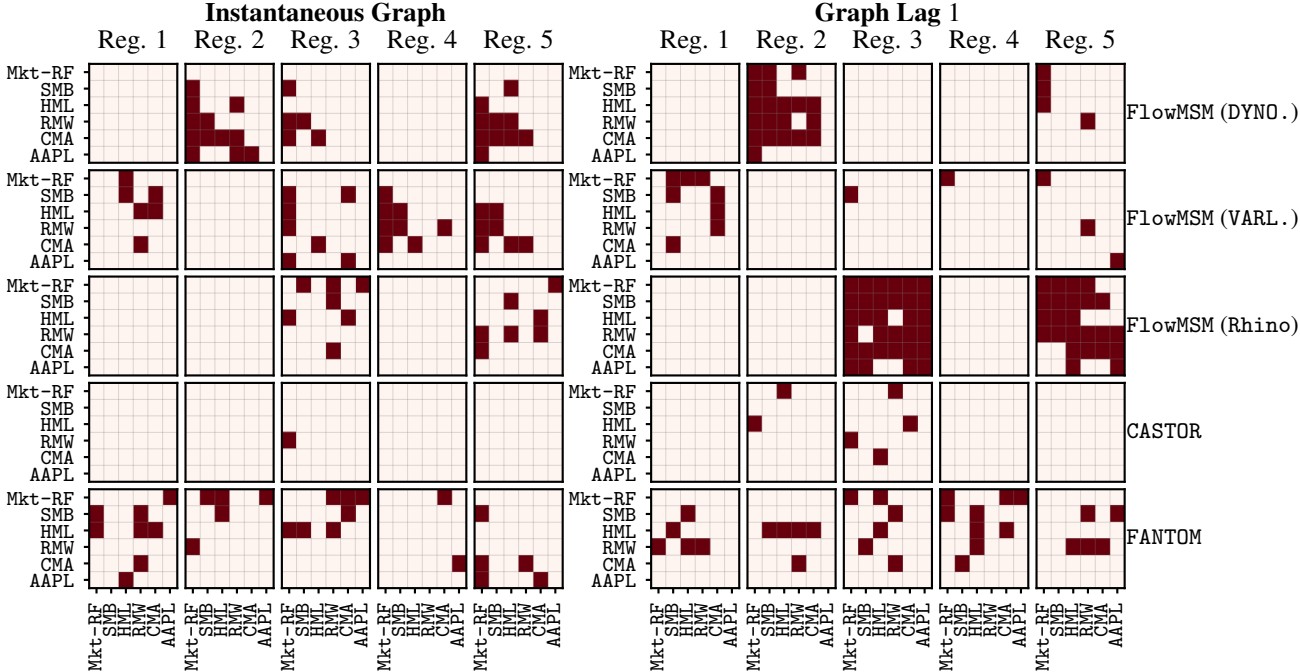

*Figure 13.* Adjacency matrices for the estimated window graphs corresponding to the regimes shown in Fig. 5. The methods are ran on daily data of the Fama-French model and `AAPL` stock. The index $(i, j, k, l)$ indicates the (lagged) edge $X_{t-l,i} \rightarrow X_{t,j}$ in regime $k$.

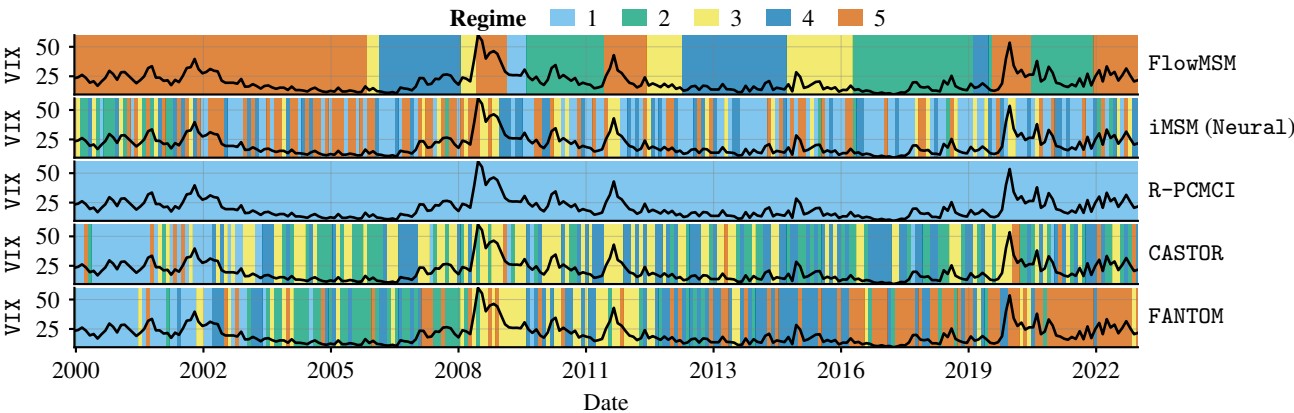

*Figure 14.* Estimated regimes on monthly data from the Fama-French model and `AAPL` stock, with an overlay of the `VIX` volatility index.

## E.6. Fama-French Five-Factor Asset-Pricing Model

In the main text, we explore regime detection in asset pricing to qualitatively evaluate the Fama-French five-factor asset-pricing model (Fama & French, 2015). Following Sadeghi et al. (2024), we add excess returns of Apple's stock (`AAPL`). We run `FlowMSM` and baselines using the random initialisation seed 0, without peeking at the results for fairness of the evaluation. For interpretability, we plot the `VIX` volatility index over the estimated regimes, although it is omitted in training.

Even though the Fama-French model is posited as a *statistical* rather than a *causal* model, we investigate its causal implications here. That is, under the efficient market hypothesis, causal dependencies across time steps should be absent. However, we argue that perhaps periods of market distress may give rise to arbitrage opportunities. Moreover, we hypothesise that the Fama-French risk factors act as causal drivers of `AAPL` excess returns, rather than the reverse.

The estimated window graphs in Fig. 13 correspond to the regimes displayed in Fig. 5 in the main text. For `FlowMSM`, we interpret regime 4 as an indicator of stable financial markets, while regime 5 signals market distress. The other regimes are effectively redundant, since these only contain a tiny number time steps. We find only partial support for the aforementioned hypotheses on the associated window graphs. For example, `FlowMSM (DYNOTEARS)` and `FlowMSM (Rhino)` produce empty graphs in regime 4, and find several causal connections in regime 5. This is in line with the efficient market hypothesis and our additional hypothesis on how it might breach in financially turbulent times. In a similar spirit, `FlowMSM (VARLiNGAM)` mostly finds instantaneous connections instead of lagged dependencies in both regime 4 and 5, which roughly aligns with the efficient market hypothesis on the low predictive signal in historic data.

However, we do not find convincing evidence that the Fama-French risk factors act as causal drivers of `AAPL` excess returns, or that the risk factors are *independent* causal factors explaining excess return on `AAPL` stock. On the contrary, we find few edges pointing into `AAPL`, and many instantaneous edges between the risk factors. In an extreme instance, `FlowMSM (Rhino)` outputs a dense graph in regime 5 that is challenging to assess qualitatively. This suggests that causal interpretations of our estimated causal structures of the Fama-French model in this dataset are limited in scope.

For completeness, we repeat the analysis on *monthly* instead of *daily* data, leaving 276 time steps. The estimated regimes and window graphs on monthly data are visualised in Figs. 14 and 15. `FlowMSM` and baselines show vastly different behaviour, yet we fail to identify meaningful interpretations of the estimated regimes and regime-dependent window graphs for any of the methods. We posit the available time steps are too few to derive substantiated conclusions.

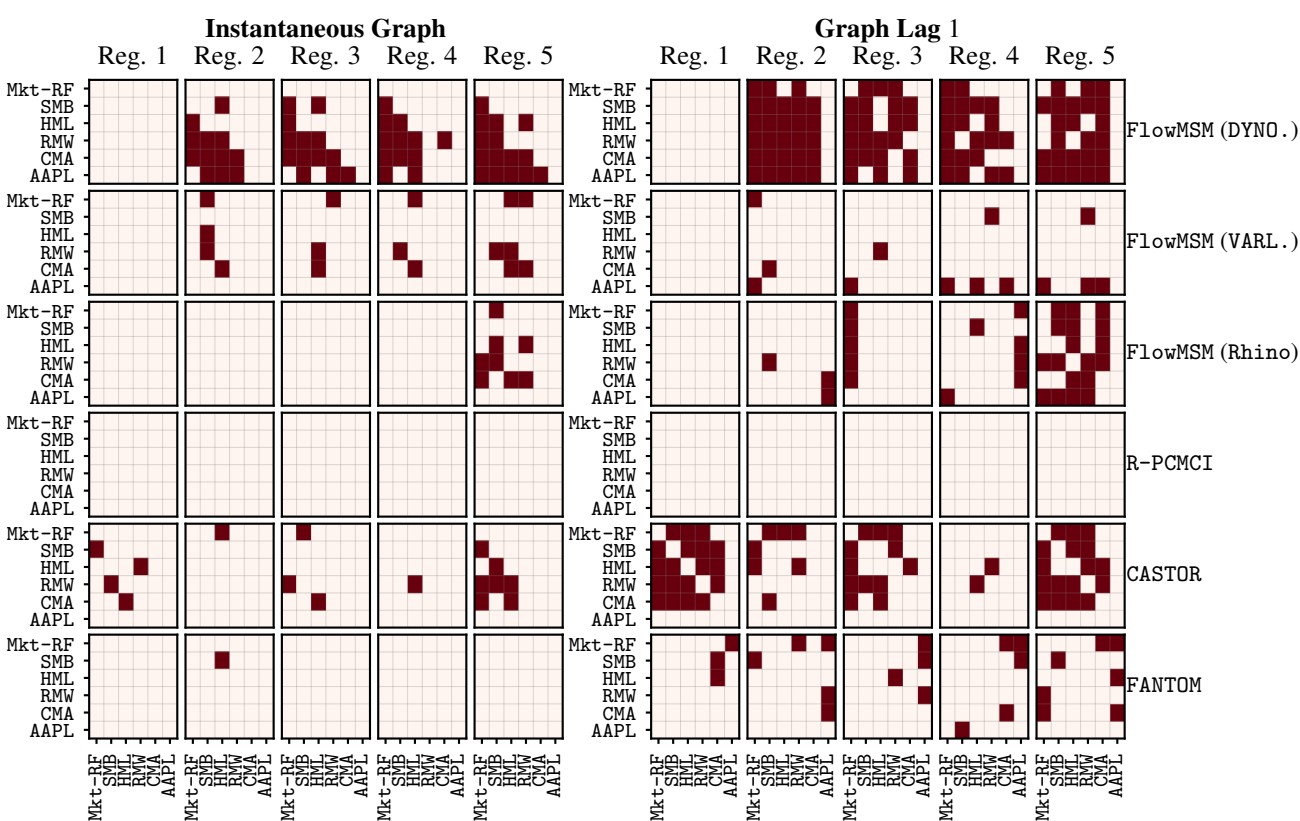

*Figure 15.* Adjacency matrices for the estimated window graphs corresponding to the regimes shown in Fig. 14. The methods are ran on monthly data of the Fama-French model and `AAPL` stock. The index $(i, j, k, l)$ indicates the (lagged) edge $X_{t-l,i} \to X_{t,j}$ in regime $k$.

