# OpenReview forum: "Identifiable Markov Switching Models with Instantaneous Effects and Exponential Families"
_ICML.cc/2026/Conference — ICML 2026 regular_

### Official Review · Reviewer_Y5aq · 2026-03-09

**Soundness:** 3
**Presentation:** 2
**Significance:** 3
**Originality:** 3
**Overall Recommendation:** 5
**Confidence:** 3

**Summary:**

This paper introduces a model for nonstationary temporal processes based on a latent variable with a Markov-switching dynamics. The authors prove their model to be identifiable under several conditions. They introduce a inference procedure called FlowMSM that allows to recover the latent states. They validate their method on synthetic datasets generated under different scenarios. Finally, they provide an illustration on a financial dataset.

**Compliance With Llm Reviewing Policy:**

Affirmed.

**Final Justification:**

I think that the identifiability theory for Markov-switching models developed by the authors is a valuable contribution. I had some concerns regarding the empirical validation, but they have been addressed during the rebuttal.
Therefore I increased my score from 4 to 5.

**Key Questions For Authors:**

• The authors mention that the hyperparameter K « can be optimised in a hold-out validation set » but it seems that in practice, it is not estimated but fixed and that a sensitivity analysis is provided. Is this correct? Would it be possible to estimate it for instance with a standard penalized likelihood criterion?
    • How has the value of K be chosen in the real-world dataset? Has it been set to 2 or is it estimated? In particular, the results of  FlowMSM appear easier to interpret but the number of hidden states is smaller than for competitors, leading to less dynamics changes.
    • Since the overfitting of the parameter appears as a critical point, I think this at least be stated in the main text instead of being only in Appendix E.5.

**Limitations:**

The limitations mentioned are quite vague and could be discussed in more details.

**Strengths And Weaknesses:**

**Strengths**:
    • The authors propose a flexible framework to model non-stationary processes, that can be used in many applications.
    • The identifiable result is, to the best of my knowledge, novel and quite general. I think it is a valuable contribution for the interpretability of such switching-Markov models.
    • The authors provide a code that can be used to reproduce the results.

**Weaknesses**:
    • The choice of the number of hidden classes K appears as a critical point in the inference method. While the authors state in the main document « Importantly, we presume no knowledge of the number of regimes K », the numerical studies displayed in the appendix show that overfitting this parameter has a strong impact on the quality of the results.
    • The organization of the paper is quite unusual and sometimes not easy to follow. In particular, a large amount of space is dedicated to the introduction of the assumptions for establishing the identifiability result, leaving very little space for the inference procedure. Most of the necessary details to understand the inference method are postponed to appendices, and should at least be referred to clearly in the main document to facilitate the reading.

---

> ### Author Rebuttal · Authors · 2026-03-30
>
> We thank the reviewer for their time and effort reviewing our work.
>
> ## Choosing the model number of regimes ##
> In Fig. 3, we set $\widetilde{K}=3$ for $\texttt{FlowMSM}$ and $\texttt{iMSM}$. This hyperparameter does not exist in $\texttt{CASTOR}$ and $\texttt{FANTOM}$, where instead the initial window size (here set to 1000, given a sequence length of 5786) mostly determines the estimated number of regimes. Indeed this setup might result in more estimated regimes for $\texttt{CASTOR}$ and $\texttt{FANTOM}$, hence we increase $\widetilde{K}=5$ to provide a more fair comparison in a replacement of Fig. 3 (https://github.com/anonymousacademic00/flowmsm/blob/main/rebuttal_1.png). The estimated regimes of $\texttt{FlowMSM}$ are approximately the same as for $\widetilde{K}=3$, but for both instances of $\texttt{iMSM}$ we now observe more frequent regime switches and a higher number of estimated regimes.
>
> > … the numerical studies displayed in the appendix show that overfitting (K) has a strong impact on the quality of the results.
>
> There might be a misunderstanding in the interpretation of Fig. 8, where in the bottom-left plot we overfit the model number of regimes $\widetilde{K}$, while keeping the oracle number of regimes $K=3$ fixed. Although we observe some slight decrease in performance for higher $\widetilde{K}$, overshooting the number of regimes still seems a reasonable strategy. We believe the reviewer might have looked at the middle-left plot in Fig. 8, where we simultaneously increase the model number of regimes $\widetilde{K}$ and the oracle number of regimes $K$, without overshooting the model number of regimes.
>
> > … (K) is not estimated but fixed (...) Would it be possible to estimate it for instance with a standard penalized likelihood criterion?
>
> The reviewer is correct that the hyperparameter $\widetilde{K}$ is fixed and set equal to the oracle value throughout most of our experiments (see line 1280). This is to provide a fair comparison to other methods with the same hyperparameter. Two strategies in practice are (1) to optimise $\widetilde{K}$ using an elbow method for the estimated likelihood on a hold-out validation set (see line 293), or (2) to overshoot the model number of regimes compared to the oracle value, as discussed in the previous paragraph. If the oracle mixture distribution is in the hypothesis class and $\widetilde{K}\geq K$, then the GEM likelihood objective in Eq. (39) is maximised when $\widetilde{K}=K$, hence already implicitly incorporating penalisation of solutions where the estimated number of regimes is unequal to $K$. In practice, the GEM algorithm is guaranteed to converge to a local optimum (Dempster et al., 1977).
>
> For computational efficiency, (2) is the recommended strategy in practice, and we will adjust the main text to make this clear. This strategy is illustrated in the real-world financial dataset, where the oracle $K$ is unknown, see the replacement of Figure 3 introduced above. While the initial number of regimes is $\widetilde{K}=5$, nearly all time-steps are assigned to either one of two regimes, illustrating that in practice our method is able to discard unused regimes.
>
> > the overfitting of the parameter appears as a critical point
>
> As mentioned before, we believe there to be a misunderstanding of Fig. 8 in App. E.5. Furthermore, we will adjust the main text to reflect that overshooting $\widetilde{K}$ is the recommended strategy in practice.
>
> ## Paper structure ##
> We will extend and improve the explanation of the inference procedure in the revised version. At the same time, we believe our identifiability theory to be at the heart of our contribution, therefore dedicating substantial space to explain the underlying assumptions, which can be abstract for readers who are unfamiliar with either the finite mixture model or causal literature. We plan to re-examine this balance in the revised version.
>
> ## Limitations ##
> We will add a discussion of the limitations of our approach in the revised version, including the practical applicability of our assumptions. We provide a short summary here. The exponential family noise is a key assumption in our work, both in our identifiability theory and as building blocks of the normalising flows in $\texttt{FlowMSM}$, which is a potential limitation. As discussed in the rebuttal for reviewer qtR9, we included new experiments in misspecified settings, showing that the performance of $\texttt{FlowMSM}$ degrades, yet so do the baseline methods. Such complex non-stationary time-series warrant novel theory and methods. Moreover, our identifiability theory requires that the regime variable $R_t$ is not affected by the observations $\textbf{X}_{0:t-1}$. Balsells-Rodas et al. (2025) relax this assumption, but only for Gaussian settings. Extending to exponential families is not trivial and an interesting future direction.

---

> > ### Author Rebuttal · Reviewer_Y5aq · 2026-04-01
> >
> > I thank the authors for the clarifications and I appreciate the additional numerical validation and visualization.
> > I agree with the authors regarding the identifiability theory being the core of their contribution and the rebuttal allowed to address my concerns regarding the empirical validation.
> > Therefore I will increase my score from 4 to 5.

---

### Official Review · Reviewer_vFiC · 2026-03-12

**Soundness:** 3
**Presentation:** 3
**Significance:** 3
**Originality:** 3
**Overall Recommendation:** 4
**Confidence:** 3

**Summary:**

This paper studies the identifiability of the Markov Switching Model (MSM), a class of hidden Markov models. The authors establish identifiability of both the latent regimes and the regime-dependent causal structures under temporal regime dependencies. Based on this result, they propose the FlowMSM framework, which can be combined with causal discovery methods to recover temporal causal structures. Experimental results on both simulated and real-world datasets demonstrate the effectiveness of the proposed approach.

**Compliance With Llm Reviewing Policy:**

Affirmed.

**Key Questions For Authors:**

See Weakness.

**Limitations:**

NAN

**Strengths And Weaknesses:**

##Strengths

1. The paper is generally well written and clearly organized.

2. The authors provide a good review of the related literature.

3. The identifiability results look sound and extend the identification results of existing models (e.g.,  finite-state HMMs, LSNMs). Specifically, the LSNMs assume that there are no latent confounders, while the author's approach relaxes this assumption by introducing the identifiability of the mixture model.

##Weakness:

1. The contribution is primarily theoretical. However, the computational efficiency of the proposed estimation procedure and its practical scalability are not sufficiently discussed.

2. The identifiability results rely on several strong assumptions (e.g., Assumption 3.4). It would be helpful to discuss how realistic these assumptions are in practical applications.

3. The paper does not clearly explain the relationship between the theoretical identifiability results and the consistency of the proposed estimation procedure. In particular, it would be useful to clarify whether the estimator is guaranteed to be consistent under the stated assumptions.

---

> ### Author Rebuttal · Authors · 2026-03-30
>
> We thank the reviewer for their thoughtful remarks. We reply to the points raised one by one.
>
> ## Computational efficiency ##
> The running times of our method and all baselines are displayed in Fig. 10 in the appendix, together with how running times scale with several data generation and model hyperparameters. On the default parameters specified in Table 2, $\texttt{FlowMSM}$ runs for +-200s when combined with $\texttt{DYNOTEARS}$ or $\texttt{VARLiNGAM}$, or for +-300s when combined with $\texttt{Rhino}$. Especially for the combination with $\texttt{DYNOTEARS}$ and $\texttt{VARLiNGAM}$, the running time of our method is competitive compared to baselines, where only $\texttt{iMSM (Neural)}$ is faster, yet that baseline does not do causal discovery. All methods appear to follow roughly similar computational scaling laws with respect to e.g. the sample size $N$ and number of dimensions $D$, although some baselines suffer from memory constraints.
>
> ## Motivation of assumptions in practical applications ##
> We believe Assumption 3.4 is a mild restriction that is likely satisfied in practical applications. In essence, this assumption requires that sufficient variability between regimes exists in some finite polynomial subspace of the potentially infinite space spanned by the sufficient statistic. Hence, for this assumption to be violated, two regimes would need to be indistinguishable in any finite polynomial subspace, and the differences would *only* materialize in the infinite and/or non-polynomial remainder term. We struggle to construct a concrete counterexample where this is the case, and although such edge cases definitely exist, we believe them to be merely theoretical.
>
> Regarding the other assumptions, conditional causal stationarity is inherent to the regime-dependent nature of our proposed setting, and (approximately) holds in many real-world settings, such as sleep-stage dynamics, ENSO and glucose fluctuations mentioned in the introduction. Furthermore, acyclicity, causal sufficiency, causal Markov and faithfulness are standard in the causal literature, where causal sufficiency in our work is assumed conditional on the latent regime, which is a relaxation of the standard assumption. We would like to refer to our response to Reviewer qtR9 for a discussion on exponential family noise, which is indeed a key assumption in our theory and method and could be interesting to relax in future work. Continuing, we do acknowledge that the restriction to real-analytic diffeomorphisms in Assumption 3.2 somewhat limits generality of the functional model class, yet it is essential in our identifiability theory to prevent two regimes from collapsing to the same transition distribution. Finally, Assumption 3.3 is meant to exclude some rare regimes exploiting distributional symmetries, such as rotations of isotropic Gaussians, which we do not expect to arise in practical applications.
>
> ## Consistency of estimator ##
> We acknowledge some ambiguity as to how to interpret the question of the reviewer, and therefore we provide several answers that we will also include in the revised version of the paper. First, we address the conceptual fit/consistency between our identifiability theory and our $\texttt{FlowMSM}$ estimation method. As mentioned in line 1123, the normalising flows employed to estimate the regime-dependent transition distributions fit well with our theoretical narrative of transforming exponential family noise through regime-dependent real-analytic diffeomorphisms. We do highlight that the invertible transformations estimated by a normalising flow do not equal the functional relationships in a regime-dependent SCM.
>
> Second, the GEM algorithm for maximum likelihood estimation of finite mixture distributions is guaranteed to converge to a local optimum of the likelihood objective (Dempster et al., 1977) (see line 297). Hence, the reviewer is right to question consistency, since there is no formal guarantee that our method converges to the oracle solution as $N\to\infty$. However, this is common in applications of the GEM algorithm, following from the use of gradient ascent, which is widely applied in machine learning settings, as well as similar methods such as $\texttt{iMSM}$ and $\texttt{FANTOM}$.

---

> > ### Author Rebuttal · Reviewer_vFiC · 2026-04-05
> >
> > I thank the authors for their rebuttal and will keep my score.

---

### Official Review · Reviewer_qtR9 · 2026-03-13

**Soundness:** 3
**Presentation:** 3
**Significance:** 3
**Originality:** 3
**Overall Recommendation:** 4
**Confidence:** 2

**Summary:**

The paper studies non-stationary time series with discrete latent regimes, modeling them with Markov switching models. The authors establish identifiability of latent regimes and regime-dependent causal structures under nonlinear lagged and instantaneous effects, and independent noise from the exponential family. The authors develop a regime detection framework that can be paired with any stationary causal discovery method to recover regime-dependent causal structures. The method is validated on synthetic and real-world financial economics data.

**Compliance With Llm Reviewing Policy:**

Affirmed.

**Final Justification:**

Most of my concerns have been addressed and I maintain my positive score.

**Key Questions For Authors:**

- What is the running time of the method?
- Could the authors discuss the challenges of extending their result beyond the exponential family, and whether such an extension might be feasible in future work?

**Limitations:**

A discussion of limitation was not found

**Strengths And Weaknesses:**

Strengths:
- The paper is overall well written and easy to follow. The notations are clearly defined and the assumptions are stated clearly with explanation before introducing the result.
- Non-stationarity is common in real-world applications. The work helps make causal discovery more practical and applicable in realistic real-world scenarios.
-  The theoretical result appears to be sound and rigorous (although I did not carefully check all proof details), and the approach is reasonable.

Weaknesses:
- The exogenous noise is assumed to belong to exponential family, which might limit the applicability of the method/result.
- The experiment on real-world financial data is interesting but the explanation is rather brief. Providing a more detailed explanation of the results/insights would help clarify the practical implications of the method.
- Providing empirical studies on more real-world dataset, both for regime detection and causal discovery, would strengthen the experiment part.

---

> ### Author Rebuttal · Authors · 2026-03-30
>
> We thank the reviewer for their careful review.
>
> ## Exponential families and beyond ##
> Exponential family noise is indeed a key assumption in our work. To achieve identifiability of finite mixture distributions, linear independence is a necessary and sufficient condition (Yakowitz and Spragins, 1968). Hence, in full generality, any family of regime-dependent SCMs that induces linear independent PDFs induces identifiable MSMs. However, linear independence is an abstract notion, and it is often unclear when it is satisfied, i.e., in terms of restrictions on the exogenous noise and functional model class in a regime-dependent SCM. Making this notion concrete in our setting is part of our contributions.
>
> We focus on exponential families since Barndorff-Nielsen (1965) introduced sufficient conditions to obtain linearly independent functions in this setting. Relaxing these conditions generally destroys linear independence, for example when the exponential family cannot be reparametrized to one with a common sufficient statistic, which in our work is ruled out by Ass. 3.4. Moving beyond exponential families, while preserving linear independence, is a non-trivial task that we think might warrant a vastly different proof strategy. Nonetheless, it might be feasible given the vast literature on finite mixture distributions that sprouted from the 1960s, and this would be interesting for future work.
>
> To explore the empirical feasibility of extending $\texttt{FlowMSM}$ beyond exponential families, we include **new experiments with non-exponential noise distributions** (https://github.com/anonymousacademic00/flowmsm/blob/main/rebuttal_2.png), i.e., the $\mathcal{U}[-\sqrt{6},\sqrt{6}]$ and $\text{Cauchy}(0,1)$ distributions. The parameters are chosen such that both are symmetric around zero, and for the Uniform distribution with unit variance, in line with the experiments in Fig. 6. We observe that $\texttt{FlowMSM}$ still outperforms baselines, yet performance degrades substantially compared to Fig. 6, except for the SVAR model with uniform noise. For the Cauchy distribution, we observe a small number of large outliers in the generated time-series, which likely destabilizes training for all methods. We will include these experiments in the revised version of the paper.
>
> ## Explanation of financial data ##
> We provide a more detailed explanation in App. E.6, where the causal implications of the Fama-French model are explored, but for completeness we summarize it here. We hypothesize that (1) temporal dependencies are absent due to the efficient market hypothesis, and (2) the five risk factors could be causal drivers of $\texttt{AAPL}$ excess returns, rather than the reverse. However, we only find partial support for these hypotheses, suggesting causal interpretations of the Fama-French model on this dataset are limited in scope. Furthermore, on monthly instead of daily data, we fail to identify meaningful interpretations for any method, likely due to a limited amount of data.
>
> The regime estimates in Fig. 3 are explained on p. 8. For $\texttt{FlowMSM}$, we interpret one regime as an indicator of stable financial markets, while the other signals market distress. The baselines do not seem to recover these regimes, and a meaningful interpretation of the (in)frequent regime switches remains unclear. For the $\texttt{iMSM}$ variants, the financial time-series might not be easily approximated by Gaussian distributions, whereas $\texttt{FlowMSM}$ appears to more flexibly accommodate this.
>
> ## More empirical studies ##
> While we agree that this would strengthen the paper, the limited evaluation on real-world settings is a well-known issue in causal discovery, since it is challenging to find datasets with ground truth causal graphs. Even the widely used benchmark by Sachs et al., (2005) only provides the consensus of experts in 2005. This is even more complex in non-stationary environments, where the regimes are unknown and we have multiple regime-dependent graphs. A dataset we considered is the TUH EEG Seizure Detection Corpus (Shah et al., 2018) used in (Rahmani and Frossard, 2025b), but unfortunately, the preprocessing the authors used for this dataset is not available publicly. Nonetheless, we will try to include it in the revised version of the paper.
>
> (Sachs, K., et al.) Causal Protein-Signaling Networks Derived from Multiparameter Single-Cell Data. Science 308:523-529, 2005.
>
> (Shah, V., et al.) The Temple University Hospital Seizure Detection Corpus. Frontiers in Neuroinformatics, 12, 2018.
>
> ## Running time ##
> We report running times of all methods in Fig. 10 in the appendix, showing that our methods are competitive in most settings, and further discussed in the rebuttal for reviewer vFiC.
>
> ## Limitations ##
> We discuss some limitations of our approach in the rebuttal for reviewer Y5aq, and will include the discussion in the revised version of the paper.

---

> > ### Author Rebuttal · Reviewer_qtR9 · 2026-04-03
> >
> > Thanks for the response. Most of my concerns have been addressed and I maintain my positive score.

---

### Decision · Program_Chairs · 2026-04-30

**Decision:**

Accept (regular)

**Comment:**

Reviewers aknowledged the importance of the addressed problem, the soundness and novelty of the theory, the relevance and quality of the experiments. While some weaknesses have been pointed out and discussed, notably regarding the clarity of main text and inference details, they not justify rejection. I recommend acceptance and encourage the authors to use the reviewers's suggestions to improve the manuscript for the camera ready version.